# Structured Low-Rank Tensors for Generalized Linear Models

**Batoul Taki**  *batoul.taki@rutgers.edu*
*Department of Electrical and Computer Engineering*
*Rutgers University-New Brunswick*

**Anand D. Sarwate**  *anand.sarwate@rutgers.edu*
*Rutgers University-New Brunswick*

**Waheed U. Bajwa**  *waheed.bajwa@rutgers.edu*
*Rutgers University-New Brunswick*

**Reviewed on OpenReview:** *https://openreview.net/forum?id=qUxBs3Ln41*

## Abstract

Recent works have shown that imposing tensor structures on the coefficient tensor in regression problems can lead to more reliable parameter estimation and lower sample complexity compared to vector-based methods. This work investigates a new low-rank tensor model, called Low Separation Rank (LSR), in Generalized Linear Model (GLM) problems. The LSR model – which generalizes the well-known Tucker and CANDECOMP/PARAFAC (CP) models, and is a special case of the Block Tensor Decomposition (BTD) model – is imposed onto the coefficient tensor in the GLM model. This work proposes a block coordinate descent algorithm for parameter estimation in LSR-structured tensor GLMs. Most importantly, it derives a minimax lower bound on the error threshold on estimating the coefficient tensor in LSR tensor GLM problems. The minimax bound is proportional to the intrinsic degrees of freedom in the LSR tensor GLM problem, suggesting that its sample complexity may be significantly lower than that of vectorized GLMs. This result can also be specialised to lower bound the estimation error in CP and Tucker-structured GLMs. The derived bounds are comparable to tight bounds in the literature for Tucker linear regression, and the tightness of the minimax lower bound is further assessed numerically. Finally, numerical experiments on synthetic datasets demonstrate the efficacy of the proposed LSR tensor model for three regression types (linear, logistic and Poisson). Experiments on a collection of medical imaging datasets demonstrate the usefulness of the LSR model over other tensor models (Tucker and CP) on real, imbalanced data with limited available samples.

## 1 Introduction

In machine learning, regression models are used to understand the relationship between a set of independent variables (also known as covariates) and a dependent outcome. More formally, given a vector of covariates, $\mathbf{x}$, and outcome, $y$, jointly distributed according to $\mathbb{P}_{\mathbf{x}y}$, the goal is to produce a function that will predict $y$ when given $\mathbf{x}$. Under the Mean Squared Error (MSE) criterion, the solution would be finding the conditional mean $\mathbb{E}_{y|\mathbf{x}}[y|\mathbf{x}]$, which is obtained by modeling the conditional probability $\mathbb{P}_{y|\mathbf{x}}$ and estimating the model class parameters. For example, in linear regression, the predictor is given by $\mathbf{b}^T \mathbf{x}$, where the model class parameters are denoted by a vector $\mathbf{b}$ and estimated through a set of $n$ training data samples $\{\mathbf{x}_i, y_i\}_{i=1}^n$, where $\mathbf{x}_i$ is the $i^{th}$ sample vector of covariates. Different regression models suit different types of prediction problems: some common examples are linear, logistic and Poisson regression, all of which fall under a broader class of models called Generalized Linear Models (GLMs). GLMs were introduced to encompass classes of models that cannot be appropriately modeled as a simple 'linear-response model' (McCullagh & Nelder, 2019). In particular, GLMs refer to a parametric statistical framework that models the conditional

probability of a scalar response variable $y_i$ that follows an exponential family distribution. Because the exponential family distribution encompasses a large range of widely-used distributions, GLMs allow one to study a broader class of regression problems.

GLM regression models are used for wide array of datasets in a multitude of applications. However, modern-day technologies are creating data-intensive environments and collecting increasingly high-dimensional data. In particular, we often encounter data in the form of variegated and structured multi-dimensional arrays (tensors), where the number of available data samples is far smaller than the number of variables (the dimensionality of the data). Prominent examples of two-dimensional arrays include biological imaging data such as electroencephalography (EEG) and fiber-bundle imaging (Dumas et al., 2019). Examples of three-dimensional arrays include functional Magnetic Resonance Images (fMRI) (Bellec et al., 2017) and Magnetic Resonance Angiography (MRA) (Yang et al., 2020). Though such data has been used in many instances throughout the literature (Li et al., 2018; Hung & Wang, 2013; Zhou et al., 2013; Dumas et al., 2019), classical parameter estimation methods assume vector-structured covariates and estimate a corresponding coefficient vector. There are two major concerns associated with multidimensional (a.k.a. tensor) data and its vectorization. First, vectorizing data that was originally in tensor form destroys its underlying structure (which often contains rich information valuable for regression analysis). Secondly, the resulting vector model exhibits a very large number of parameters, in the sense that in the high-dimensional setting the GLM regression model becomes ill-posed. This is an instance of 'the curse of dimensionality' (Hung & Wang, 2013; Zhang & Jiang, 2018).

A common solution to the curse of dimensionality in tensor GLM problems is to impose structure on the model parameters: this is the focus of this work. If the covariates are *tensor* structured, we expect to estimate coefficient tensors whilst exploiting the multidimensional structure of the data and rich information lying in the correlation between tensor modes. Common structures include: the addition of a sparsity regularizer or a low-rank inducing regularizer to the regression problem (Abramovich & Grinshtein, 2018; Seber & Lee, 2003; Zhang & Jiang, 2018; An & Zhang, 2020; Raskutti et al., 2019); the imposition of some tensor factorisation on the model parameters (Ahmed et al., 2020; Li et al., 2018; Zhou et al., 2013; Zhang et al., 2020; Tan et al., 2013; Zhang & Jiang, 2016; Wu et al., 2022; Taki et al., 2021); or both of the above. The imposition of such structures should ultimately lead to the estimation of fewer parameters, improving the computational complexity and performance of our parameter estimation problem.

Two commonly used tensor factorisations are the CANDECOMP/PARAFAC (CP) and Tucker decompositions (Kolda & Bader, 2009). These decompositions impose a compact structure on the coefficient tensors, thereby restricting the class of possible solutions in the parameter estimation problem. Compared to simple vector regression, these decompositions can decrease the number of training samples needed for reliable coefficient estimation (also referred to as 'sample complexity'). A smaller sample complexity can lower the variance of the model, yet the restrictive nature of these decompositions can also reduce the representation power of the coefficient tensors for many classes of tensors, causing a non-favourable bias-variance trad-eoff. A less studied decomposition – particularly in regression works – is the Block Tensor Decomposition (BTD) (De Lathauwer, 2008), which can be expressed as a Tucker decomposition with block diagonal core tensor. We further discuss BTD in relation to our work in Section 3.

In this work we impose a new decomposition on the coefficient tensors that we will promptly refer to as the Low Separation Rank (LSR) model. In fact we will show that the LSR decomposition is a generalization of the Tucker decomposition, and a special case of the BTD model (De Lathauwer, 2008). The LSR model maintains a lower sample complexity than vectorization-based tensor GLM regression but is less restrictive than Tucker or CP models of same sized core tensor. Though estimating an LSR-structured tensor introduces a greater sample complexity than the aforementioned decompositions, this increase is compensated by a stronger representation power, leading to better parameter estimation performance. In other words, we show that the increase in variance of the LSR model is conquered by its stronger representation power for a larger class of tensors (ergo, its decrease in bias), leading to a more favourable bias-variance trade-off (model compactness vs. representation power).

## 1.1 Contributions

In this paper we make the following contributions. We introduce the LSR-structured tensor problem under the GLM framework that we appropriately denote as LSR-TGLM. GLMs encompass various flavours of regression including linear, logistic and Poisson. We focus on the high-dimensional setting and discuss the various parameters of an LSR-structured tensor (such as 'rank' and 'separation rank', terms we will introduce in Section 2) that reduce the sample complexity of parameter estimation in GLMs. We also compare sample complexities between different tensor decomposition models and the LSR model.

Additionally, we explore two problems at the core of this work. First, we propose a parameter estimation algorithm (which we name LSRTR) for the LSR-TGLM problem. The main idea is that parameter estimation in GLMs can be achieved through Maximum Likelihood Estimation (MLE) (McCullagh & Nelder, 2019). However, for structured tensor settings, such as estimating CP or Tucker-structured tensors, the objective function of the MLE problem is highly non-convex (Li et al., 2018; Zhang & Jiang, 2016; Zhou et al., 2013; Tan et al., 2013). This is also true when the tensor is LSR-structured. To overcome this, we observe that the problem can be partitioned into several convex sub-problems that can then be solved alternately. On the basis thereof we propose a block coordinate descent algorithm to find the MLE of the LSR-structured coefficient tensor. Secondly, and perhaps most importantly we investigate the fundamental error threshold of the LSR-TGLM problem by deriving a minimax lower bound on the estimation error. This minimax bound is useful for assessing the performance of the proposed algorithm and ascertaining the parameters that may affect the sample complexity of the parameter estimation problem. The bound is general and can be specialised to previously introduced regression types under the GLM framework, such as CP and Tucker tensor GLMs. Obtaining the bound requires a special construction of a packing set of LSR-structured tensors. The methods we develop are systematic and can be appropriate in other works that consider similar topological properties of structured tensors (i.e., LSR-structured tensors). We also assess the tightness of our bounds in two ways: 1) Through a numerical study where we show that the ratio of the empirical error through LSRTR and the minimax bound is approximately constant with increasing sample size, and 2) We specialise our minimax bound to the Tucker linear regression case and show that our bound matches the optimal error rates for Tucker linear regression found in recent works (Zhang et al., 2020).

Finally, we evaluate the performance of our algorithm through extensive numerical experiments on synthetic data. We also test the performance of imposing the LSR structure on several classification problems with multidimensional medical imaging datasets. We show that while the LSR model outperforms the vector model, its rich representation power also allows for enhanced performance over the Tucker (and CP) case.

## 1.2 Relation To Prior Work

Regression problems have been a major focus of high-dimensional statistics for many years (Giraud, 2021). Some works on linear and logistic regression impose sparsity on the model parameter in order to reduce the sample complexity of the vector-based regression problems (Abramovich & Grinshtein, 2018; Sun & Zhang, 2012). However, in very high-dimensional regimes such as when the data is tensor-structured, i.e., we have $\{\underline{\mathbf{X}}_i\}_{i=1}^n$, sparsity assumptions do not provide enough reduction in the sample complexity (Raskutti et al., 2019; Lee & Courtade, 2020). Several works overcome the limitations of sparse vector regression by extending regular regression to the high-dimensional and low-rank matrix settings. Low-rank assumptions on data are common throughout the literature and are used to reduce the sample complexity of estimation problems (Barnes & Özgür, 2019; Shi et al., 2014). Such works propose regularized matrix linear and logistic regression models to obtain low-rank and/or sparse estimates of the coefficient matrix in regression problems, such as those on inference on images or graph data (Hung & Wang, 2013; Zhang & Jiang, 2018; Shi et al., 2014; An & Zhang, 2020; Berthet & Baldin, 2020). Some works directly impose low-rank structures on coefficient matrices through the rank-$r$ singular value decomposition (SVD) (Taki et al., 2021).

Additionally, though tensors and their decompositions have long since been introduced in the literature (Kolda & Bader, 2009), their applications in regression analysis have recently become established. Analogous to the low-rank matrix regression works, a variety of works have introduced low-rank structures on coefficient tensors for tensor regression problems. For logistic regression, Tan et al. (2013) first introduced using a low-rank and/or sparse CANDECOMP/PARAFAC (CP) decomposition. A more flexible generalization of this

work is imposing the Tucker decomposition on the coefficient tensor in tensor logistic regression (Zhang & Jiang, 2016; Wu et al., 2022). The Tucker structure has also been introduced for tensor linear regression (Zhang et al., 2020; Ahmed et al., 2020; Wu et al., 2022). To the best of our knowledge, the BTD structure has yet to be introduced for regression and GLMs.

More works to our interest generalize tensor linear and logistic regression works by imposing the CP and Tucker decompositions in tensor GLMs (Li et al., 2018; Zhou et al., 2013). Both structures have been shown to significantly reduce the number of learnable parameters, leading to efficient estimation and prediction in a variety of regression problems, particularly with medical imaging data. The aforementioned works develop efficient estimation algorithms and provide empirical results on their performance. The proposed approaches outperform vector-based methods (in terms of estimation and prediction accuracy) in the high-dimensional setting when the number of available samples is limited. However, these matrix and tensor structures are aimed at being compact (in the sense that they decrease the number of learnable parameters in a given problem), and are therefore also quite restrictive in their representation power of the true coefficient tensor. A more general and flexible tensor model is required to achieve accurate and efficient estimation while maintaining a useful level of compactness.

In terms of theoretical guarantees, various regression works provide local identifiability guarantees of the proposed CP and Tucker tensor models for GLMs, and asymptotic consistency and normality results for the MLE estimator of the model parameter (Li et al., 2018; Zhang et al., 2020; Zhou et al., 2013). Some works on high-dimensional regression also provide sample complexity bounds of the proposed model in the form of risk upper bounds (Zhang et al., 2020; Ahmed et al., 2020) or minimax lower bounds (Barnes & Özgür, 2019; Zhang et al., 2020; Raskutti et al., 2019; Foster et al., 2018; Abramovich & Grinshtein, 2018; Lee & Courtade, 2020; Abramovich & Grinshtein, 2016; Raskutti et al., 2011); however, these works are specific to vector-based logistic regression or Tucker linear regression. Current works in tensor logistic regression or tensor-based GLMs do not provide any theoretical guarantees for sample complexity (upper or lower bounds).

In terms of the LSR model, its motivational roots are two fold. First, a special case of the LSR model was introduced by Tsiligkaridis & Hero (2013) for covariance estimation problems. An extension of this model has only recently been used on tensor data for dictionary learning (Ghassemi et al., 2020). Secondly, the LSR model can be rearranged into a specialised form of the BTD model, equipped with further constraints. In terms of the GLM framework, to the best of our knowledge, our work is the first to consider the LSR (or BTD) model in regression problems.

### 1.3 Organization

The organization of this paper is as follows. In Section 2 we establish a background on various tensor models, as well as the LSR tensor model. In Section 3 we formulate the LSR-TGLM model and introduce two objectives: parameter estimation and minimax risk. In Section 4 we discuss the estimation problem of LSR-structured coefficient tensors in GLMs and propose an efficient algorithm for parameter estimation. In Section 5 we provide a numerical study of the LSRTR algorithm with synthetic data and experiments on real data. In Section 6 we introduce a sample complexity bound in the form of a minimax lower bound on the estimation error of the low-rank LSR-GLM model and provide a formal proof in Section 6.3. We conclude our work in Section 7. Proofs of lemmas for the main theorem, and additional numerical results are provided in the appendix.

## 2 Preliminaries

This work is based on structured tensor decompositions. For a more comprehensive tutorial on tensor decompositions, see the survey of Kolda & Bader (2009). We will now list some necessary preliminaries regarding tensors and tensor structures.

We use the following notation convention throughout the paper: $x$, $\mathbf{x}$, $\mathbf{X}$ and $\underline{\mathbf{X}}$ denote scalars, vectors, matrices and tensors, respectively. Specifically, $\mathcal{X}$ is the tensor defined as the aggregation of $n$ tensors: $\mathcal{X} = [\underline{\mathbf{X}}_1, \underline{\mathbf{X}}_2, \ldots, \underline{\mathbf{X}}_n]$. Given a fixed tensor $\underline{\mathbf{X}}$, $\mathbf{x} \triangleq \mathrm{vec}(\underline{\mathbf{X}})$ is the column-wise vectorization of $\underline{\mathbf{X}}$. The

tensor $\underline{\mathbf{I}}_m$ is the $m \times \cdots \times m$ identity tensor, such that for any tensor $\underline{\mathbf{S}}$, the product $\underline{\mathbf{I}} \cdot \underline{\mathbf{S}} = \underline{\mathbf{S}} \cdot \underline{\mathbf{I}} = \underline{\mathbf{S}}$. Given $n$ $K$-mode tensors $\{\underline{\mathbf{X}}_i\}_{i=1}^n$ of dimension $m_1 \times m_2 \times \cdots \times m_K$, $\mathcal{X}$ is the combined (aggregated) tensor of $n$ samples of dimension $m_1 \times m_2 \times \cdots \times m_K \times n$. For a positive integer $K$, the set $[K] = \{1, 2, \ldots, K\}$ so that $(\mathbf{X}_k)_{k \in [K]}$ is the ordered set $(\mathbf{X}_1, \mathbf{X}_2, \ldots, \mathbf{X}_K)$. The symbol $-[K]$ for the reverse order, so that $(\mathbf{X}_k)_{k \in -[K]} = (\mathbf{X}_K, \mathbf{X}_{K-1}, \ldots, \mathbf{X}_1)$. For a matrix $\mathbf{X}$, the vector $\mathbf{x}^{(j)}$ is the $j^{th}$ column of $\mathbf{X}$ and the vector $\mathbf{x}^{T(j)}$ is its $j^{th}$ row. For a vector $\mathbf{x}$, $\mathbf{x}(j)$ is the $j^{th}$ element of $\mathbf{x}$. If $x \in \mathbb{R}$ then $\lfloor x \rfloor$ is the greatest integer less than or equal to $x$. We use the standard notation $\|\mathbf{x}\|_p$ for the $p$-norm ($p \geq 1$) of a vector $\mathbf{x}$. For a matrix $\mathbf{X}$ or tensor $\underline{\mathbf{X}}$, the Frobenius norms are $\|\mathbf{X}\|_F$ and $\|\underline{\mathbf{X}}\|_F$, respectively. For two vectors $\mathbf{x}_1$ and $\mathbf{x}_2$, $\mathbf{x}_1 \circ \mathbf{x}_2$ denotes their outer product. Similarly, for two matrices $\mathbf{X}_1$ and $\mathbf{X}_2$, $\mathbf{X}_1 \otimes \mathbf{X}_2$ denotes their Kronecker product. The inner product between two vectors, matrices or tensors is denoted as $\langle \cdot, \cdot \rangle$. For a set of $K$ vectors $\{\mathbf{x}_i\}_{i=1}^K$, $\mathbf{x}_1 \circ \mathbf{x}_2 \circ \cdots \circ \mathbf{x}_K$ produces a $k$-dimensional rank-1 tensor. For $K$ matrices $\{\mathbf{X}_i\}_{i=1}^K$, $\bigotimes_{k \in [K]} \mathbf{X}_k \triangleq \mathbf{X}_1 \otimes \mathbf{X}_2 \otimes \cdots \otimes \mathbf{X}_K$ produces a '$K$-order Kronecker-structured matrix'. We call a matrix a $K$-order Kronecker-structured matrix if it is a product of $K \geq 2$ matrices. The mode-$k$ matricization of tensor $\underline{\mathbf{X}}$ is $\underline{\mathbf{X}}_{(k)}$ (Kolda & Bader, 2009), and given a matrix $\mathbf{B}$, $\underline{\mathbf{X}} \times_k \mathbf{B}$ denotes the multiplication of $\underline{\mathbf{X}}$ by $\mathbf{B}$ along mode $k$. Finally, for all $k \in [K]$ we have $\underline{\mathbf{X}} \times_{[K]} \mathbf{B}_k \triangleq \underline{\mathbf{X}} \times_1 \mathbf{B}_1 \times_2 \cdots \times_K \mathbf{B}_K$.

We now formally define GLMs for vector-structured covariates, as discussed in the literature.

**Definition 1** (Vector-structured Generalized Linear Models). *Consider an observation $y$, a vector of covariates $\mathbf{x} \in \mathbb{R}^m$, and a bias $z$ and regression coefficient vector $\mathbf{b}$, both to be estimated. Let $y$ be a response variable generated from a distribution in the exponential family with probability mass/density function as follows:*

$$\mathbb{P}(y, \eta) = b(y) \exp(\eta T(y) - a(\eta)). \tag{1}$$

*Here, $\eta$ is called the natural parameter, $T(y)$ is the sufficient statistic and $a(\eta)$ is the log-partition (or cumulant) function. Consider a given regression problem of estimating $y$ given $\mathbf{x}$. This requires minimising the MSE as follows:*

$$\widehat{y}_{\mathrm{MMSE}} = \arg\min_{\widehat{y}} \mathbb{E}_{\mathbf{x}, y}[(\widehat{y} - y)^T (\widehat{y} - y)] \tag{2}$$

*The Minimum MSE (MMSE) solution is the expected value of $y$ conditioned on $\mathbf{x}$, or $\mathbb{E}[y|\mathbf{x}] = \mu$. Now, let $H$ be the support of $y$ and consider a strictly increasing and invertible link function $g(\cdot) : H \to \mathbb{R}$. The Generalized Linear Model is then defined as*

$$g(\mu) = \eta \triangleq \langle \mathbf{b}, \mathbf{x} \rangle + z, \tag{3}$$

*where the natural parameter $\eta$ is the linear predictor, $\mu = g^{-1}(\eta)$, and $z$ is the bias.*

The core idea behind GLMs is that though many regression problems are not linear, the distribution of the observation $y$ is only affected by the linear combination $\langle \mathbf{b}, \mathbf{x} \rangle + z$. If the distribution of $y$ falls under the exponential family (which it often does), then $y$ is related to the linear predictor via $g(\cdot)$. In the case of linear regression where $y \in \mathbb{R}$ is assumed to be a continuous response variable with Gaussian distribution, we have $\mu = \langle \mathbf{b}, \mathbf{x} \rangle + z$ and $g(\cdot)$ is just the identity function. In logistic regression $y$ is Bernoulli distributed, taking values $y \in \{0, 1\}$, and we have $\mu = \frac{1}{1+\exp(-\langle \mathbf{b}, \mathbf{x} \rangle + z)}$ and $g(\mu) = \log\left(\frac{\mu}{1-\mu}\right)$. In Poisson regression $y \in \mathbb{N}$ is a Poisson distributed random variable that expresses the count of an event occurring in a fixed time interval, while $\mu = \exp(\langle \mathbf{b}, \mathbf{x} \rangle + z)$ and $g(\mu) = \log(\mu)$.

**Definition 2** (CP Decomposition (Kolda & Bader, 2009; Zhou et al., 2013)). *Consider a $K$-mode tensor $\underline{\mathbf{B}} \in \mathbb{R}^{m_1 \times \cdots \times m_K}$. The rank-$r$ CANDECOMP/PARAFAC (CP) decomposition decomposes $\underline{\mathbf{B}}$ into a sum of $r$ rank-1 tensors as follows:*

$$\underline{\mathbf{B}} = \sum_{i \in [r]} \mathbf{b}_{1,i} \circ \cdots \circ \mathbf{b}_{K,i}, \tag{4}$$

where $\mathbf{b}_{k,i} \in \mathbb{R}^{m_k}$, $k \in [K]$, $i \in [r]$ is a column vector. Equivalently, (4) can be expressed in vector form as follows:

$$\text{vec}(\underline{\mathbf{B}}) \triangleq \mathbf{b} = \sum_{i \in [r]} \mathbf{b}_{K,i} \otimes \cdots \otimes \mathbf{b}_{1,i}. \tag{5}$$

**Definition 3** (Tucker decomposition (Kolda & Bader, 2009; Li et al., 2018)). *Consider a $K$-mode tensor $\underline{\mathbf{B}} \in \mathbb{R}^{m_1 \times \cdots \times m_K}$. The rank-$(r_1, \ldots, r_k)$ Tucker decomposition decomposes $\underline{\mathbf{B}}$ as follows:*

$$\underline{\mathbf{B}} = \underline{\mathbf{G}} \times_1 \mathbf{B}_1 \times_2 \cdots \times_K \mathbf{B}_K, \tag{6}$$

*where $\underline{\mathbf{G}} \in \mathbb{R}^{r_1 \times \cdots \times r_K}$ denotes the core tensor and $\{\mathbf{B}_k \in \mathbb{R}^{m_k \times r_k}\}_{k \in [K]}$ denote the factor matrices. Equivalently, by defining $\mathbf{g} \triangleq \text{vec}(\underline{\mathbf{G}})$, (6) can be expressed in vector form as follows:*

$$\text{vec}(\underline{\mathbf{B}}) \triangleq \mathbf{b} = \big(\mathbf{B}_K \otimes \cdots \otimes \mathbf{B}_1\big)\mathbf{g}. \tag{7}$$

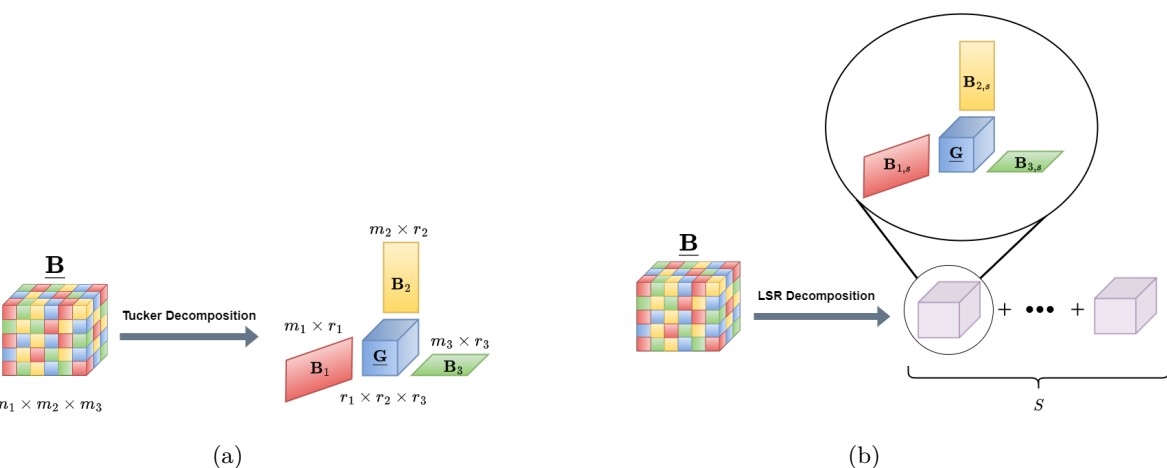

(a)               (b)

Figure 1: (*a*): A third-order tensor under the Tucker model. Tensor $\underline{\mathbf{B}}$ is decomposed into a core tensor $\underline{\mathbf{G}}$ and factor matrices $\mathbf{B}_k$ multiplied along the $k^{th}$ mode of $\underline{\mathbf{G}}$ for $k \in [3]$. The CP model appears as a special case of the Tucker model where $\underline{\mathbf{G}}$ is a diagonal tensor of equal dimension along each mode. (*b*): A third-order tensor under the LSR decomposition. Tensor $\underline{\mathbf{B}}$ is comprised of a sum of Tucker-structured tensors, with core tensor $\underline{\mathbf{G}}$ fixed across all summands.

A visual depiction of the Tucker model is in Figure 1a. The Tucker model decomposes a tensor into a core tensor $\underline{\mathbf{G}}$ of dimension $r_1 \times \cdots \times r_K$ which is then multiplied by a factor matrix $\mathbf{B}_k$ along each mode $k \in [K]$. Assuming $\underline{\mathbf{G}}$ is small (i.e., $r_k \ll m_k \ \forall k \in [K]$), the factor matrices $\mathbf{B}_k \ \forall k \in [K]$ are tall, rank $r_k$ matrices. Additionally, the CP model appears as a specialised case of the Tucker model as follows: Fix the number of basis vectors along all modes to some $r \in \mathbb{R}$ (i.e., fix the rank of all factor matrices to $r$) and impose $\underline{\mathbf{G}} \in \mathbb{R}^{\overbrace{r \times \cdots \times r}^{k \text{ times}}}$ as a diagonal tensor. The CP model lends a desirable compactness as the number of learnable parameters in a CP-structured tensor can be far fewer than that of an unstructured tensor. Despite this, however, the restrictive nature of the CP model (specifically its rank restriction, where each factor matrix must have equal rank) renders it unfavourable against the more 'rank-flexible' Tucker model.

Our second observation refers to (7). The vectorization of a Tucker-structured tensor shows the Kronecker-structured matrix composed of the $K$ factor matrices. However this 'Kronecker' structure – which implies that the coefficient vector $\mathbf{b}$ is composed of separable sub-matrices weighted by some vector $\mathbf{g} = \text{vec}(\underline{\mathbf{G}})$ – is also quite restrictive, when considering different tensor structures having the same rank. To overcome this, the Block Tensor Decomposition (BTD) was introduced and studied in recent works (De Lathauwer, 2008; Rontogiannis et al., 2021; Fu et al., 2020). An alternative way of viewing the BTD structure is as a summation of $S$ Tucker-structured tensors. The LSR decomposition we define next is a special case of a BTD that uses the concept of matrix separation rank.

**Definition 4** (Matrix Separation Rank (Tsiligkaridis & Hero, 2013)). *Fix $\mathbf{m} = (m_1, m_2, \ldots, m_K) \in \mathbb{N}^K$ and $\mathbf{r} = (r_1, r_2, \ldots, r_K) \in \mathbb{N}^K$, and set $\widetilde{m} = \prod_{k \in [K]} m_k$ and $\widetilde{r} = \prod_{k \in [K]} r_k$. Then for a matrix $\mathbf{B} \in \mathbb{R}^{\widetilde{m} \times \widetilde{r}}$, its separation rank $\mathfrak{S}_{m,p}^K(\cdot)$ is the minimum number $S$ of $K$-order Kronecker-structured matrices such that*

$$\mathbf{B} = \sum_{s \in [S]} \mathbf{B}_{(1,s)} \otimes \cdots \otimes \mathbf{B}_{(K,s)}, \tag{8}$$

*where $\mathbf{B}_{(k,s)} \in \mathbb{R}^{m_k \times r_k}$.*

The LSR model in (8) generalizes (7) by replacing the single Kronecker-structured matrix in (7) with a sum of the form in (8). It poses that a matrix can be expressed as a sum of $S$ Kronecker-structured matrices, with $S = 1$ being a specialised case and what we observe in the Tucker model when the tensor is vectorized, as shown in (7). Having introduced the concept of separation rank, we are now ready to define Low Separation Rank (LSR) tensors.

**Definition 5** (Low Separation Rank (LSR) Tensor Decomposition). *Consider a $K$-mode tensor $\underline{\mathbf{B}} \in \mathbb{R}^{m_1 \times \cdots \times m_K}$. The rank-$(r_1, \ldots, r_k)$ LSR decomposition with separation rank $S$ decomposes $\underline{\mathbf{B}}$ as follows:*

$$\underline{\mathbf{B}} = \sum_{s \in [S]} \underline{\mathbf{G}} \times_1 \mathbf{B}_{(1,s)} \times_2 \cdots \times_K \mathbf{B}_{(K,s)}, \tag{9}$$

*where $\underline{\mathbf{G}} \in \mathbb{R}^{r_1 \times \cdots \times r_K}$ denotes the core tensor and $\mathbf{B}_{(k,s)} \in \mathbb{R}^{m_k \times r_k}$, $k \in [K]$, $s \in [S]$ denote the Kronecker-structured factor matrices. Equivalently, by defining $\mathbf{g} \triangleq \mathrm{vec}(\underline{\mathbf{G}})$, (9) can be expressed in vector form as follows:*

$$\mathrm{vec}(\underline{\mathbf{B}}) \triangleq \mathbf{b} = \sum_{s \in [S]} \left( \mathbf{B}_{(K,s)} \otimes \cdots \otimes \mathbf{B}_{(1,s)} \right) \mathbf{g}. \tag{10}$$

A visual depiction of the LSR model is in Figure 1b. For the purposes of this work, we pose some constraints on the LSR decomposition. First, we assume that the $KS$ factor matrices $\mathbf{B}_{(k,s)}$ are 'tall' and rank $r_k$ (i.e., $r_k \ll m_k \ \forall k \in [K]$) and have orthonormal columns. Secondly, the LSR model corresponds to tensor $\underline{\mathbf{B}}$ having a separation rank that is relatively small so that $1 \leq S < \min \left( \prod_{k \in [K]} m_k, \prod_{k \in [K]} r_k \right)$. Thus, defining $\mathbb{O}^{m \times r}$ as the $m \times r$ Stiefel manifold, and for a fixed tensor rank $(r_1, r_2, \ldots, r_K)$ and separation rank $S$, the LSR structured tensor $\underline{\mathbf{B}}$ belongs to the following parameter space $\mathcal{P}_{\{r_k\},S}$:

$$\mathcal{P}_{\{r_k\},S} \triangleq \left\{ \underline{\mathbf{B}}' = \sum_{s \in [S]} \underline{\mathbf{G}}' \times_1 \mathbf{B}'_{(1,s)} \times_2 \cdots \times_K \mathbf{B}'_{(K,s)} \in \mathbb{R}^{m_1 \times \cdots \times m_K} : \underline{\mathbf{G}}' \in \mathbb{R}^{r_1 \times \cdots \times r_K}, \right.$$

$$\left. \mathrm{rank}(\underline{\mathbf{B}}') = (r_1, \ldots, r_K), \ \mathbf{B}_{(k,s)} \in \mathbb{O}^{m_k \times r_k}, \ k \in [K], \ s \in [S] \right\} \tag{11}$$

The orthonormality assumption on the columns of $\mathbf{B}_{(k,s)}$ is a common assumption made in the GLM literature (Zhang & Jiang, 2016). Similar assumptions such as unit-norm columns and columns with fixed entries are also common (Zhou et al., 2013; Li et al., 2018).

We have described how the LSR structure in (9) is a generalization of the Tucker (and therefore of CP) decomposition. The form in (9) is also a special case of BTD, or a summation of $S$ Tucker-structured tensors, equipped with orthogonality constraints and a common core tensor among all summands. The BTD can also be rearranged into a specialised Tucker-structured tensor with block-diagonal core tensor of dimensions $Kr_1 \times \cdots \times Kr_K$ and factor matrices of size $m_k \times Sr_k$. In our work, however, we compare the number of learnable parameters between tensor decompositions (CP, Tucker and LSR) of same-sized core tensor (of same rank). When comparing tensor decompositions for a fixed rank $(r_1, \ldots, r_K)$, LSR generalizes the Tucker decomposition. We also remark that while the differences between LSR and a general BTD might appear nuanced, they are significant, particularly the critical requirement for a common core tensor in the LSR structure, which allows for summing the Kronecker-structured factor matrices. This distinction

is fundamental for retaining the LSR matrix structure, thereby offering a number of benefits, on which we will elaborate in Sections 3, 4, and 6.

Moving forward, Table 1 reviews the number of learnable parameters for the three models of CP, Tucker and LSR, for a fixed rank $(r_1, \ldots, r_K)$. The CP model contains the least number of parameters, especially if $r$ is small. The LSR model is the most complex of the three models in that it has more parameters than the Tucker model. The working hypothesis throughout this work is that the price we are likely to pay for an increase in sample complexity (by using the LSR model for tensor GLM problems) is worth the gain we are likely to achieve in representation power and estimation accuracy.

|  | CP | Tucker | LSR |
|---|---|---|---|
| **Parameters** | $\sum_{k=1}^{K}(m_k r) + r$ | $\sum_{k=1}^{K}(m_k r_k) + \prod_{k=1}^{K} r_k$ | $S \sum_{k=1}^{K}(m_k r_k) + \prod_{k=1}^{K} r_k$ |

Table 1: Number of learnable parameters in the three tensor models (CP, Tucker and LSR).

## 3 Problem Statement

We are now ready to propose the Low Separation Rank Tensor Generalized Linear Model (LSR-TGLM). Consider a response variable $y$ with probability distribution belonging to the exponential family with parameter $\eta$, as in (1). Consider also tensor-structured covariates $\underline{\mathbf{X}} \in \mathbb{R}^{m_1 \times \cdots \times m_K}$ and a coefficient tensor $\underline{\mathbf{B}} \in \mathbb{R}^{m_1 \times \cdots \times m_K}$ that assumes a low-rank LSR structure as shown in (9). Given a link function $g(\cdot)$, the LSR-TGLM model assumes $\eta$ is given by

$$g(\mu) = \eta \triangleq \left\langle \sum_{s \in [S]} \underline{\mathbf{G}} \times_1 \mathbf{B}_{(1,s)} \times_2 \cdots \times_K \mathbf{B}_{(K,s)}, \underline{\mathbf{X}} \right\rangle + z. \tag{12}$$

It is important to note here that we assume that the tensor rank $(r_1, r_2, \ldots, r_K)$ and LSR rank $S$ are known for reasons we will discuss in Section 3.1. Additionally for algebraic simplicity, from this point forward we will consider the standard case where $z = 0$ in (3) and without loss of generality we will express the LSR-TGLM model as

$$g(\mu) = \left\langle \sum_{s \in [S]} \underline{\mathbf{G}} \times_1 \mathbf{B}_{(1,s)} \times_2 \cdots \times_K \mathbf{B}_{(K,s)}, \underline{\mathbf{X}} \right\rangle. \tag{13}$$

The LSR-TGLM model in (13) can be written as a standard GLM by vectorizing the parameters as follows

$$g(\mu) = \langle \text{vec}(\underline{\mathbf{B}}), \text{vec}(\underline{\mathbf{X}}) \rangle = \left\langle \sum_{s \in [S]} \left( \mathbf{B}_{(K,s)} \otimes \cdots \otimes \mathbf{B}_{(1,s)} \right) \mathbf{g}, \mathbf{x} \right\rangle. \tag{14}$$

The sum of Kronecker-structured matrices in (14) is due to the LSR-TGLM model in (13). We can now formally define the two goals that are at the core of this work: 1) Parameter estimation for LSR-TGLM and 2) Minimax lower bound on the estimation error. One of the benefits of the requirement of a common core tensor is an intuitive reasoning pertaining to the role of the core tensor in the LSR-structured tensor. If one was to view $\underline{\mathbf{G}}$ as a 'weight tensor', the same $\underline{\mathbf{G}}$ implies that the $S$ groups of factor matrices must be given the same weight. We also emphasise here that the conceptual novelty of our work resides not in introducing the concept of LSR-structured tensors, but in our unique application of the LSR decomposition to the GLM model and our comprehensive analysis of the resulting GLM, which is non-trivial.

### 3.1 Parameter Estimation for LSR-TGLM

The objective in regression theory is to predict an outcome $y$ based on $\underline{\mathbf{X}}$, which is achieved by first estimating the model parameter. For LSR-TGLMs, we wish to find an estimate of $\underline{\mathbf{B}}$ that best fits the model in (13).

The underlying (true) $\underline{\mathbf{B}}$ is a low-rank LSR-structured tensor belonging to a constraint set $\mathcal{C}$:

$$\mathcal{C} \triangleq \left\{ \underline{\mathbf{B}} \in \mathbb{R}^{m_1 \times \cdots \times m_K} : \underline{\mathbf{B}} \in \mathcal{P}_{\{r_k\}, S}, r_k \leq R_K, S \leq \min \left\{ \prod_k m_k, \prod_k r_k \right\}, k \in [K] \right\}. \tag{15}$$

This is the set of all LSR-structured tensors with constrained tensor rank tuple $(r_1, r_2, \ldots, r_K)$ and constrained separation rank $S$. Now consider the space of tensor-structured covariates $\mathbb{X} \subset \mathbb{R}^{m_1 \times m_2 \times \cdots \times m_K}$, and space of scalar observations $\mathbb{Y} \subset \mathbb{R}$. There exists a probability measure, denoted as $\mathbb{P}_{\mathbf{x}y}$, that allows a learning procedure to randomly draw points $\{\underline{\mathbf{X}} \in \mathbb{X}, y \in \mathbb{Y}\}$ from the product space $\mathbb{X} \times \mathbb{Y}$. We do not know $\mathbb{P}_{\mathbf{x}y}$, but we have access to $n$ independently sampled observations $\{\underline{\mathbf{X}}_i, y_i\}_{i=1}^n$. Therefore, we find an estimate of $\underline{\mathbf{B}}$ through Maximum Likelihood Estimation (MLE), making the the objective function the negative log-likelihood:

$$\mathcal{L}_n(\underline{\mathbf{B}}) = \sum_{i=1}^n \log(b(y_i)) + \sum_{i=1}^n \left( \langle \underline{\mathbf{B}}, \underline{\mathbf{X}}_i \rangle T(y_i) - a(\langle \underline{\mathbf{B}}, \underline{\mathbf{X}}_i \rangle) \right). \tag{16}$$

To find the MLE, we minimise (16) over the constraint set $\mathcal{C}$:

$$\underset{\underline{\mathbf{B}} \in \mathcal{C}}{\arg\min} \sum_{i=1}^n \log(b(y_i)) + \sum_{i=1}^n \left( \langle \underline{\mathbf{B}}, \underline{\mathbf{X}}_i \rangle T(y_i) - a(\langle \underline{\mathbf{B}}, \underline{\mathbf{X}}_i \rangle) \right). \tag{17}$$

The vectorized LSR-structured tensor uses a sum of $S$ Kronecker-structured matrices. Lemma 1 from Ghassemi et al. (2020) shows that there is a 1-to-1 mapping between Kronecker-structured matrices with separation rank $S$ and a set of rank $S$ tensors. Finding the rank $r$ of a tensor is NP-hard, (Håstad, 1990), and thus so is finding the separation rank of a Kronecker-structured matrix. Therefore, in the context of our work on LSR-TGLM, finding the LSR rank $S$ is NP-hard, and the problem in (17) is therefore intractable. To mitigate this issue we first assume that the tensor rank $(r_1, r_2, \ldots, r_K)$ and separation rank $S$ are known and we solve the following factorised problem for parameter estimation for LSR-TGLMs:

$$\underset{\{\mathbf{B}_{(k,s)}\}, \underline{\mathbf{G}}}{\arg\min} = \sum_{i=1}^n \log(b(y_i)) + \sum_{i=1}^n \left\langle \sum_{s=1}^S \underline{\mathbf{G}} \times_{[K]} \mathbf{B}_k, \underline{\mathbf{X}}_i \right\rangle T(y_i) - a\left( \left\langle \sum_{s=1}^S \underline{\mathbf{G}} \times_{[K]} \mathbf{B}_k, \underline{\mathbf{X}}_i \right\rangle \right). \tag{18}$$
$$\text{subject to } \mathbf{B}_{(k,s)} \in \mathbb{O}^{m_k \times r_k}.$$

The expression in (18) provides a tractable relaxation for (17), where the coefficient tensor is explicitly written in terms of the core tensor $\underline{\mathbf{G}}$ and factor matrices $\mathbf{B}_{(k,s)}$ of the low-rank LSR structure. In this work, we will study the problem in (18).

## 3.2   Minimax Lower Bound for LSR-TGLM

Our second goal is to derive a lower bound on the minimax risk of estimating LSR-structured coefficient tensors for the LSR-TGLM problem in (13). Minimax bounds can be useful tools in developing an insight into the parameters on which an achievable error of a given problem might depend and shed light on the benefits of imposing tensor structures in regression problems. They also provide a means of quantifying the performance of existing algorithms. Previous studies of tensor-structured GLMs (Zhou et al., 2013; Li et al., 2018) fall short of providing a sample complexity analysis. The analysis in this work is thus instrumental in revealing the potential benefits of imposing structure on the coefficient tensor within the GLM context. We will show that the LSR model exhibits a lower sample complexity as compared to the vector case, and provides an expressive representation of tensor data.

We adopt a local analysis and assume that the LSR-TGLM's underlying (true) $\underline{\mathbf{B}}$ resides within a neighbourhood with known radius around a fixed point. However, for a sufficiently large neighborhood, the minimax lower bounds derived in this work effectively become independent of the radius. Therefore we assume that for a fixed tensor rank $(r_1, r_2, \ldots, r_K)$ and separation rank $S$, the underlying $\underline{\mathbf{B}}$ belongs to the set

$$\mathcal{B}_d(\underline{\mathbf{0}}) \triangleq \{\underline{\mathbf{B}}' \in \mathcal{P}_{\{r_k\}, S} : \rho(\underline{\mathbf{B}}', \underline{\mathbf{0}}) < d\}, \tag{19}$$

the ball of radius $d$ with distance metric $\rho = \|\cdot\|_F$, which resides in the parameter space $\mathcal{P}_{\{r_k\},S}$ defined in (11). Thus $\mathcal{B}_d(\mathbf{0}) \subset \mathcal{P}_{\{r_k\},S}$ and $\underline{\mathbf{B}}$ has energy bounded by $\|\underline{\mathbf{B}}\|_F^2 < d^2$. Note that we fix the reference point as the tensor of all zero-elements $\underline{\mathbf{0}}$ without loss of generality. Indeed, any neighbourhood $\mathcal{B}_d(\underline{\mathbf{A}})$ around a point $\underline{\mathbf{A}} \in \mathcal{P}_{\{r_k\},S}$ is just a translation from $\mathcal{B}_d(\underline{\mathbf{0}})$ with known distance $\|\underline{\mathbf{A}}\|_F$. Additionally, we note that the point $\underline{\mathbf{0}}$ also belongs to the parameter space $\mathcal{P}_{\{r_k\},S}$. The minimax risk is defined as the minimum worst-case behaviour for any estimator. Mathematically, it is expressed as follows.

$$\varepsilon^* = \inf_{\widehat{\underline{\mathbf{B}}}} \sup_{\underline{\mathbf{B}} \in \mathcal{B}_d(\underline{\mathbf{0}})} \mathbb{E}_{\mathbf{y},\mathcal{X}} \left\{ \phi(\widehat{\underline{\mathbf{B}}}, \underline{\mathbf{B}}) \right\}. \tag{20}$$

Here, $\widehat{\underline{\mathbf{B}}}$ denotes an estimator of $\underline{\mathbf{B}}$, $\mathbf{y} = [y_1, y_2, \ldots, y_n]$, $\mathcal{X} = [\underline{\mathbf{X}}_1, \underline{\mathbf{X}}_2, \ldots, \underline{\mathbf{X}}_n]$, and $\phi$ is a function with $\phi(0) = 0$. If we define $\phi = \|\cdot\|_F^2 \triangleq \mathbb{R}_+ \to \mathbb{R}_+$ with $\phi(0) = 0$ then the minimax risk is simply the worst-case Mean Squared Error (MSE) for the best estimator, i.e.,

$$\varepsilon^* = \inf_{\widehat{\underline{\mathbf{B}}}} \sup_{\underline{\mathbf{B}} \in \mathcal{B}_d(\underline{\mathbf{0}})} \mathbb{E}_{\mathbf{y},\mathcal{X}} \left\{ \left\| \widehat{\underline{\mathbf{B}}} - \underline{\mathbf{B}} \right\|_F^2 \right\}. \tag{21}$$

Proving a lower bound $\varepsilon^* > \varepsilon_0$ on the minimax risk shows that any estimator must have a risk lower bounded by $\varepsilon_0$. Existing minimax bounds on the parameter estimation problem in GLMs or regression models provided in the literature (Abramovich & Grinshtein, 2016; Lee & Courtade, 2020; Raskutti et al., 2011) cannot be applied here for two reasons. Primarily, these bounds do not account for the impact of the structural assumptions we make on our model. In fact, we require bounds that accurately reflect the sample complexity of estimation algorithms for LSR-structured coefficient tensors. Secondly, the link function in GLMs also involves the analysis of the space of LSR-structured coefficients, making this part of our analysis non-trivial and fundamentally different to such works. We elaborate this point further in Section 6.3. The derived minimax bound in this section conveniently generalizes the CP and Tucker tensor structures and can be specialised to existing bounds in the literature such as that for Tucker-structured linear regression (Zhang et al., 2020).

We now introduce some standard lemmas and assumptions used in this work.

**Assumption 1** (Covariate Distribution). *For $\underline{\mathbf{X}} \in \mathbb{R}^{m_1 \times \cdots \times m_K}$, define $\mathrm{vec}(\underline{\mathbf{X}}) \triangleq \mathbf{x}$ and $\widetilde{m} = \prod_{k \in [K]} m_k$.*

*Then, $\mathbf{x} \sim \mathcal{N}(\mathbf{0}, \boldsymbol{\Sigma}_x)$, and thus $\mathbb{E}[\mathbf{x}] = \mathbf{0}$ and $\mathbb{E}[\mathbf{x}\mathbf{x}^T] = \boldsymbol{\Sigma}_x$.*

**Lemma 1** ((McCullagh & Nelder, 2019)). *Any observation $y$ generated according to a distribution from the exponential family has mean $a'(\eta)$, i.e., the first derivative of $a(\eta)$, and variance $a''(\eta)$, i.e., the second derivative of $a(\eta)$.*

**Assumption 2.** *The first derivative of the cumulant function, $a(\eta)$, with respect to $\eta$ is bounded uniformly by a constant $M \geq 0$: $a'(\eta) \leq M$.*

**Lemma 2.** *Consider the standard GLM problem in Definition 1 with negative log-likelihood function*

$$\sum_{i=1}^{n} \log(b(y_i)) + \sum_{i=1}^{n} \left( \langle \mathbf{b}, \mathbf{x}_i \rangle^T T(y_i) - a(\langle \mathbf{b}, \mathbf{x}_i \rangle) \right). \tag{22}$$

*The gradient of (22) with respect to $\mathbf{b}$ is $\sum_{i=1}^{n} \left( T(y_i) - g^{-1}(\langle \mathbf{b}, \mathbf{x}_i \rangle) \right) \mathbf{x}_i$.*

The proof of Lemma 2 follows the same steps explained in McCullagh & Nelder (2019). Assumption 1 states that $\mathrm{vec}(\underline{\mathbf{X}}) \triangleq \mathbf{x}$ is a zero-mean Gaussian random variable with covariance matrix $\boldsymbol{\Sigma}_x$. We do not place any further assumptions on $\boldsymbol{\Sigma}_x$. Assumption 2 implies that the mean of any GLM observation $y$ is bounded. This is a common assumption made in the literature (Lee & Courtade, 2020). For binary logistic regression, e.g., this assumption is satisfied with $M = 1$. For linear or Poisson regression, this assumption implies that the energy of $y$ is bounded. Though the range of $y$ is $\mathbb{R}$ and $\mathbb{N}$ for linear and Poisson regression, respectively, any outcome can be bounded by a positive constant $M$ with high probability in most instances.

# 4 Estimation Problem and Algorithm

To solve (18) we propose an approach similar to those found in prior works (Li et al., 2018; Zhou et al., 2013; Zhang & Jiang, 2016; Tan et al., 2013). The main idea behind this approach is to recognise that although the objective function in (18) is non-convex in all elements $\underline{\mathbf{G}}$ and $\{\mathbf{B}_{(k,s)}\}_{k\in[K],s\in[S]}$ jointly, it is convex with respect to each element separately. In this work, we propose a Block Coordinate Descent (BCD) algorithm that estimates each element of the LSR structured tensor $\underline{\mathbf{B}}$ alone while holding all other elements constant. We note for the reader that BCD is simply Alternating Minimisation (AM) when there are only two factors to be estimated. Additionally, one may use any solver to solve each convex sub-problem to estimate each element; however, we choose to solve it via gradient steps in this work. First, for every $k' \in [K]$ and $s' \in [S]$, we estimate the factor matrix $\mathbf{B}_{k',s'}$ while keeping all other factor matrices $\{\mathbf{B}_{(k,s)}\}_{k\in[K]\backslash k',s\in[S]\backslash s'}$ and core tensor $\underline{\mathbf{G}}$ fixed. Secondly, we estimate the core tensor $\underline{\mathbf{G}}$ while keeping all factor matrices $\{\mathbf{B}_{(k,s)}\}_{k\in[K],s\in[S]}$ fixed. We also point out that though our BCD approach is similar to some existing works, in this work we also show how each sub-problem reduces to a smaller scale GLM problem with fewer learnable parameters. We also evaluate the computational complexity of each sub-problem. We derive the optimisation problem for each step of the algorithm below.

## 4.1 Estimating Factor Matrices

To estimate the factor matrices we perform gradient descent over our objective function and project each iterate onto the Stiefel manifold, with the objective function corresponding to the block $\mathbf{B}_{(k',s')}$ given by:

$$
\mathcal{L}_n\left(\mathbf{B}_{(k',s')}\right) \tag{23}
$$
$$
= \sum_{i=1}^{n} \left( \left\langle \mathbf{B}_{(k',s')}, \mathbf{X}_{i_{(k')}} \left( \bigotimes_{-[K]\backslash k'} \mathbf{B}_{(k,s')} \right) \mathbf{G}_{(k')}^T \right\rangle + \sum_{s\in[S]\backslash s'} \left\langle \underline{\mathbf{G}} \times_{[K]} \mathbf{B}_{(k,s)}, \underline{\mathbf{X}}_i \right\rangle \right) T(y_i)
$$
$$
- a \left( \left\langle \mathbf{B}_{(k',s')}, \mathbf{X}_{i_{(k')}} \left( \bigotimes_{-[K]\backslash k'} \mathbf{B}_{(k,s')} \right) \mathbf{G}_{(k')}^T \right\rangle + \sum_{s\in[S]\backslash s'} \left\langle \underline{\mathbf{G}} \times_{[K]} \mathbf{B}_{(k,s)}, \underline{\mathbf{X}}_i \right\rangle \right).
$$

In order to derive the expression in (23) for every $\mathbf{B}_{(k',s')}$ for $k' \in [K]$ and $s' \in [S]$, notice that:

$$
\langle \underline{\mathbf{B}}, \underline{\mathbf{X}} \rangle = \left\langle \underline{\mathbf{G}} \times_1 \mathbf{B}_{(1,s')} \times_2 \cdots \times_K \mathbf{B}_{(K,s')}, \underline{\mathbf{X}} \right\rangle + \sum_{s\in[S]\backslash s'} \left\langle \underline{\mathbf{G}} \times_1 \mathbf{B}_{(1,s)} \times_2 \cdots \times_K \mathbf{B}_{(K,s)}, \underline{\mathbf{X}} \right\rangle
$$
$$
= \left\langle \mathbf{B}_{(k',s')}, \mathbf{X}_{(k')} \left( \mathbf{B}_{(K,s')} \otimes \cdots \otimes \mathbf{B}_{(k'+1,s')} \otimes \mathbf{B}_{(k'-1,s')} \otimes \cdots \otimes \mathbf{B}_{(1,s')} \right) \mathbf{G}_{(k')}^T \right\rangle \tag{24}
$$
$$
+ \sum_{s\in[S]\backslash s'} \left\langle \underline{\mathbf{G}} \times_1 \mathbf{B}_{(1,s)} \times_2 \cdots \times_K \mathbf{B}_{(K,s)}, \underline{\mathbf{X}} \right\rangle.
$$

Now we define the following notations in order to keep the expressions in (24) concise. From the first summand in (24) we define

$$
\boldsymbol{\omega}_{(k',s')} = \mathrm{vec}\left( \mathbf{X}_{(k')} \left( \mathbf{B}_{(K,s')} \otimes \cdots \otimes \mathbf{B}_{(k'+1,s')} \otimes \mathbf{B}_{(k'-1,s')} \otimes \cdots \otimes \mathbf{B}_{(1,s')} \right) \mathbf{G}_{(k')}^T \right), \tag{25}
$$

and from the second summand in (24) we define

$$
\gamma_{(k',s')} = \sum_{s\in[S]\backslash s'} \left\langle \underline{\mathbf{G}} \times_1 \mathbf{B}_{(1,s)} \times_2 \cdots \times_K \mathbf{B}_{(K,s)}, \underline{\mathbf{X}} \right\rangle. \tag{26}
$$

We can then express (24) as

$$
\langle \underline{\mathbf{B}}, \underline{\mathbf{X}} \rangle = \left\langle \mathrm{vec}(\mathbf{B}_{(k',s')}, \boldsymbol{\omega}_{(k',s')}) \right\rangle + \gamma_{(k',s')} = \left\langle \left[ \mathrm{vec}(\mathbf{B}_{(k',s')}), 1 \right], \left[ \boldsymbol{\omega}_{(k',s')}, \gamma_{(k',s')} \right] \right\rangle. \tag{27}
$$

By defining $\widetilde{\mathbf{x}} \triangleq \left[ \boldsymbol{\omega}_{(k',s')}, \gamma_{(k',s')} \right]$, we rewrite (23) as

$$\underset{\mathbf{B}_{(k',s')} \in \mathbb{O}^{m_{k'} \times r_{k'}}}{\arg\min} \sum_{i=1}^n \left( \left\langle \left[ \mathrm{vec}(\mathbf{B}_{(k',s')}), 1 \right], \widetilde{\mathbf{x}}_i \right\rangle \right) T(y_i) - a \left( \left\langle \left[ \mathrm{vec}(\mathbf{B}_{(k',s')}), 1 \right], \widetilde{\mathbf{x}}_i \right\rangle \right). \tag{28}$$

The most important insight here is that the resulting problem can be viewed as a parameter estimation problem for GLMs with $\mathbf{B}_{(k',s')}$ as the 'parameter' and $\widetilde{\mathbf{x}}_i$ as the 'predictor' or structured covariates. Estimating $\mathbf{B}_{(k',s')}$ alone in particular results in a low-dimensional problem with $m_{k'} r_{k'}$ parameters. We solve (23) via projected gradient descent. Using the above defined notations, the gradient of $\mathcal{L}_n(\mathbf{B}_{(k',s')})$ with respect to $\mathbf{B}_{(k',s')}$ is

$$\frac{\partial \mathcal{L}_n}{\partial \mathrm{vec}(\mathbf{B}_{k',s'})} = \sum_{i=1}^n \left( T(y_i) - g^{-1} \left( \left\langle \left[ \mathrm{vec}(\mathbf{B}_{(k',s')}), 1 \right], \widetilde{\mathbf{x}}_i \right\rangle \right) \right) \widetilde{\mathbf{x}}_i, \tag{29}$$

and the projection operator $\mathcal{H} : \mathbb{R}^{m_{k'} \times r_{k'}} \longrightarrow \mathbb{O}^{m_{k'} \times r_{k'}}$ that projects the obtained iterate onto the manifold of orthogonal matrices is defined as

$$\mathcal{H}(\mathbf{B}_{(k',s')}) \triangleq \underset{\widehat{\mathbf{B}} \in \mathbb{O}^{m_{k'} \times r_{k'}}}{\arg\min} \left\| \widehat{\mathbf{B}} - \mathbf{B}_{(k',s')} \right\|_F^2. \tag{30}$$

The solution to the problem in (30) is simply obtained using the QR decomposition of $\mathbf{B}_{(k',s')}$.

## 4.2 Estimating the Core Tensor

For core tensor $\underline{\mathbf{G}}$, we are only required to perform gradient descent. The optimisation problem is

$$\underset{\underline{\mathbf{G}}}{\arg\min} \, \mathcal{L}_n\left(\underline{\mathbf{G}}\right), \text{ where} \tag{31}$$

$$\mathcal{L}_n\left(\underline{\mathbf{G}}\right) = \sum_{i=1}^n \left( \left\langle \mathbf{g}, \sum_{s\in[S]} \left( \bigotimes_{-[K]} \mathbf{B}_{(k,s)} \right)^T \mathbf{x}_i \right\rangle \right) T(y_i) - a \left( \left\langle \mathbf{g}, \sum_{s\in[S]} \left( \bigotimes_{-[K]} \mathbf{B}_{(k,s)} \right)^T \mathbf{x}_i \right\rangle \right).$$

In order to see the derivative of the expression in (31), notice that,

$$\langle \underline{\mathbf{B}}, \underline{\mathbf{X}} \rangle = \langle \mathrm{vec}(\underline{\mathbf{B}}), \mathrm{vec}(\underline{\mathbf{X}}) \rangle = \sum_{s\in[S]} \left\langle \left( \bigotimes_{-[K]} \mathbf{B}_{(k,s)} \right) \mathbf{g}, \mathbf{x} \right\rangle = \left\langle \mathbf{g}, \sum_{s\in[S]} \left( \bigotimes_{-[K]} \mathbf{B}_{(k,s)} \right)^T \mathbf{x} \right\rangle. \tag{32}$$

With a slight overload of notation we also define the following:

$$\widetilde{\mathbf{x}} = \sum_{s\in[S]} \left( \mathbf{B}_{(K,s)} \otimes \cdots \otimes \mathbf{B}_{(1,s)} \right)^T \mathbf{x}, \tag{33}$$

and further express (32) and (31) as

$$\langle \underline{\mathbf{B}}, \underline{\mathbf{X}} \rangle = \langle \mathbf{g}, \widetilde{\mathbf{x}} \rangle \tag{34}$$

and

$$\underset{\underline{\mathbf{G}}}{\arg\min} \sum_{i=1}^n \left( \langle \mathbf{g}, \widetilde{\mathbf{x}}_i \rangle \right) T(y_i) - a \left( \langle \mathbf{g}, \widetilde{\mathbf{x}}_i \rangle \right), \tag{35}$$

respectively. Once again, the resulting problem can be viewed as a parameter estimation problem for GLMs with $\underline{\mathbf{G}}$ as the 'parameter' and $\widetilde{\mathbf{x}}_i$ as the 'predictor' or structured covariates. Estimating $\underline{\mathbf{G}}$ alone results in a low-dimensional problem with $\prod_{k\in[K]} r_k$ parameters. We solve (31) via gradient descent, where the gradient of $\mathcal{L}_n(\underline{\mathbf{G}})$ with respect to $\underline{\mathbf{G}}$ is

$$\frac{\partial \mathcal{L}_n}{\partial \mathrm{vec}(\underline{\mathbf{G}})} = \sum_{i=1}^n \left( T(y_i) - g^{-1} \left( \langle \mathbf{g}, \widetilde{\mathbf{x}}_i \rangle \right) \right) \widetilde{\mathbf{x}}_i. \tag{36}$$

## 4.3 Final Algorithm: LSRTR

We summarise the procedure discussed above in Algorithm 1 and we name our algorithm Low Separation Rank Tensor Regression (LSRTR). We also show the prediction procedure performed using the estimated coefficient tensor in Algorithm 2, where we calculate the mean $\mathbb{E}[y|\underline{\mathbf{X}}]$ of the posterior distribution based on the estimated coefficient tensor $\underline{\mathbf{B}}$ from Algorithm 1. The posterior mean allows us to make predictions for an observation $y$ and report confidence probabilities. Note that the convergence of BCD on non-convex problems is a function of the initialisation. In this work, we initialise LSRTR randomly on the constraint set. That is: an LSR-structured tensor with $(KS)$ orthogonal factor matrices, randomly generated on the Stiefel manifold, and a core tensor with random Gaussian entries. We also note that though BCD algorithms such as LSRTR are popular amongst tensor-structured regression works and have been shown to be effective in practice (Zhou et al., 2013; Tan et al., 2013; Li et al., 2018; Zhang & Jiang, 2016), BCD algorithms in prior tensor-structured GLM works do not explicitly exploit the coefficient tensor's Kronecker structure that appears upon vectorization. Keeping a common core tensor and maintaining the LSR matrix structure in (14) allows us to explore other parameter estimation algorithms that exploit the Kronecker matrix structure, similar to those in existing dictionary learning works (Ghassemi et al., 2020). However, we leave this for future work.

We next provide a brief discussion on the per-iteration computational complexity of each sub-problem of LSRTR. For factor matrix $\mathbf{B}_{(k,s)}$, each gradient step has complexity of order $\mathcal{O}(m_k r_k)$, and the QR projection step has complexity of order $\mathcal{O}((m_k r_k)^3)$. Thus the per-iteration computational complexity of estimating a factor matrix is $\mathcal{O}((m_k r_k)^3)$. Since there is no projection when estimating the core tensor $\underline{\mathbf{G}}$, its per-iteration sample complexity is just $\mathcal{O}(\prod_k r_k)$.

---

**Algorithm 1** LSRTR: A block coordinate descent algorithm for LSR-TGLMs

1: **Input:** $n$ training samples $\{\underline{\mathbf{X}}_i, y_i\}_{i=1}^n$, step size $\alpha$, separation rank $S$, tensor rank $(r_1, r_2, \ldots, r_K)$.
2: **Initialise:** Factor matrices $\mathbf{B}_{(k,s)}^0 \ \forall \ k \in [K], s \in [S]$, core tensor $\underline{\mathbf{G}}^0$ and $t \leftarrow 0$.
3: **repeat**:
4:     **for** $s' \in [S]$ **do**
5:         **for** $k' \in [K]$ **do**
6:             $\widetilde{\mathbf{B}}_{(k',s')}^{(t)} \leftarrow \text{vec}\left(\mathbf{B}_{(k',s')}^{(t)}\right) - \alpha \sum_{i=1}^n \left(T(y_i) - g^{-1}\left(\left\langle\left[\text{vec}(\mathbf{B}_{(k',s')}), 1\right], \widetilde{\mathbf{x}}_i\right\rangle\right)\right) \widetilde{\mathbf{x}}_i$
7:             $\mathbf{B}_{(k',s')}^{(t+1)} \leftarrow \mathcal{H}\left(\widetilde{\mathbf{B}}_{(k',s')}^{(t)}\right)$
8:         **end for**
9:     **end for**
10:     $\widetilde{\underline{\mathbf{G}}}^{(t+1)} \leftarrow \text{vec}(\underline{\mathbf{G}}^{(t)}) - \alpha \sum_{i=1}^n \left(T(y_i) - g^{-1}\left(\langle \mathbf{g}, \widetilde{\mathbf{x}}_i\rangle\right)\right) \widetilde{\mathbf{x}}_i$
11:     $t \leftarrow t + 1$
12: **until** convergence
13: **return** $\widehat{\underline{\mathbf{B}}} \leftarrow \sum_{s \in [S]} \underline{\mathbf{G}}^{(t)} \times_1 \mathbf{B}_{(1,s)}^{(t)} \times_2 \cdots \times_K \mathbf{B}_{(K,s)}^{(t)}$

---

**Algorithm 2** Posterior prediction for LSR-TGLMs

    **Input** Estimate $\widehat{\underline{\mathbf{B}}} \in \mathbb{R}^{m_1 \times \cdots \times m_K}$ and $n_{te}$ test data points $\{\underline{\mathbf{X}}_i\}_{i=1}^{n_{te}}$
    **Output** Expectation $\widehat{\boldsymbol{\mu}} = [\mathbb{E}[y_1|\underline{\mathbf{X}}_1], \mathbb{E}[y_2|\underline{\mathbf{X}}_2] \ldots \mathbb{E}[y_{n_{te}}|\underline{\mathbf{X}}_{n_{te}}]$
1: Define: $\underline{\mathcal{X}} \triangleq [\underline{\mathbf{X}}_1, \underline{\mathbf{X}}_2, \ldots, \underline{\mathbf{X}}_{n_{te}}]$
2: Compute $\widehat{\boldsymbol{\mu}}$ for input $\underline{\mathcal{X}}$ as: $\widehat{\boldsymbol{\mu}} = g^{-1}(\langle\widehat{\underline{\mathbf{B}}}, \underline{\mathcal{X}}\rangle)$
3: **return** $\widehat{\boldsymbol{\mu}}$

---

In this work, we do not provide convergence guarantees for LSRTR; this is a non-trivial task that we defer to future work. The main challenge in proving convergence is the constraint set in LSRTR, which is a product space of Stiefel manifolds and is neither closed nor convex. The non-convexity of the constraint space makes a convergence analysis for LSRTR difficult since we cannot apply general results for the convergence of BCD (Bertsekas, 2016). Therefore, one can at best expect LSRTR to converge to a local minimum. The

non-convexity also presents challenges for analysing projections, in particular when using the QR decomposition. More specifically, projection operators in projected gradient methods must be non-expansive in order to prevent potential amplification of the estimation error. Since the general method of proving the non-expansiveness of a projection also assumes a closed and convex constraint set – which the Stiefel manifold is not (Bertsekas, 2009) – demonstrating the non-expansiveness of QR projection in the general sense is also non-trivial. In many works, however, the QR decomposition is a common tool for projecting onto and/or optimising over the Stiefel manifold (Absil et al., 2008), yet proving stronger guarantees on optimisation over Stiefel manifold is a larger and non-trivial problem. Therefore, we leave the theoretical analysis of Algorithm 1 as a significant direction for future works. However, we make some promising remarks on its practical performance, convergence, and projection behaviour. Specifically in Section 5 we assess the performance of our proposed algorithm on various GLM problems for 2-dimensional and 3-dimensional synthetic data. Moreover, in Section 5.2, we use the LSR-TGLM model and the proposed LSRTR algorithm to solve classification problems on several real-world medical imaging datasets.

## 5    Numerical Study and Experiments on Medical Imaging Data

In this section we provide a comprehensive numerical study in order to assess the efficacy of the LSR-TGLM regression model and the corresponding proposed LSRTR algorithm. The objectives of this study are three-fold. First, we investigate the performance of our algorithm, particularly: 1) The performance of our proposed approach (parameter estimation and prediction) against a substantial growth in number of parameters, and 2) The performance of our algorithm, particularly when faced with a large increase in sample size. Secondly, we compare the estimation and prediction accuracy gains of the LSR model to current regression models and algorithms in the literature on synthetic data (Zhou et al., 2013; Li et al., 2018; Zhang et al., 2020; McCullagh & Nelder, 2019; Seber & Lee, 2003). Thirdly, we assess the performance of the LSR-TGLM model on classification problems for medical imaging datasets.

### 5.1    Experiments on Synthetic Data

We begin with experiments on synthetic data for three GLM problems: linear regression, logistic regression and Poisson regression. The purpose of the experiments is to answer the following questions:

1. When the underlying coefficient tensor has an LSR structure, does the proposed LSRTR algorithm lead to reliable parameter estimation? Specifically, does our algorithm achieve a reduction in sample complexity and estimation error compared to vector-based methods (for a fixed sample size)?

2. Does our proposed algorithm: 1) provide reliable parameter estimation and prediction against a substantial growth in number of parameters and 2) have per-iteration computation time comparable to other tensor-based methods, when faced with a large increase in sample size?

3. Empirical convergence behaviour: does the proposed algorithm converge to a stationary point?

In our experiments we compute and report the following:

1. A coefficient tensor $\widehat{\underline{\mathbf{B}}}$ estimated through the following learning methods: ($i$) The LSRTR method (Algorithm 1), ($ii$) Low-rank Tucker model methods, specifically, a 'block relaxation' algorithm for parameter estimation of Tucker-structured GLMs proposed by Li et al. (2018) (we will call this procedure TTR), and a similar iterative procedure for logistic Tucker regression (LTuR) with Frobenius norm regularization proposed by Zhang & Jiang (2016), ($iii$) unstructured (vector-based) methods for regression. In the case of ($iii$) we use Least Squares (LS) for linear regression, a first-order method we name LR (Seber & Lee, 2003) for logistic regression, and a GLM fitting algorithm we name PR, with log link function from McCullagh & Nelder (2019) for Poisson regression. The performance of each method is evaluated via the normalised estimation error defined as $\frac{\left\|\underline{\mathbf{B}}-\widehat{\underline{\mathbf{B}}}\right\|_F^2}{\left\|\underline{\mathbf{B}}\right\|_F^2}$.

2. A predicted response vector $\widehat{\mathbf{y}}$. The prediction accuracy for linear, logistic and Poisson regression is evaluated via the normalised squared error defined as $\frac{\|\widehat{\mathbf{y}}-\mathbf{y}\|_2^2}{\|\mathbf{y}\|_2^2}$, the Mean Absolute Error (MAE) defined as $\frac{\|\widehat{\mathbf{y}}-\mathbf{y}\|_1}{n_{te}}$ for $n_{te}$ testing samples, and the normalised squared logarithmic error defined as $\frac{\|\log(\widehat{\mathbf{y}}+1)-\log(\mathbf{y}+1)\|_2^2}{\|\log(\mathbf{y}+1)\|_2^2}$, respectively.

3. The magnitude of the computed gradient of the loss function, denoted as $\|\nabla\mathcal{L}(\cdot)\|_2$, at the final iteration of every sub-problem in LSRTR.

Since we are reporting normalised estimation error, we only consider errors below 1. Indeed, the normalised error compares the performance of an estimator relative to the 'trivial estimator' that always outputs 0. A normalised error of 1 occurs when the estimate is 0, yet an algorithm can potentially perform worse than the trivial estimator, allowing an error greater than 1. Such an error is insubstantial and we disregard it when analysing the performance of models and algorithms.

### 5.1.1  Experimental Setup

In our experiments we generate an $(r_1, r_2, \ldots, r_K)$-rank LSR-structured coefficient tensor $\underline{\mathbf{B}} \in \mathbb{R}^{m_1 \times \cdots \times m_K}$ with separation rank $S$ in (9) as follows. The entries of the core tensor $\underline{\mathbf{G}}$ are sampled from a Gaussian$(0, \frac{1}{r_1 r_2 \ldots r_K})$ distribution. Each factor matrix $\mathbf{B}_{(k,s)}$ is constructed by first constructing matrix $\mathbf{A}_{(k,s)} \in \mathbb{R}^{m_k \times r_k}$ with independent standard normal entries. The QR decomposition is then applied to $\mathbf{A}_{(k,s)}$ in order to obtain the orthonormal matrix $\widetilde{\mathbf{A}}_{(k,s)} \in \mathbb{O}^{m_k \times m_k}$. The first $r_k$ columns are then extracted from $\widetilde{\mathbf{A}}_{(k,s)}$ in order to obtain $\mathbf{B}_{(k,s)}$. With the above construction, we have $\|\underline{\mathbf{B}}\|_F^2 \leq S^2 \widetilde{r} \|\underline{\mathbf{G}}\|_F^2$ (we derive a more precise bound on $\|\underline{\mathbf{B}}\|_F^2$ in Section 6.3). We also generate $n$ data samples $\{\underline{\mathbf{X}}, y\}$ used for training and estimation, and $n_{te}$ samples used for testing and prediction. We generate independent tensor-structured covariates $\{\underline{\mathbf{X}}_i\}_{i=1}^n$ and $\{\underline{\mathbf{X}}_i\}_{i=1}^{n_{te}}$ with zero-mean Gaussian entries and covariance $\mathbf{\Sigma}_x$, and observations $\{y_i\}_{i=1}^n$ and $\{y_i\}_{i=1}^{n_{te}}$ which are randomly generated according to the probabilistic model in (13) and with appropriate link function $g(\mu)$. Additionally, $z$ is assumed to follow a standard Normal distribution. Thus, for linear regression, the observation $y_i$ conditioned on data $\underline{\mathbf{X}}_i$ follows a Gaussian distribution, i.e., $y_i \sim \mathcal{N}\left(\langle\underline{\mathbf{B}}, \underline{\mathbf{X}}_i\rangle, 1\right)$. In logistic regression, the observation $y_i$ conditioned on data $\underline{\mathbf{X}}_i$ follows a Bernoulli distribution, i.e., $y_i \sim \text{Bernoulli}\left(\frac{1}{1+\exp(-\langle\underline{\mathbf{B}}, \underline{\mathbf{X}}_i\rangle)}\right)$. Finally, in Poisson regression, the observation $y_i$ conditioned on data $\underline{\mathbf{X}}_i$ follows a Poisson distribution, i.e., $y_i \sim \text{Poisson}\left(\exp(\langle\underline{\mathbf{B}}, \underline{\mathbf{X}}_i\rangle)\right)$. The experiment for each GLM problem is performed for various model sizes $(m_k)_{k\in[K]}$, tensor ranks $(r_k)_{k\in[K]}$ and separation rank $S$. The coefficient tensor $\underline{\mathbf{B}}$ and data samples $\{\underline{\mathbf{X}}_i, y_i\}_{i=1}^n$ and $\{\underline{\mathbf{X}}_i, y_i\}_{i=1}^{n_{te}}$ are generated as described above and each experiment is repeated for increasing value of $n$. We then compute the estimate $\widehat{\underline{\mathbf{B}}}$ and prediction $\widehat{\mathbf{y}} = [\widehat{y}_1, \widehat{y}_2, \ldots, \widehat{y}_{n_{te}}] = [\widehat{\mu}_1, \widehat{\mu}_2, \ldots, \widehat{\mu}_{n_{te}}]$ where $\widehat{\mu}_i = g^{(-1)}\left(\langle\widehat{\underline{\mathbf{B}}}, \underline{\mathbf{X}}_i\rangle\right)$. Finally, for each GLM problem, unless otherwise stated, we conduct 50 repetitions of each experiment (for a fixed coefficient tensor $\underline{\mathbf{B}}$) and average the error over the repetitions.

### 5.1.2  2D Synthetic Data

We begin with the two-dimensional (matrix) case with $K = 2$ as many medical imaging data fall under this case (x-rays, EEG, fiber-bundle images.). In this set of experiments the underlying coefficient matrix $\mathbf{B}$ has dimensions $m \times m$ and rank $r$. The dimensions and rank of $\mathbf{B}$ are thus represented as the tuple $(m, r)$. For all GLM problems we fix the separation rank $S = 2$; thus the number of learnable parameters is $S(mr + mr) + r^2 = 4mr + r^2$. We now proceed with the specific experimental setups of the three GLM problems under study.

For linear regression we consider various model sizes and ranks, i.e., $m \in \{64, 128, 256\}$, $r \in \{4, 8\}$. We range sample size $n$ from (roughly) the degrees of freedom (number of learnable parameters) of the smallest model size to no smaller than the degrees of freedom of the largest model size. The smallest model size is for the tuple $(m = 64, \ r = 4)$ and the largest model size is for the tuple $(m = 256, \ r = 8)$, which, under the LSR

structure, have $4(64 \times 4) + 4^2 = 1040$ and $4(256 \times 8) + 8^2 = 8256$ learnable parameters, respectively. We compare the performance of LSRTR to TTR and LS. Figure 2 reports the mean normalised estimation and prediction accuracy, respectively, across 50 repetitions. The shaded regions are one standard deviation of mean estimation and prediction accuracy, based on 50 replications.

For logistic and Poisson regression we consider $m \in \{32, 64, 128\}$ and $r \in \{4, 8\}$. Since the models are non-linear, we follow the heuristic rule of ranging sample size $n$ from (roughly) 5 times the degrees of freedom of the smallest model size to no smaller than that of the largest model size (Li et al., 2018). The smallest model size is for the tuple ($m = 32$, $r = 4$) and the largest model size is for the tuple ($m = 128$, $r = 8$), which, under the LSR structure, have $4(32 \times 4) + 4^2 = 528$ and $4(128 \times 8) + 8^2 = 4160$ learnable parameters, respectively. We compare the performance of the LSRTR method to $i$) LTuR and LR and $ii$) TTR and PR for logistic and Poisson regression, respectively. Figure 3 and Figure 10 in the appendix report the estimation and prediction accuracy for logistic and Poisson regression, respectively, with shaded regions depicting one standard deviation of mean estimation and prediction accuracy, based on 50 replications.

We pause to make a few remarks. Firstly, in the case of synthetic data, the rank $r$ and separation rank $S$ are assumed to be known and other algorithmic parameters (the step size $\alpha$) are set using separate validation experiments. Secondly, though the generated coefficient $\mathbf{B}$ is a matrix, the numerical study by Li et al. (2018), shows that the low-rank Tucker model effectively estimates two-dimensional parameters and the TTR algorithm performs well in the matrix setting even with huge increases in model and sample sizes. The study by Li et al. (2018) also shows how TTR generalizes CP tensor regression algorithms and improves upon the performance of several other methods (such as PCA and Bayesian regression methods) in terms of estimation and prediction accuracy in experiments with synthetic data and for several regression types (linear and logistic) (Li et al., 2018). For these reasons we limit our scope of comparative algorithms for matrix-structured regression to TTR.

We also note that since we report normalised errors, we do not investigate any errors greater than 1 in Figures 2, 3 and 10. The plots in Figures 2, 3 and 10 show that, for all algorithms in all GLM problems, the estimation and prediction accuracy improves and the shaded regions (which characterise the standard deviation of the error points over 50 repetitions) decay with an increase in observations. In particular, on the whole, the tensor-based methods LSRTR, TTR and LTuR outperform the vector-based methods LR, LS and PR, particularly with larger model size $m$. This is because the number of observations provided in the experiments is typically much lower than the sample complexity of the vector methods. For example, consider linear regression with $m = 256$. LS requires at least $256 \times 256 = 65{,}536$ observations for reliable parameter estimation, while if we set $r = 4$ and $S = 2$, TTR and LSRTR require only 2064 and 4112 observations for reliable parameter estimation, respectively. Additionally, for TTR, LTuR and LSRTR, as the core tensor dimensions grow (i.e., as we increase the Tucker rank of our tensor models), we require a relatively larger sample size to achieve better accuracy. This is not surprising as a larger core tensor increases the number of parameters to be learned. Figures 2, 3 and 10 also show that for all GLM problems under study (namely linear, logistic and Poisson regression), the LSRTR algorithm shows 'consistent' performance for estimation and prediction. What we mean by this is that, even with a substantial growth in model size, the estimation error induced by LSRTR decreases with increasing sample size, and the LSRTR algorithm can still achieve acceptable estimation and prediction errors as long as the number of observations provided is large enough. Additionally, we can see that LSRTR algorithm can avoid over-fitting in the prediction step even with a very large number of observations.

We also gain some additional insights from the results. First, if the underlying parameter model $\mathbf{B}$ is LSR-structured, then LSRTR can recover $\mathbf{B}$ with a relatively low number of observations, in the sense that the number of observations required is proportional to the intrinsic degrees of freedom in the LSR model (rather than the extrinsic dimensionality of $\mathbf{B}$). This implies that though we do not provide any theoretical guarantees for Algorithm 1, it can be employed successfully in practice. Second and more interestingly, if the underlying parameter model $\mathbf{B}$ is LSR-structured, then imposing the Tucker model on $\mathbf{B}$ in the parameter estimation procedure – as in TTR and LTuR algorithms – results in less accurate estimation compared to LSRTR. Particularly, we see that for relatively large model sizes, such as in Figures 2e and 2f, or for non-linear models, such as in Figures 3e, and 10e, TTR and LTuR algorithms' performance plateaus, even when the number of observations has far exceeded the degrees of freedom of a Tucker-structured coefficient matrix

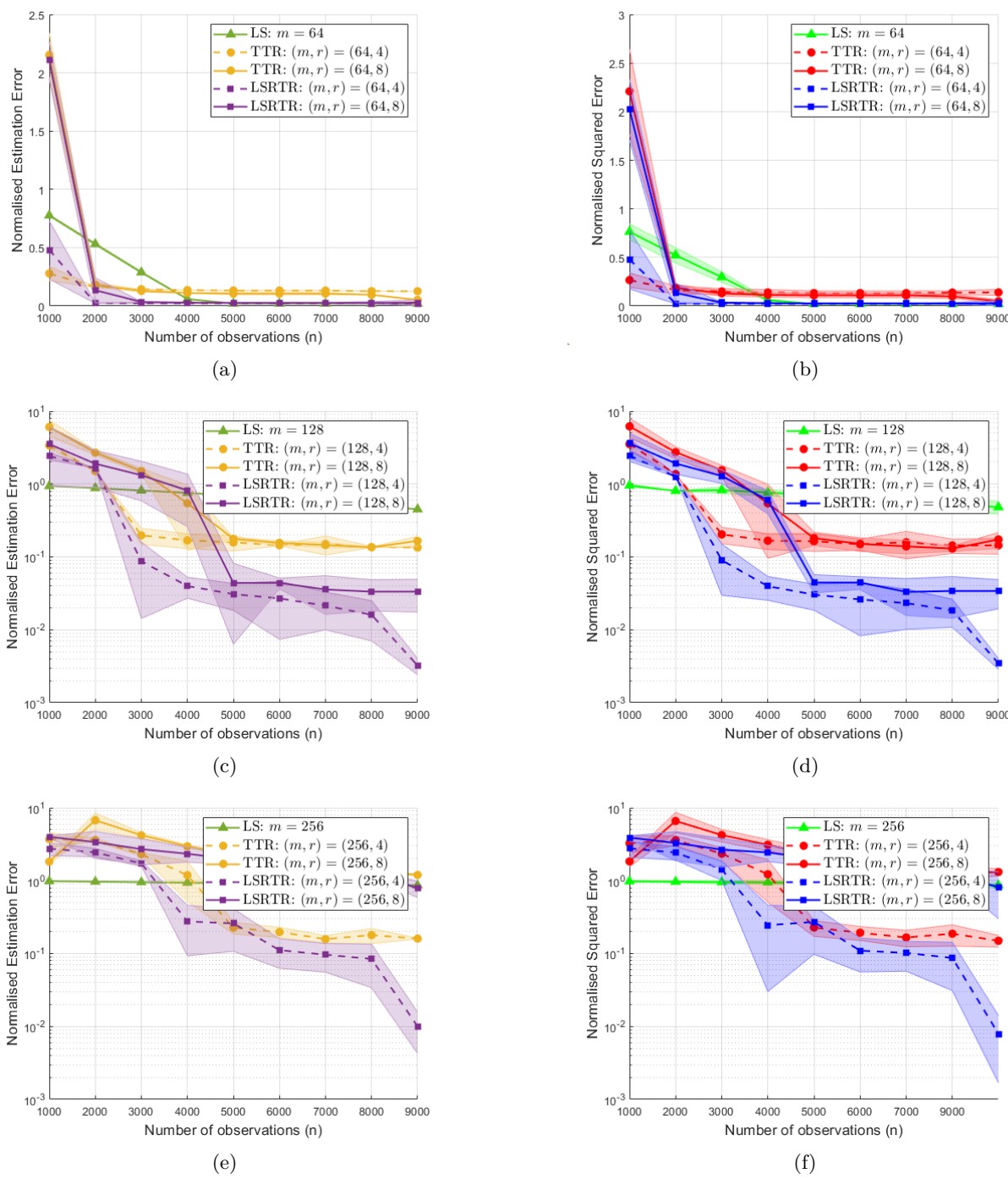

Figure 2: Comparison of LSRTR with LS and TTR for two-dimensional synthetic data when $m \in \{64, 128, 256\}$, $r \in \{4, 8\}$ and $S = 2$. Normalised estimation error for $m = 64$, 128, and 256 is shown in $(a)$, $(c)$, and $(e)$, respectively. Normalised prediction error for $m = 64$, 128, and 256 is shown in $(b)$, $(d)$, and $(f)$, respectively. Each marker represents the mean normalised estimation/prediction errors, over 50 repetitions. The shaded regions correspond to one standard deviation of the mean normalised errors.

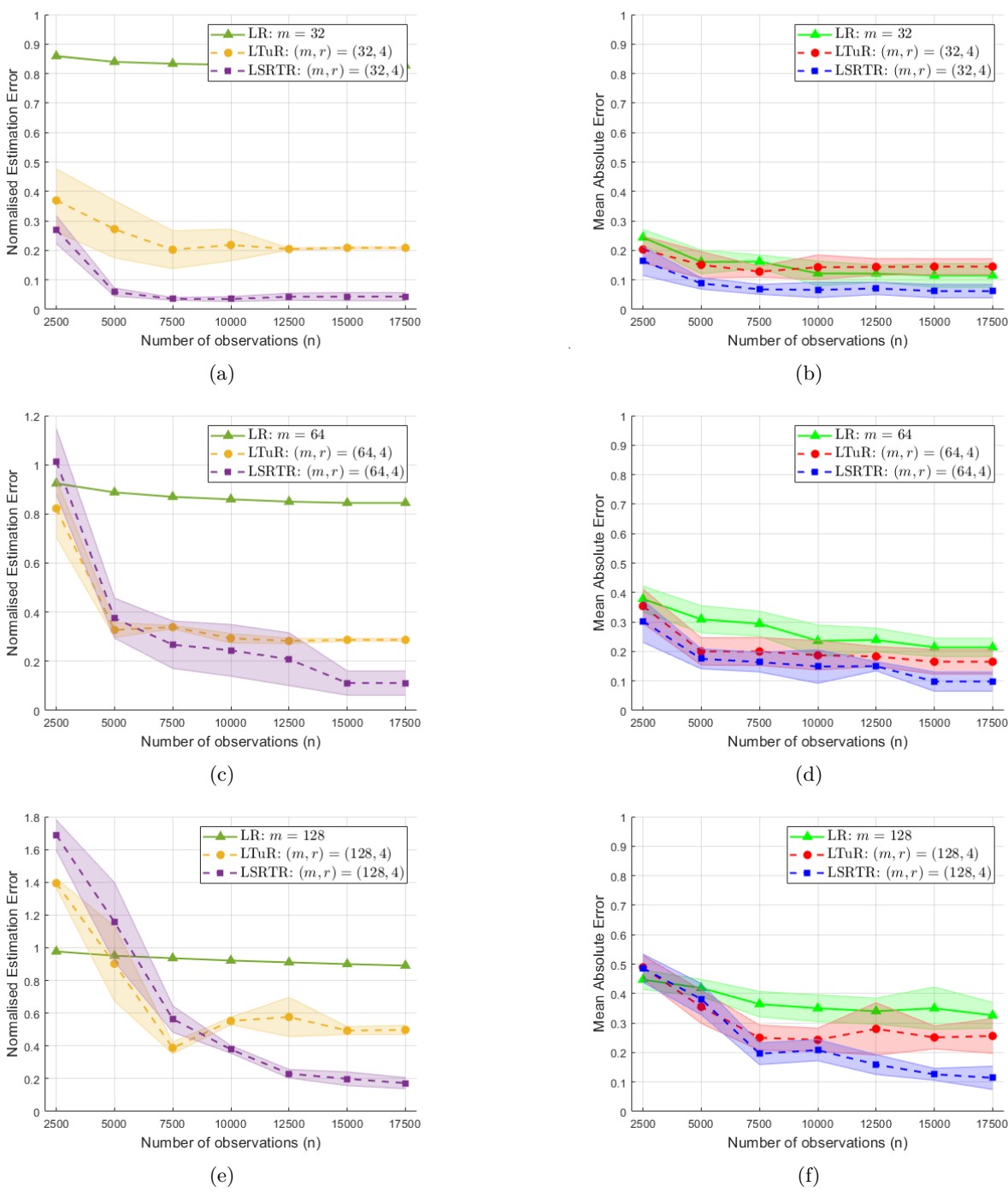

Figure 3: Comparison of LSRTR with LR and LTuR for two-dimensional synthetic data when $m \in \{32, 64, 128\}$, $r = 4$ and $S = 2$. Normalised estimation error for $m = 32$, 64, and 128 is shown in (*a*), (*c*), and (*e*), respectively. Normalised prediction error for $m = 32$, 64, and 128 is shown in (*b*), (*d*), and (*f*), respectively. Each marker represents the mean normalised estimation/prediction errors, over 50 repetitions. The shaded regions correspond to one standard deviation of the mean normalised errors.

**B**. In fact, even LS and LR outperform TTR and LTuR when the model size is small enough and enough observations have been provided. Though we expect LSRTR to perform better than TTR or LTuR when **B** is originally LSR-structured, the results suggest that if regression problems with real data have model parameters that are approximately-LSR structured, then TTR or LTuR may not be suitable algorithms and we can motivate the use of algorithms such as LSRTR that allow for the consideration of a richer class of tensors that may lead to more reliable estimation.

Notice also that in Figures 2e and 2f (as well as in Figures 3e and 3f), when $r = 8$, the normalised errors are greater than 1. Here, LSRTR is operating in the under-sampling regime, where the model size is very large and the number of available samples (9000) is just about the number of learnable parameters of the model. In such a setting we expect the estimation error to be large and to decrease with increasing samples. We also make specific remarks on Figures 3c and 3e. In the case of $m = 64$, we see that LSRTR shows lower estimation error at $n = 17500$ (0.11 vs 0.29), yet has a larger standard deviation (shaded region) over 50 experiment replications (0.04 vs 0.006). This is an acceptable standard deviation, especially considering the reduction in estimation error compared to the Tucker-based LTuR algorithm. In the case of $m = 128$, LTuR shows lower estimation error than LSRTR until about $n = 7500$ (errors of 0.6 vs 0.4). This could be due to the fact that LSRTR is operating in the undersampling regime for $n = 7500$ (as LSR has more learnable parameters than Tucker). Additionally, we see in Figure 3e that the estimation error for LTuR increases after $n = 7500$, and in Figure 3f that LSRTR outperforms LTuR, which suggests that LTuR may be overfitting the GLM model.

We also make a specific remark on Figure 10a in the appendix, where we witness a slight increase in estimation error for $(m, r) = (32, 8)$ between $n = 5000$ and $n = 7500$ (0.025 vs 0.08). However, we see that the estimation error continues to decrease after $n = 7500$, and we do not witness this phenomena in any other experiment in Figure 10.

### 5.1.3   3D Synthetic Data

We have evaluated the performance of the LSRTR algorithm against an increase in model and sample size for $K = 2$. Now, we wish to further explore the performance of our method compared to the other state-of-the-art methods with 3-dimensional data ($K = 3$). In these experiments, the underlying coefficient tensor $\underline{\mathbf{B}}$ has dimensions $m_1 \times m_2 \times m_3$ and Tucker rank $\{r, r, r\}$. The dimensions and rank of $\underline{\mathbf{B}}$ are thus represented as the tuple $(\mathbf{m}, r)$ where $\mathbf{m} = [m_1, m_2, m_3]$. We study the linear regression problem and we fix the separation rank $S = 1$, i.e., $\underline{\mathbf{B}}$ is Tucker structured. Thus $\underline{\mathbf{B}}$ is a 3-dimensional Tucker tensor of dimension $m_1 \times m_2 \times m_3$ and rank $r$ along each mode, and the number of learnable parameters is $S(m_1 r + m_2 r + m_3 r) + r^3 = r(m_1 + m_2 + m_3 + r^2)$. We consider the model size $\mathbf{m} = [16, 32, 64]$, $r = 4$. The model size under the Tucker structure is $4(16 + 32 + 64 + 4^2) = 512$. We range sample size $n$ from (roughly) two times the degrees of freedom to roughly eight times the degrees of freedom. We compare the performance of LSRTR to TTR and LS. Figure 4 reports the estimation accuracy and prediction accuracy, respectively, with shaded regions depicting one standard deviation of mean estimation accuracy, based on 50 replications. For LSRTR the rank $r$ is assumed to be known and other parameters (the separation rank $S$ and the step size $\alpha$) are set using separate validation experiments. The separation rank was thus chosen as $S = 2$.

We observe from Figure 4 the same trends as in the previous set of experiments. We also conclude that even with the underlying coefficient tensor being Tucker-structured, our proposed LSRTR method in this experiment exhibits better statistical performance with fewer observations than the 'Tucker specific' algorithm TTR as well as LS. This suggests that the LSR model induces a richer class of solutions with greater representation power that is able to capture meaningful information in the data that the more compact state-of-the-art tensor models do not – even with increased sample size. The results also suggest that the projection step in LSRTR (imposing orthonormal factor matrices) may contribute to the enhanced performance of LSRTR in terms of estimation/prediction error (as TTR does not impose such a constraint).

Finally, we numerically investigate the per-iteration computational complexity of LSRTR vs TTR for three-dimensional synthetic data, for varying sample size $n$. We repeat the previous experiment, i.e., $\mathbf{m} = [16, 32, 64]$, $r = 4$, $S = 2$, over varying sample size ($n = 100$, $n = 300$, $n = 500$), and report the

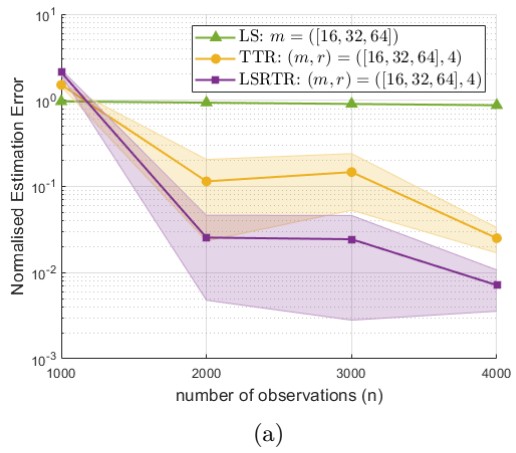 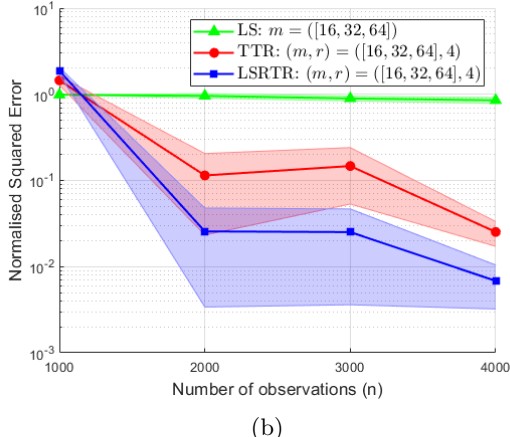

(a)  (b)

Figure 4: Comparison of LSRTR with LS and TTR for three-dimensional synthetic data when $\mathbf{m} = [16, 32, 64]$, $r = 4$, $S = 1$. For LSRTR, $S = 2$ was chosen. Normalised estimation error is shown in ($a$). Normalised prediction error is shown in ($b$). Each marker represents the mean normalised estimation/prediction errors, over 50 repetitions. The shaded regions correspond to one standard deviation of the mean normalised errors.

|  | n = 100 | n = 300 | n = 500 |
|---|---|---|---|
| **LSRTR** | $0.1$ $(7e - 5)$ | $0.37$ $(2e - 4)$ | $0.65$ $(5e - 4)$ |
| **TTR** | $0.048$ $(3e - 5)$ | $0.18$ $(1e - 4)$ | $0.31$ $(2e - 4)$ |

Table 2: Per-iteration computation time (in seconds) of LSRTR and TTR over varying sample size ($n = 100$, $n = 300$, $n = 500$). We report the mean (and variance) over 50 repetitions.

mean per-iteration computation time (in seconds) over 50 repetitions of the experiment, for LSRTR and TTR. Table 2 shows the results. We can see that LSRTR and TTR have comparable per-iteration computation times. In fact LSRTR has computation time of roughly $S = 2$ times that of TTR.

### 5.1.4 Convergence Behaviour of LSRTR

We have previously discussed the challenges of providing theoretical guarantees for the proposed algorithm, specifically an analysis of its convergence and the non-expansiveness of the QR projection. Nonetheless, we are still interested in developing an understanding of the practical behaviour of LSRTR. For two-dimensional synthetic data, for fixed sample size $n$, we repeat the experiment from Section 5.1.2, i.e., for linear regression of model sizes $(64, 4)$, $(128, 4)$ and $(256, 4)$, with $n = 7000$, and logistic regression of model sizes $(32, 4)$, $(64, 4)$ and $(128, 4)$, with $n = 7500$, and with $S = 2$. We report the per-iteration normalised estimation error for LSRTR and the magnitude of the computed gradient at the final iteration of every sub-problem in Algorithm 1. Figure 5 depicts the convergence behaviour of LSRTR. We see that the per-iteration normalised estimation error decreases with increasing number of iterations. In regards to the non-expansiveness of the QR projection, these results suggest that the error does not amplify, and the QR step does not prohibit LSRTR from finding a 'good estimate' of the underlying coefficient tensor. The decrease in the estimation error also suggests that LSRTR converges. Moreover, Table 3 depicts the magnitude of the computed gradient of the loss function at the final iteration of every sub-problem in LSRTR. We can see in all cases that the magnitudes of the gradients are very close to 0, suggesting the convergence of LSRTR to a stationary point.

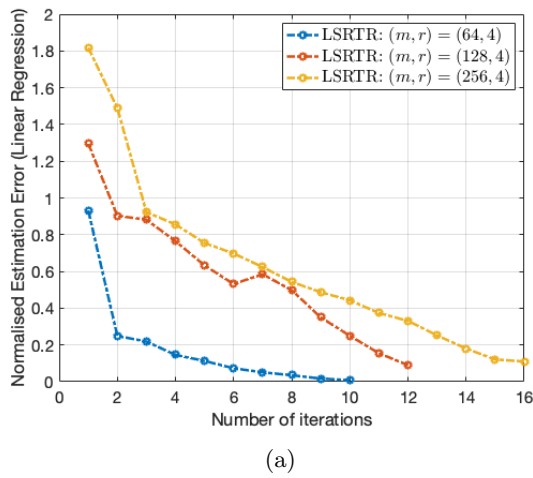
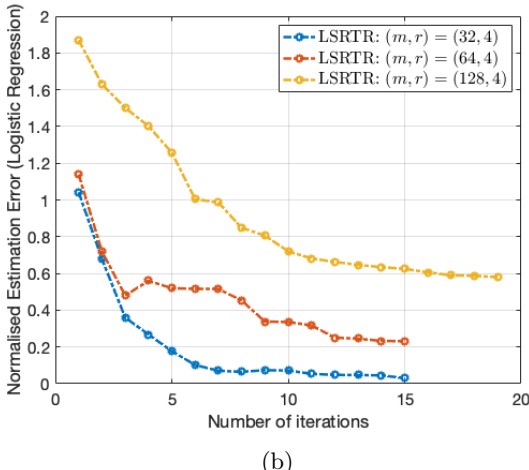

|(a)|(b)|

Figure 5: Convergence behaviour of LSRTR for two-dimensional synthetic data for linear regression when $m \in \{64, 128, 256\}$, and $r = 4$, for $n = 7000$, and logistic regression when $m \in \{32, 64, 128\}$, and $r = 4$, for $n = 7500$. For LSRTR, $S = 2$ was chosen. Normalised estimation error for linear and logistic regression is shown in $(a)$ and $(b)$, respectively.

| | **Linear Regression** $(m, r)$ | | | **Logistic Regression** $(m, r)$ | | |
|---|---|---|---|---|---|---|
| **gradient norm** | $(64, 4)$ | $(128, 4)$ | $(256, 4)$ | $(32, 4)$ | $(64, 4)$ | $(128, 4)$ |
| $\left\|\nabla\mathcal{L}(\mathbf{B}_{(1,1)})\right\|_2$ | $3.1 \times 10^{-3}$ | $5.2 \times 10^{-3}$ | $4.3 \times 10^{-3}$ | $0.24$ | $0.38$ | $0.21$ |
| $\left\|\nabla\mathcal{L}(\mathbf{B}_{(1,2)})\right\|_2$ | $7.8 \times 10^{-3}$ | $1.6 \times 10^{-3}$ | $9.3 \times 10^{-3}$ | $0.2$ | $0.32$ | $0.3$ |
| $\left\|\nabla\mathcal{L}(\mathbf{B}_{(2,1)})\right\|_2$ | $1.7 \times 10^{-2}$ | $4.7 \times 10^{-3}$ | $8.73 \times 10^{-4}$ | $0.4$ | $0.1$ | $0.18$ |
| $\left\|\nabla\mathcal{L}(\mathbf{B}_{(2,2)})\right\|_2$ | $3.5 \times 10^{-2}$ | $2.1 \times 10^{-3}$ | $4 \times 10^{-3}$ | $0.29$ | $0.15$ | $0.23$ |
| $\left\|\nabla\mathcal{L}(\underline{\mathbf{G}})\right\|_2$ | $7.42 \times 10^{-4}$ | $3.8 \times 10^{-3}$ | $1.2 \times 10^{-3}$ | $0.37$ | $0.28$ | $0.19$ |

Table 3: Magnitude of the final iteration gradient, per sub-problem, in linear and logistic regression.

## 5.2 Experiments on Medical Imaging Data

We move on to investigating our approach on medical imaging data. Regression analysis of medical imaging data can be a useful tool in medical decision making. We chose this type of data for several reasons. Medical images are usually multi-dimensional, and since data acquisition in medical sciences is expensive, the regression problems under study with such data is very high dimensional. Another major reason is that medical images model complex biological structures (such as brain maps). One expects the model parameter in the corresponding regression model to model this complexity. Lastly, the set of observations in medical imaging data is usually severely imbalanced. In other words, positive medical diagnoses are relatively rare, and thus a given set of observations will contain far fewer positive diagnoses than negative diagnoses. The high-dimensionality, complexity and imbalanced observation set of medical imaging data serve as an excellent assessment of the efficiency of our proposed model and algorithm. That is, we contend that due to the LSR model's favourable bias-variance trade-off and augmented representational power, LSR-TGLM may be well-suited to handling such data and may perform more robustly than some other compact tensor-structured GLMs. Here we study three datasets; ABIDE Autism (Craddock et al., 2013; Lodhi & Bajwa, 2020), ADHD200 (Bellec et al., 2017) and Vessel MNIST 3D (Yang et al., 2021b).

### 5.2.1 ABIDE Autism

The Autism Brain Imaging Data Exchange (ABIDE) dataset contains the resting state fMRI data collected from 98 subjects. The data has already been preprocessed for motion realignment and correction, slice timing correction and image normalisation. Each data sample corresponds to 111 cortical and sub-cortical brain regions scanned over 116 time periods (Craddock et al., 2013). Therefore each sample is a $111 \times 116$ matrix of fMRI data from one subject. Each observation is a binary response variable $y \in \{0, 1\}$ depicting a subject's diagnosis of either having autism ($y = 1$) or not ($y = 0$). The goal is to classify the subjects as either autistic or not. We perform a single train-test split procedure of 80 and 14 training and testing samples respectively. As for the autistic-control split, we choose 40 autistic and 40 control samples uniformly at random for training, and 14 test subjects the same way. The case-control ratio is this dataset is thus $1:1$. ABIDE Autism is the only balanced dataset in this set of experiments.

### 5.2.2 ADHD200

ADHD200 is a repository of resting state fMRI images of subjects from 8 research sites: Peking University, Brown University, Kennedy Krieger Institute, Donders Institute, NYU Child Study Center, Oregon Health and Science University, University of Pittsburgh and Washington University in St. Louis. The data includes brain maps of fractional Amplitude of Low-Frequency Fluctuations (fALFF) of 959 child subjects. fALFF brain maps can be useful in predicting Attention Deficit Hyperactivity Disorder (ADHD) in children. The data has been preprocessed, (ADHD-200 Consortium, 2012; Bellec et al., 2017), and we also perform standard preprocessing of the data such as removing missing observations or images with poor quality (a list of images with poor quality has been made public, (ADHD-200 Consortium, 2012)). Each data sample corresponds to a 3-dimensional patient fMRI brain scan (T1-weighted image) of size $121 \times 145 \times 121$. Figure 6 shows the axial, sagittal and coronal views (views across each tensor mode) of a fMRI data sample. Each observation is a binary response variable $y \in \{0, 1\}$ depicting a child subject's diagnosis of either having ADHD ($y = 1$) or being typically developing ($y = 0$). The objective here is classification of children subjects with ADHD. The dataset has already undergone a train-test split procedure. The training dataset consists of 762 samples, 280 of which are labelled as ADHD (hyperactive/impulsive, inattentive and combined) and 482 of which are labelled as typical (control). The testing dataset consists of 197 samples, where 76 and 93 are labelled as ADHD and control, respectively. The data is slightly imbalanced where the case-control ratio in the training and testing sets is $3:5$ and $4:5$, respectively.

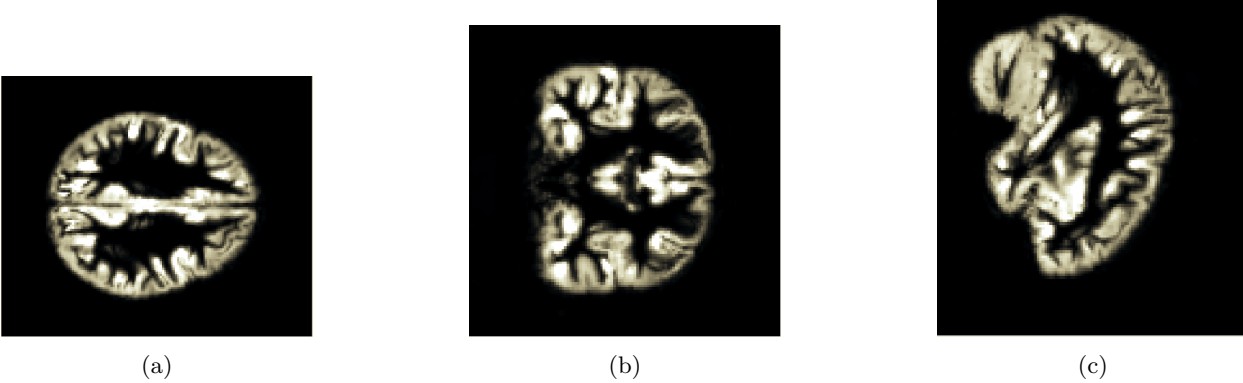

| (a) | (b) | (c) |

Figure 6: Axial ($a$), coronal ($b$), and sagittal ($c$) views of an fMRI data sample.

### 5.2.3 Vessel MNIST 3D

MedMNIST (Yang et al., 2021b) is a collection of medical imaging datasets suitable for various tasks such as classification and regression. One of the datasets is the Vessel 3D dataset (Yang et al., 2020) containing 1909 reconstructions of vessel segments from brain Magnetic Resonance Angiography (MRA) data of different

| Dataset | Covariate Dimensions | Train Size | Test Size |
|---|---|---|---|
| ABIDE Autism | $(111, 116)$ | 80 | 14 |
| ADHD200 | $(121, 145, 121)$ | 762 | 169 |
| Vessel MNIST 3D | $(28, 28, 28)$ | 1335 | 382 |

Table 4: Description of three medical imaging datasets (ABIDE ADHD, ADHD 200, and Vessel MNIST 3D). Described are the model dimensions and train and test sizes for all three datasets.

subjects. Each data point corresponds to 3-dimensional vessel segment of dimension $28 \times 28 \times 28$. Each vessel segment may or may not exhibit an intracranial aneurysm and the dataset is diverse in the sense that it includes a variety of shapes and scales of intracranial aneurysms. The data has also been preprocessed to restore incomplete scans made by neurosurgeons and to remove duplicated data. Each observation is a binary response variable $y \in \{0, 1\}$ depicting a vessel segment diagnosis of either having an aneurysm ($y = 1$) or being healthy ($y = 0$). The goal is to therefore classify each vessel segment as either exhibiting an aneurysm or being healthy. The dataset has already undergone a train-test split procedure. The training dataset consists of 1335 samples, 150 of which are labelled as 'aneurysm' and 1185 of which are labelled as 'healthy'. The testing dataset consists of 382 samples, where 43 and 339 are labelled as 'aneurysm' and 'healthy', respectively. The data is severely imbalanced where the case-control ratio in both the training and testing sets is $3 : 25$.

### 5.2.4 Results of Experiments on Medical Imaging Data

All three medical imaging datasets support a logistic regression problem. We estimate the unknown coefficient tensor $\underline{\mathbf{B}}$ through our LSRTR algorithm and use our estimate to predict the responses of the test data through a logistic regression model, i.e., we calculate the posterior probability using the sigmoid function $s = (1/(1 + \exp(\langle \underline{\mathbf{B}}, \underline{\mathbf{X}}_i \rangle + z)))$ for every test subject $i$ and use a cut-off threshold of 0.5 to classify each response as 1 or 0, where the positive class is chosen if $s > 0.5$. Parameters including the Tucker rank $(r_k)_{k \in [K]}$, the separation rank $S$ and the step size $\alpha$ are set – for each dataset – using separate validation experiments. On all three datasets we compare our method with four other methods, two of which have been previously introduced, namely, LR and LTuR. The third method is a low-rank CP model method (which we name LCPR), specifically a block coordinate descent approach for parameter estimation of CP structured logistic regression models proposed by Tan et al. (2013). The fourth method is a Support Vector Machine (SVM) with Gaussian Kernel (Hearst et al., 1998). In the case of the Vessel MNIST dataset, we also compare our method with an established baseline in the literature for Medical MNIST datasets: ResNet 50, which is residual neural network that is 50 layers deep. This neural network has also been augmented with a 3D convolutional layer designed to handle 3-dimensional data.

To evaluate the performance of each method we report the following scores: Specificity defined as the true negative rate, sensitivity defined as the true positive rate, the F1 score, the average accuracy score defined as the MAE, and the Area Under the Curve (AUC). We note that we penalise false positives and false negatives with the same severity and compute the MAE to be consistent with the current literature (Li et al., 2018; Zhou et al., 2013; Zhang & Jiang, 2016). We also note that it is important to use the F1 score and AUC, as they are more appropriate metrics in the case of class imbalance, compared to average accuracy. Table 4 summarises the description of each dataset and Tables 5a, 5b, and 5c summarise the results. The chosen rank $(r_k)_{k \in [K]}$ for ABIDE Autism, ADHD200 and Vessel MNIST 3D are $(6, 6)$, $(6, 6, 6)$ and $(3, 3, 3)$, respectively. The chosen LSR rank for LSRTR for ABIDE Autism, ADHD200 and Vessel MNIST 3D are $S = 2$, $S = 3$ and $S = 2$, respectively. The results show that the LSRTR method performs well in terms of sensitivity and average accuracy for all datasets and the vector based-method LR has poorer performance with all datasets, where for some datasets all observations are predicted as positive. Particularly with the ADHD200 dataset, we are faced with the challenge of efficient estimation as the sample size is very small with respect to the model size (762 vs. 2,097,152). The relatively good performance of LSRTR suggests that LSRTR performs well in the high-dimensional setting (i.e., when the sample size is significantly smaller than the number of

covariates). We notice that LR performs very poorly with ADHD200, as there are over 2 million parameters to estimate. With the chosen rank, the LSR model decreases the dimensionality to $2322 \times S + 216$ which enables significantly more efficient classification. LTuR and LTR decrease the dimensionality to $2322 + 216$ and 2322, respectively, but exhibit poorer performance than LSRTR (and even SVM), suggesting that CP and Tucker models cause a loss of representation power that is essential to parameter estimation for some datasets. With the Vessel MNIST dataset we have more training samples than ADHD200 and a lower-dimensional problem (only 21,952). For this reason all methods perform relatively well, with LSRTR having one of the best performances. When compared to ResNet 50, our method approaches its performance in terms of F1 score and beats its performance in terms of average accuracy.

We make a remark on Table 5a. The sensitivity for LSRTR is 1, which is likely due to the fact that we use only 14 test samples from the ABIDE Autism dataset. In order to provide better interpretability, Figure 7 shows a histogram of the posterior probabilities for the 14 test data samples. We see that all of the samples labelled as $y = 1$ are classified as such since their posterior probabilities are all over 0.5. Most samples of Class 1 are classified with high confidence (in the sense that posterior probabilities are well above the 0.5 threshold). Only one sample of Class 0 was mis-classified.

To provide a more comprehensive comparison between LSRTR and other state-of-the-art neural networks, we also compare the performance of LSRTR on vessel MNIST dataset to a benchmark analysis (Yang et al., 2021b). We compare three ResNet 18 methods, with various additional convolution layers (namely 2.5D, $3D$ and $ACS$, or, 'axial-coronal-sagittal'), two additional ResNet 50 methods, with additional convolution layers (namely 2.5D and ACS) (Yang et al., 2021a)), and AutoKeras. The results are shown in Table 6. The results show that, although LSRTR may be outperformed by some complex methods, it exhibits comparable and occasionally superior performance to several others. Additionally, we note that while neural networks often outperform regression models in certain fields like computer vision due to their increased flexibility, their potential overfitting in a small sample regime (limited sample sizes are often associated with medical datasets from clinical studies) remains a concern. Moreover, regression models are frequently termed 'white-box models' owing to the transparency and interpretability of their parameters. This interpretability – an indispensable attribute in numerous applications – allows for a detailed analysis of covariates' statistical significance and the generation of confidence intervals (not just accuracy scores) as posterior probabilities – qualities that are challenging to obtain with neural networks (Dreiseitl & Ohno-Machado, 2002; Issitt et al., 2022). Such confidence probabilities are important information for medical professionals to consider in medical studies, as they provide valuable insights into the classification results. Additionally, unlike the neural network methods, LSRTR is easily extendable beyond the 3D case, and is trivially applied to other regression types (such as linear and Poisson), and does not require fine tuning for each case.

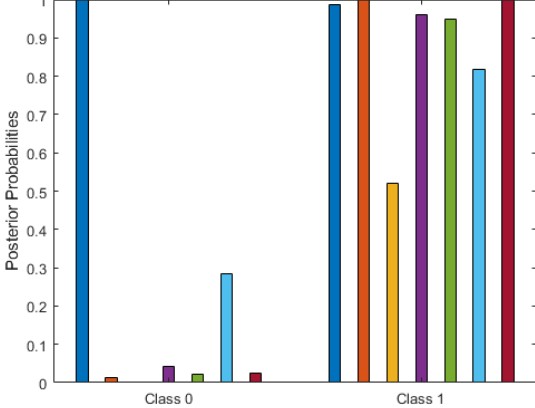

Figure 7: Histogram of posterior probabilities for test data in ABIDE Autism Dataset for the case of LSRTR.

|  | SVM | LR | LCPR | LTuR | LSRTR |
|---|---|---|---|---|---|
| **Sensitivity** | 0.71 | 0.71 | 0.71 | 0.71 | 1 |
| **Specificity** | 0.14 | 0.71 | 0.85 | 0.85 | 0.85 |
| **F1 score** | 0.55 | 0.71 | 0.77 | 0.77 | **0.93** |
| **AUC** | 0.42 | 0.51 | 0.84 | 0.84 | **0.9** |
| **Average Accuracy** | 0.43 | 0.71 | 0.78 | 0.78 | **0.92** |

(a) **ABIDE Autism**

|  | SVM | LR | LCPR | LTuR | LSRTR |
|---|---|---|---|---|---|
| **Sensitivity** | 0.41 | 1 | 0.62 | 0.51 | 0.83 |
| **Specificity** | 0.78 | 0 | 0.34 | 0.52 | 0.4 |
| **F1 score** | 0.5 | 0.62 | 0.35 | 0.48 | **0.65** |
| **AUC** | 0.62 | 0.5 | 0.56 | 0.62 | **0.67** |
| **Average Accuracy** | **0.62** | 0.45 | 0.47 | 0.51 | 0.59 |

(b) **ADHD200**

|  | SVM | LR | LCPR | LTuR | LSRTR | ResNet 50 + 3D |
|---|---|---|---|---|---|---|
| **Sensitivity** | 0.39 | 0.53 | 0.26 | 0.32 | 0.47 | 0.85 |
| **Specificity** | 0.95 | 0.55 | 0.946 | 0.94 | 0.96 | 0.86 |
| **F1 score** | 0.44 | 0.21 | 0.3 | 0.37 | 0.55 | **0.57** |
| **AUC** | 0.84 | 0.52 | 0.6 | 0.66 | 0.81 | **0.9** |
| **Average Accuracy** | 0.89 | 0.55 | 0.869 | 0.87 | **0.91** | 0.85 |

(c) **Vessel MNIST 3D**

Table 5: Comparison of LSRTR with LR, LCPR and LTuR for diagnosis of test subjects in datasets: (*a*)ABIDE Autism (*b*) ADHD200 and (*c*) Vessel MNIST 3D. In the case of Vessel MNIST 3D, LSRTR is also compared to ResNet 50.

|  | RN18+2.5D | RN18+3D | RN18+ACS | RN50+2.5D | RN50+ACS | Autokeras |
|---|---|---|---|---|---|---|
| **AUC** | 0.74 | 0.87 | 0.93 | 0.75 | 0.9 | 0.773 |

Table 6: Neural network benchmark analysis for classification diagnosis of test subjects in dataset Vessel MNIST 3D. RN stands for ResNet.

# 6 Minimax Lower Bounds for Tensor-Structured GLMs

In this section we derive lower bounds on the minimax risk of the LSR-TGLM problem[1]. We numerically assess the tightness of the derived bound in Section 6.1. In the results we will derive, the minimax risk depends explicitly on the parameters of the LSR-structured tensor model, namely $\{r_k\}_{k \in [K]}$, $\{m_k\}_{k \in [K]}$, $S$, the distribution of $\underline{\mathbf{X}}$ and the number of samples $n$. Specifically, we derive lower bounds on $\varepsilon^*$ using an argument for estimation problems based on Fano's inequality as described by Yu (1997). This approach relates the minimax risk of coefficient tensor $\underline{\mathbf{B}}$ to a multiple hypothesis testing problem. It states that if there exists an estimator (let us call it $\widehat{\underline{\mathbf{B}}}(\mathbf{y}, \mathcal{X})$, and which can be any parameter estimation algorithm),

---

[1]Preliminary results appeared in a conference paper taki2021minimax.

with error matching the minimax risk $\varepsilon^*$ (i.e., $\left\|\widehat{\mathbf{B}}(\mathbf{y}, \mathcal{X}) - \underline{\mathbf{B}}\right\|_F^2 = \varepsilon^*$), then this estimator $\widehat{\mathbf{B}}(\mathbf{y}, \mathcal{X})$ can be used to solve a multiple hypothesis testing problem (MHTP) (Khas'minskii, 1979). In this MHTP, the hypothesis set consists of a collection, $\mathcal{B}_L$, of $L$ LSR-structured coefficient tensors, where the goal of the estimator, $\widehat{\mathbf{B}}(\mathbf{y}, \mathcal{X})$, is to detect the correct 'generating' LSR-structured tensor. Fano's inequality provides a fundamental limit/bound on the error probability for the multiple hypothesis testing problem. This limit in turn provides a lower bound on the minimax risk $\varepsilon^*$. As mentioned in Section 3, our analysis is local and our approach to deriving a lower bound on the minimax risk is information-theoretic and thus involves the analysis of the mutual information (defined as $\mathbb{I}(\mathbf{y}; l)$) between observations $\mathbf{y} = [y_1 \ldots y_n]$ and hypothesis $l \in [L]$;

$$\mathbb{I}(\mathbf{y}; l) \triangleq \mathbb{E}_{\mathbf{y}, l}\left[\log \frac{f(\mathbf{y}, l)}{f(\mathbf{y}) f(l)}\right], \tag{37}$$

where $f(\cdot, \cdot)$ and $f(\cdot)$ are joint and marginal probability distribution functions respectively. We are now ready to state our main result.

**Theorem 6.** *Consider the rank-$(r_1, \cdots, r_K)$ and separation rank $S$ LSR-TGLM problem in (13) with $n$ i.i.d observations, $\left\{\underline{\mathbf{X}}_i, y_i\right\}_{i=1}^n$, where $\mathrm{vec}(\underline{\mathbf{X}}_i) \sim \mathcal{N}(\mathbf{0}, \mathbf{\Sigma}_x)$, and the true coefficient tensor $\|\underline{\mathbf{B}}\|_F^2 < d^2$. The minimax risk $\varepsilon^*$ is then lower bounded by*

$$\varepsilon^* \geq \frac{\left(\frac{1}{16}\right) S \sum_{k \in [K]}(m_k - 1)r_k + \prod_{k \in [K]}(r_k - 1) - 1}{128 M \left\|\mathbf{\Sigma}_x\right\|_2 n}, \tag{38}$$

*where $M$ is defined in Assumption 2.*

Theorem 6 can be specialised to the Tucker and CP decomposition regression problems in the existing literature. That is, when $S = 1$ we have the Tucker model, and when $r_1 = r_2 = \cdots = r_K = r$ and $\underline{\mathbf{G}}$ is constructed as a diagonal tensor, we have the CP model. With these specialisations we obtain the following corollaries:

**Corollary 1.** *Consider the rank-$(r_1, \cdots, r_K)$ Tucker-TGLM problem (Li et al., 2018; Zhang et al., 2020; Zhang & Jiang, 2016) with $n$ i.i.d observations, $\left\{\underline{\mathbf{X}}_i, y_i\right\}_{i=1}^n$, where $\mathrm{vec}(\underline{\mathbf{X}}_i) \sim \mathcal{N}(\mathbf{0}, \mathbf{\Sigma}_x)$ and the true coefficient tensor $\|\underline{\mathbf{B}}\|_F^2 < d^2$. The minimax risk for this problem is lower bounded by*

$$\varepsilon^* \geq \frac{\left(\frac{1}{16}\right) \sum_{k \in [K]}(m_k - 1)r_k + \prod_{k \in [K]}(r_k - 1) - 1}{128 M \left\|\mathbf{\Sigma}_x\right\|_2 n}. \tag{39}$$

**Corollary 2.** *Consider the rank-$r$ CP-TGLM problem (Zhou et al., 2013; Tan et al., 2013) with $n$ i.i.d observations, $\left\{\underline{\mathbf{X}}_i, y_i\right\}_{i=1}^n$, where $\mathrm{vec}(\underline{\mathbf{X}}_i) \sim \mathcal{N}(\mathbf{0}, \mathbf{\Sigma}_x)$ and the true coefficient tensor $\|\underline{\mathbf{B}}\|_F^2 < d^2$. The minimax risk for this problem is lower bounded by*

$$\varepsilon^* \geq \frac{\left(\frac{1}{16}\right) \sum_{k \in [K]}(m_k - 1)r + (r - 1) - 1}{128 M \left\|\mathbf{\Sigma}_x\right\|_2 n}. \tag{40}$$

Table 7 provides a summary of the minimax lower bounds from this work and the relevant current literature. Specifically, it shows the order-wise lower bounds on the minimax risk for logistic regression, linear regression and GLMs for several tensor structures on the model parameter. We make three remarks prior to our discussion of Table 7. First, we define the terms $\widetilde{m} = \prod_{k \in [K]} m_k$ and $\widetilde{r} = \prod_{k \in [K]} r_k$. Additionally, $\sigma_y^2$ is the variance of observation $y$, $D$ is a fixed constant and an upper bound on the second derivative of the cumulant function ($a''(\eta)$), and we place a dash ($-$) in the cells where there are no specific results in the existing literature. However, note that our derived minimax bounds for tensor-structured GLMs cover these gaps in the literature (namely the case of CP-structured linear and logistic regression, and Tucker-structured logistic regression). Secondly, the order-wise lower bounds reported for the unstructured case are in fact bounds on the minimax risk for prediction error, rather than estimation error. Nonetheless we include them in our comparison as the sample complexities of estimation and prediction in a given problem tend to be

| Regression | Structure of $\underline{\mathbf{B}}$ | | | |
|:---:|:---:|:---:|:---:|:---:|
| | **Unstructured** | **CP** | **Tucker** | **LSR** |
| **Linear** | $\dfrac{\sigma_y^2 \widetilde{m}}{n}$ 

 (Raskutti et al., 2011) | — | $\dfrac{\sigma_y^2 \left( \displaystyle\sum_{k \in [K]} m_k r_k - r_k^2 + \widetilde{r} \right)}{n}$ 

 (Zhang et al., 2020) | — |
| **Logistic** | $\dfrac{\widetilde{m}}{n}$ 
 (Abramovich & Grinshtein, 2016) | — | — | — |
| **GLM** | $\dfrac{\sigma_y^2 \widetilde{m}}{Dn}$ 

 (Lee & Courtade, 2020) | $\dfrac{\displaystyle\sum_{k \in [K]} m_k r + r}{M \left\| \mathbf{\Sigma}_x \right\|_2 n}$ 

 Corollary 2 | $\dfrac{\displaystyle\sum_{k \in [K]} m_k r_k + \widetilde{r}}{M \left\| \mathbf{\Sigma}_x \right\|_2 n}$ 

 Corollary 1 | $\dfrac{S \displaystyle\sum_{k \in [K]} m_k r_k + \widetilde{r}}{M \left\| \mathbf{\Sigma}_x \right\|_2 n}$ 

 Theorem 6 |

Table 7: Summary of order-wise lower bounds on the minimax risk for linear regression, logistic regression and GLM settings and for various tensor structures (unstructured, CP, Tucker and LSR). For unstructured problems, $\widetilde{m} = \prod_{k \in [K]} m_k$ is the total number of elements of the tensor. For Tucker and LSR models, $\widetilde{r} = \prod_{k \in [K]} r_k$.

proportional. Thirdly, here we have reported lower bounds on the minimax risk for non-sparse regression, as we are not studying sparsity in the scope of this work.

In the first two rows of the table, we report the order-wise minimax lower bounds for linear and logistic regression. In all cells of the first two rows, we see that the minimax risk decreases proportionally with increasing sample size. In the unstructured case, the minimax risk is proportional to the number of learnable parameters in the model ($\prod_k m_k$). For linear regression, however, this only applies if the number of parameters is less than $n$. This condition puts this result at a disadvantage for the high-dimensional setting under study. We can see that the minimax risk of linear regression in the Tucker-structured tensor case is proportional to the number of parameters in the Tucker model ($\sum_k m_k r_k + \prod_k r_k$). The numerators ($\prod_k m_k$) and ($\sum_k m_k r_k + \prod_k r_k$) give insights into the parameters on which an achievable minimax risk might depend, and thus give insights into the sample complexity of the parameter estimation problem. We can see that imposing a low-rank Tucker structure may significantly decrease the number of learnable parameters.

The third row is the row of interest as it summarises results for GLMs. Similar to the previous rows, the minimax risk decreases with an increase in sample size. We also have a dependence on $\sigma_y^2$ and $\frac{1}{D}$. Intuitively speaking, this means that the minimax risk increases with the variance of $y$. In the unstructured case the sample complexity is ($\prod_k m_k$), and does not account for any tensor-structured GLM settings. Theorem 6 and Corollaries 1 and 2 make up the last three columns and address this gap in the literature. Therefore, our proposed minimax lower bound is a unified bound, as it encompasses minimax lower bounds for the Tucker and CP-structured GLM models that were introduced in prior works (Zhou et al., 2013; Li et al., 2018), a feature that has yet to be examined in existing literature. The minimax risk is proportional to the number of parameters for the CP, Tucker and LSR case. This is intuitively pleasing as it suggests that by imposing either the CP, Tucker or LSR structure on $\underline{\mathbf{B}}$, one can significantly reduce the sample complexity from the unstructured case to $\sum_k m_k r + r$, $\sum_k m_k r_k + \prod_k r_k$, and $S \sum_k m_k r_k + \prod_k r_k$, respectively, and that we can develop algorithms (such as LSRTR in Algorithm 1) that meet these bounds. As previously mentioned, the CP model induces the lowest number of parameters and the LSR model has $(S - 1) \sum_k m_k r_k$ more parameters than the Tucker model. Additionally, our results also imply an inverse relationship between the

minimax risk and $\|\mathbf{\Sigma}_x\|_2$ and $M$, which is an intuitively pleasing outcome of our analysis. The variance of the covariates $\mathbf{x}$ (and thus $\|\mathbf{\Sigma}_x\|_2$) symbolises the signal power held by the GLM, in the sense that the signal-to-noise ratio increases with increasing variance of $\mathbf{x}$. To illustrate perspicuously, we can take the specific case of binary logistic regression: In this setting, an increased variance of $\mathbf{x}$ induces a more varied natural parameter, $\eta$, which in turn causes easier distinction between the two response classes. In a similar manner, a smaller variance of $\mathbf{x}$ induces a more difficult classification problem (the classes are harder to distinguish). In the extreme case, a variance of 0 causes all observations to collapse onto a single point, making the classes indistinguishable. Therefore with increased variance of $\mathbf{x}$ (and thus $\|\mathbf{\Sigma}_x\|_2$), the estimation error should decrease. This argument is also consistent with various other minimax lower bounds in the literature (Barnes & Özgür, 2019; Shakeri et al., 2016).

Lastly, though we do not provide any theoretical guarantees for the tightness of our bounds in Theorem 6 and Corollaries 1 and 2, we can see that our minimax bound for the Tucker-GLM meets that of the Tucker linear regression works by Zhang et al. (2020), (we do not consider the $r_k^2$ term in (Zhang et al., 2020) as that only accounts for the non-singular transformation indeterminacy, which we do not discuss in the scope of this work). Zhang et al. (2020) have shown that their bounds are indeed optimal, which suggests that the minimax lower bounds derived in Theorem 6 and Corollaries 1 and 2 are also tight. Moreover, we now provide a numerical analysis to assess the tightness of our bounds.

## 6.1 Tightness of Theorem 6

We utilise the results of our experiments on synthetic data in Section 5 to investigate the tightness of the minimax lower bound in Theorem 6. We have shown that our result matches an existing bound in the literature for the specific case of Tucker linear regression. Now, we perform an empirical assessment of our result. Figure 8 shows the ratio of the mean empirical error (over 50 repetitions) of LSRTR from our experiments in Section 5.1.2 to our obtained lower bound in Theorem 6. We do this for linear and logistic regression and we can see the ratio is approximately constant as a function of sample size for both regression types. Such a result suggests that our bound may be achievable in the sense that we can develop algorithms that meet the minimax lower bound and take advantage of the LSR tensor structure to lower the sample complexity of GLM problems.

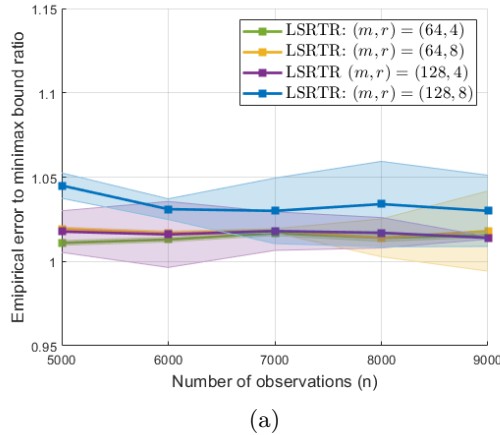
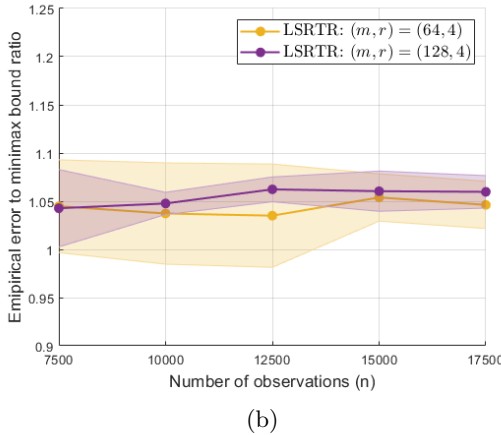

|         (a)          |          (b)          |

Figure 8: Comparison of the empirical error in LSRTR to the minimax lower bound for linear and logistic regression. (*a*) shows the ratio of the empirical error in linear regression to the minimax lower bound for $m = 64, 128$ and $r = 4, 8$. (*b*) shows the ratio of the empirical error in logistic regression to the minimax bound for $m = 64, 128$ and $r = 4$.

## 6.2 Roadmap for Proof of Theorem 6

We begin by introducing a few important concepts. First, in order to set up the multiple hypothesis testing problem consider the constructed packing set

$$\mathcal{B}_L = \{\underline{\mathbf{B}}_l \in \mathcal{P}_{\{r_k\},S} \colon l \in [L]\} \subset \mathcal{B}_d(\mathbf{0}), \ L \in \mathbb{N}, \tag{41}$$

from which the true index $l$ corresponding to the true coefficient tensor is generated uniformly at random. In order to ensure a tight bound on our minimax risk, $\mathcal{B}_L$ must possess the following properties: ($i$) The minimum packing distance between any two distinct hypotheses $\underline{\mathbf{B}}_l, \underline{\mathbf{B}}_{l'} \in \mathcal{B}_L$, $l, l' \in [L]$, $l \neq l'$ is large. Specifically, for a strictly positive parameter $\delta$ we require a construction such that $\|\underline{\mathbf{B}}_l - \underline{\mathbf{B}}_{l'}\|_F \geq \sqrt{8\delta}$. ($ii$) The hypothesis testing problem is hard in the sense that it is difficult to detect the true index $l$ (and thus the true coefficient tensor) based on an observation $y$. ($iii$) The construction of $\mathcal{B}_L$ must induce lower bounds on the minimax risk that reflects the reduction in the sample complexity of the LSR tensor structure, i.e., we require a bound that exhibits a relation between the intrinsic degrees of freedom of the LSR tensor $(\{m_k\}_{k\in[K]}, \{r_k\}_{k\in[K]}, S)$ and the number of samples ($n$). Secondly, since the objective of the multiple hypothesis testing problem is to detect the index $l$, it is solved through the minimum distance decoder

$$\widehat{l}(\mathbf{y}) \triangleq \underset{l'\in[L]}{\arg\min} \left\| \widehat{\underline{\mathbf{B}}} - \underline{\mathbf{B}}_{l'} \right\|_F, \tag{42}$$

where $\widehat{\underline{\mathbf{B}}}$ is estimated through any learning algorithm, such as Algorithm 1 proposed in this work. In the LSR-TGLM problem we have the following hypothesis detection criteria:

- If $\left\| \widehat{\underline{\mathbf{B}}} - \underline{\mathbf{B}}_l \right\|_F \leq \sqrt{2\delta}$: $\mathbb{P}(\widehat{l}(\mathbf{y}) \neq l) = 0$.

- If $\left\| \widehat{\underline{\mathbf{B}}} - \underline{\mathbf{B}}_l \right\|_F > \sqrt{2\delta}$: A detection error **might** occur, $\mathbb{P}(\widehat{l}(\mathbf{y}) \neq l) \geq 0$.

The minimum distance decoder detects the true hypothesis if the estimate $\widehat{\underline{\mathbf{B}}}$ lies within the open ball $\mathcal{B}(\underline{\mathbf{B}}_l, \sqrt{2\delta})$, and a detection error can only occur if $\left\| \widehat{\underline{\mathbf{B}}} - \underline{\mathbf{B}}_l \right\|_F > \sqrt{2\delta}$. Thus the probability of error is bounded by

$$\mathbb{P}(\widehat{l}(\mathbf{y}) \neq l) \leq \mathbb{P}\left( \left\| \widehat{\underline{\mathbf{B}}} - \underline{\mathbf{B}}_l \right\|_F \geq \sqrt{2\delta} \right). \tag{43}$$

In this approach we interpret the coefficient estimation problem as a communication problem. From the set in (41), the source selects the true hypothesis $\underline{\mathbf{B}}_l$ (the true coefficient tensor) uniformly at random. The hypothesis $\underline{\mathbf{B}}_l$ then generates the channel output according to a 'channel model' as in (13) with a chosen link function $g(\cdot)$. Specifically, the channel outputs $y_i$ such that $\mathbb{E}[y_i|\underline{\mathbf{X}}_i] = \mu_i = g^{-1}(\langle \underline{\mathbf{B}}_l, \underline{\mathbf{X}}_i \rangle)$. The minimum distance decoder then successfully recovers the true hypothesis if the estimator $\widehat{\underline{\mathbf{B}}}$ is within a ball of radius $\sqrt{2\delta}$ around $\underline{\mathbf{B}}_l$. Thirdly, we recognise that the problem under study is supervised and the response variables $y_i \ \forall i \in [n]$ are conditioned on input data $\underline{\mathbf{X}}_i \ \forall i \in [n]$. On the basis thereof we opt to evaluate the conditional mutual information $\mathbb{I}(\mathbf{y}; l|\mathcal{X}) \geq \mathbb{I}(\mathbf{y}; l)$.

We use the aforementioned concepts to achieve our result in (38) through the accomplishment of the following tasks: 1) We must construct the set $\mathcal{B}_L$ as an exponentially large (with respect to the dimensions of an LSR-structured tensor) family of $L$ distinct LSR structured tensors, and satisfying the properties described above. This involves constructing $KS + 1$ individual sets (for the $KS$ factor matrices $\{\mathbf{B}_{(k,s)}\}_{k\in[k], \ s\in[S]}$ and core tensor $\underline{\mathbf{G}}$) and deriving conditions under which all sets can exist simultaneously (with high probability). 2) We must find tight bounds on the conditional mutual information $\mathbb{I}(\mathbf{y}; l|\mathcal{X})$. As explained in Section 3, 'Off-the-shelf' packing sets and bounds on the minimax risk for the unstructured (vector-based), low-rank matrix, or low-rank Tucker-structured parameter estimation problems are not useful in our application as these results do not capture the LSR tensor structure and would yield suboptimal lower bounds. Thus designing the right geometric insight into LSR tensors – by constructing a packing that respects the factorisation structure and constraints (e.g., orthogonality) on the factor matrices – is novel, as our goal is to understand

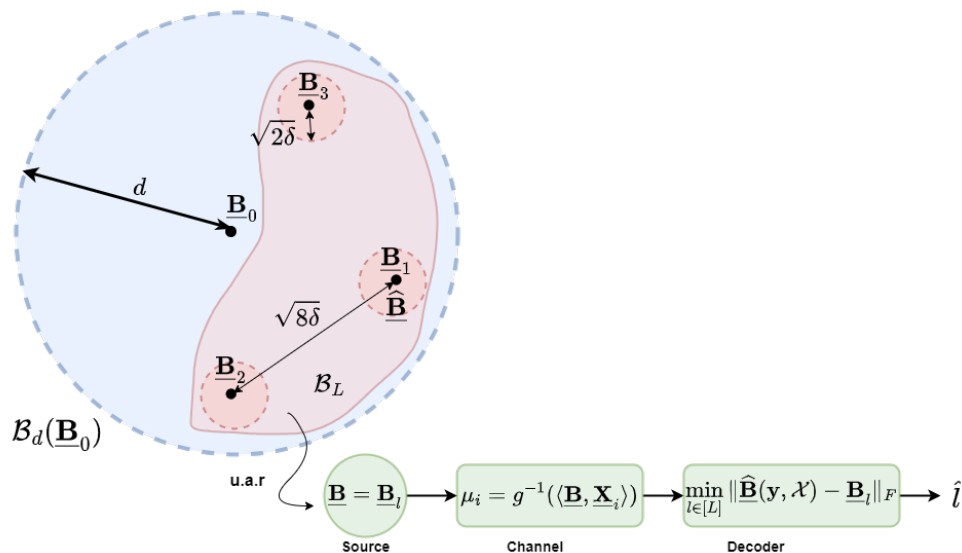

Figure 9: Information theoretic approach for deriving bounds on the minimax risk. Consider a packing set $\mathcal{B}_L = \{\underline{\mathbf{B}}_l : l \in [L]\} \subset \mathcal{B}_d(\mathbf{0})$ with minimum distance between any two elements as $\sqrt{8\delta}$. The true coefficient tensor is selected uniformly at random (u.a.r) and the LSR-TGLM model generates the channel output. We detect the true coefficient tensor using the minimum distance decoder. Here, the minimum distance decoder will detect $\underline{\mathbf{B}}_1$ as the true hypothesis.

the benefits of imposing the LSR structure on the problem's sample complexity. The novelty in our work is that we explicitly leverage this structure leading to a hypothesis set $\mathcal{B}_L$ with a structure that cannot be achieved through current methods. Another challenge in using this technique is actually bounding the relevant information-theoretic quantities. To that end, we derive a new tight bound on the Kullback-Leibler divergence in tensor-structured GLMs. Our bound involves explicitly accounting for the link function with an LSR-structured coefficient tensor. This is a non-trivial derivation, as analysing the link function entails the careful construction of a packing set and the derivation of tight bounds on the packing distance as in Lemma 5. The results in Lemma 5 and the bounds on the KL-divergence can be used in other problems involving LSR tensors and distinguish our work from previous studies. Lastly, though our analysis is local and depends on a fixed tensor in a neighbourhood with known radius, our minimax risk becomes independent of this radius if it becomes sufficiently large.

## 6.3 Proof of Theorem, 6

Four lemmas lay the foundation to the formal proof of Theorem 6. Through Lemmas 4, 5, 6 and 7 we achieve the following: We construct the hypothesis set $\mathcal{B}_L$ containing $L$ distinct tensors. Since each $\underline{\mathbf{B}}_l \in \mathcal{B}_L$ is LSR-structured, as defined in (9), we construct $\mathcal{B}_L$ by constructing $KS + 1$ individual sets that correspond to $KS$ sets for $\mathbf{B}_{(k,s)}$ ($k \in [K]$, $s \in [S]$) and a set for $\mathbf{G}$. Our construction of $\mathcal{B}_L$ results in tight upper and lower bounds on the distance $\|\underline{\mathbf{B}}_l - \underline{\mathbf{B}}_{l'}\|_F^2$ for any two distinct $l, l' \in [L]$, sampled uniformly at random (u.a.r), which will be used to derive bounds on the mutual information $\mathbb{I}(\mathbf{y}; l)$ and minimax risk $\varepsilon^*$. Complete proofs of these lemmas are in the appendix. Lemma 4 (and Corollary 4) introduce the sets of 'generating' vectors (and matrices) from which we will later construct $\mathbf{G}$ (and $\mathbf{B}_{(k,s)}$ for all $k \in [k]$, $s \in [S]$), that are in turn used to construct each $\underline{\mathbf{B}}_l$. Specifically, for some $\alpha \in \mathbb{N}$, our aim is to construct a set of $(\alpha)$-dimensional vectors with entries sampled uniformly at random from $\left\{ \frac{-1}{\sqrt{\alpha}}, \frac{1}{\sqrt{\alpha}} \right\}$, with some designated minimum packing distance in the Hamming metric between any two vectors. Topologically speaking, this can also be viewed as the construction of a subset of an $(\alpha)$-dimensional hypercube with a required minimum Hamming distance between any two distinct elements in the set. Lemma 4 and its subsequent corollary utilise the Gilbert-Varshamov bound on constructing binary codes with minimum distance in order to construct the 'generating'

sets we have just described. The following can be found in the book by Tsybakov (Tsybakov, 2009, Lemma 2.9), where we restate the bound here in the interest of keeping this work self-contained.

**Lemma 3** (Gilbert-Varshamov Bound, Lemma 2.9 (Tsybakov, 2009)). *Let $d \geq 8$. Then there exists a subset $\mathcal{C} \subset \{0,1\}^d$ of size*

$$|\mathcal{C}| \geq 2^{d/8} \tag{44}$$

*such that for any pair $\mathbf{v}, \mathbf{v}' \in \mathcal{C}$, we have $\|\mathbf{v} - \mathbf{v}'\|_1 \geq \frac{d}{8}$.*

**Corollary 3.** *For any $d \geq 8$, there exists a set of binary vectors $\mathcal{C} \subseteq \{-\alpha, \alpha\}^d$ of size $|\mathcal{C}| \geq 2^{d/8}$ such that for any pair $\mathbf{v}, \mathbf{v}' \in \mathcal{C}$, we have $\|\mathbf{v} - \mathbf{v}'\|_0 \geq \frac{d}{8}$.*

Lemma 4 and Corollary 4 introduce a set of 'generating' binary vectors and matrices, respectively, with minimum distance.

**Lemma 4.** *Let $r_k > 0 \ \forall k \in [K]$ and $\widetilde{r} = \prod_{k \in [K]} r_k$. For any integers $(r_1, r_2, \cdots, r_k)$ such that $\widetilde{r} \geq 8$, there exists a collection of $F \geq 2^{(\widetilde{r}-1)/8}$ vectors $\{\mathbf{s}_f \in \frac{1}{\sqrt{\widetilde{r}-1}}\{-1,1\}^{\widetilde{r}-1} : f \in [F]\}$ such that for all $f, f' \in [F]$ we have $\|\mathbf{s}_f - \mathbf{s}_{f'}\|_0 \geq (\widetilde{r}-1)/8$.*

**Corollary 4.** *Let $m_k > 0$ and $r_k > 0 \ \forall k \in [K]$. For any integers $(m_k, r_k)$ such that $(m_k-1)r_k \geq 8$ there exists a collection of $P = 2^{(m_k-1)r_k/8}$ matrices $\{\mathbf{S}_p \in \frac{1}{\sqrt{(m_k-1)r_k}}\{-1,1\}^{(m_k-1)r_k} :$ $p \in [P]\}$ such that for all $p, p' \in [P]$ we have $\|\mathbf{S}_p - \mathbf{S}_{p'}\|_0 \geq (m_k-1)r_k/8$.*

The sets generated in Lemma 4 and Corollary 4 can be used to construct the set $\mathcal{B}_L$ of $L$ tensors in (41). We now introduce Lemma 5, which derives conditions on $L$ such that the sets in Lemma 4 and Corollary 4 exist simultaneously. In Lemma 5 we construct $\mathcal{B}_L$ with a certain set of properties. Note that every element in $\mathcal{B}_L$ must have an LSR structure. That is, every element is comprised of $KS$ low-rank and orthogonal factor matrices (of dimensions $m_k \times r_k$, $k \in [K]$), and a core tensor (of dimensions $r_1 \times \cdots \times r_K$), that we carefully construct from the set of 'generating' matrices and vectors defined in Corollary 4 and Lemma 4, respectively, according to (9). Additionally, any two distinct elements $\underline{\mathbf{B}}_l, \underline{\mathbf{B}}_{l'} \in \mathcal{B}_L$ must be a suitable distance apart. Lemma 5 derives an upper and lower bound on this distance; the lower bound determines the minimum packing distance, while the upper bound is used to derive an upper bound on the conditional mutual information in Lemma 6. The LSR tensor structure induced by having a common core tensor $\underline{\mathbf{G}}$ is critical to such an analysis, specifically in deriving the lower bound in (46), which is a result of the algebraic steps shown between (74) and (75), in the proof of Lemma 5 in the appendix. Not only would these steps be considerably more challenging without the convenience of the LSR matrix structure of (14), but the effect of not achieving tight bounds on the packing distance may ultimately lead to minimax bounds that do not reflect the intrinsic degrees of freedom of the chosen GLM model.

**Lemma 5.** *Define $\widetilde{r} = \prod_{k \in [K]} r_k$. Let $S > 0$, $m_k > 0$, $r_k > 0$, and $(m_k-1)r_k \geq 8 \ \forall k \in [K]$. There exists a collection of $L$ tensors $\mathcal{B}_L \triangleq \{\underline{\mathbf{B}}_l : l \in [L]\} \subset \mathcal{B}_d(\mathbf{0})$ for some $d > 0$ of cardinality*

$$L \geq 2^{\frac{1}{8}\left[S \sum_{k \in [K]} (m_k-1)r_k + (\widetilde{r}-1)\right]} \tag{45}$$

*such that for any*

$$\frac{1}{S}\sqrt{\frac{32(\widetilde{r}-1)}{\widetilde{r}}} < \varepsilon \leq \frac{d}{S}\sqrt{\frac{\widetilde{r}-1}{\widetilde{r}}}, \tag{46}$$

*we have*

$$S^2 \frac{\widetilde{r}}{\widetilde{r}-1}\varepsilon^2 \leq \|\underline{\mathbf{B}}_l - \underline{\mathbf{B}}_{l'}\|_F^2 \leq 4S^2 \frac{\widetilde{r}}{\widetilde{r}-1}\varepsilon^2. \tag{47}$$

The bounds in (47) match (up to a constant) and are tight, which help ensure that the upper bound on the conditional mutual information is also tight. Until now, we have completed the tasks of constructing

the set $\mathcal{B}_L$ with the packing distance $8\delta = \frac{S^2}{4} \frac{\widetilde{r}}{r-1} \varepsilon^2$ and with packing number (cardinality) parameterised by the parameters $\{r_k\}_{k \in [K]}, \{m_k\}_{k \in [K]}, S$. We transition to the next task of deriving upper bounds on the conditional mutual information, $\mathbb{I}(\mathbf{y}; l | \mathcal{X})$. Upper bounds on $\mathbb{I}(\mathbf{y}; l | \mathcal{X})$ involve the evaluation of the Kullback-Leibler (KL) divergence. For this we employ well-established results in the literature (Cover & Thomas, 2012; Jung et al., 2016). Specifically, define $f_l(\mathbf{y}|\mathcal{X})$ and $f_{l'}(\mathbf{y}|\mathcal{X})$ as the conditional probability distribution of $\mathbf{y}$ given $\mathcal{X}$ with any two distinct coefficient tensors $\underline{\mathbf{B}}_l$ and $\underline{\mathbf{B}}_{l'}$ respectively. Denote $D_{KL}$ as the relative entropy, then

$$\mathbb{I}(\mathbf{y}; l | \mathcal{X}) \leq \frac{1}{L^2} \sum_{l,l'} \mathbb{E}_{\mathcal{X}} D_{KL}(f_l(\mathbf{y}|\mathcal{X}) || f_{l'}(\mathbf{y}|\mathcal{X})). \tag{48}$$

We note here that we require the hypothesis test to be hard, which corresponds to a relatively small KL divergence. Therefore, our aim is to derive a tight upper bound on the KL divergence in (48), by relying on the tight bounds in (47). Lemma 6 thus derives an upper bound on $\mathbb{I}(\mathbf{y}; l | \mathcal{X})$.

**Lemma 6.** *Consider the LSR-TGLM problem given by the model in* (13) *such that* $\underline{\mathbf{B}} \in \mathcal{B}_d(\mathbf{0})$ *for some* $d > 0$, *and consider the set* $\mathcal{B}_L$ *constructed in Lemma* 5. *Consider* $n$ *i.i.d observations* $y_i$ *following a probability distribution belonging to the exponential family when conditioned on* $vec(\underline{\mathbf{X}}_i) \sim \mathcal{N}(\mathbf{0}, \Sigma_x)$. *Then we have:*

$$\mathbb{I}(\mathbf{y}; l | \mathcal{X}) \leq 4 S^2 M n \|\Sigma_x\|_2 \frac{\widetilde{r}}{\widetilde{r}-1} \varepsilon^2. \tag{49}$$

The steps between (77) and (80) in the proof of Lemma 6 in the appendix elucidate how the analysis of the KL-divergence involves both the link function and the bounds appearing in (47). Specifically, the KL-divergence between two distributions $f_l, f_{l'}$ in the exponential family is a function of the link functions $\eta_l = \langle \underline{\mathbf{B}}_l, \underline{\mathbf{X}} \rangle$ and $\eta_{l'} = \langle \underline{\mathbf{B}}_{l'}, \underline{\mathbf{X}} \rangle$. Additionally, a tight upper bound on the KL-divergence requires a tight upper bound on $\eta_l - \eta_{l'}$, or alternatively, $\|\underline{\mathbf{B}}_l - \underline{\mathbf{B}}_{l'}\|_F$. Since any $\underline{\mathbf{B}}_l$ is LSR-structured, we require the result in (47). The final step is to lower bound $\mathbb{I}(\mathbf{y}; l | \mathcal{X})$ using Fano's inequality (Yu, 1997), stated below:

$$\mathbb{I}(\mathbf{y}; l | \mathcal{X}) \geq \mathbb{I}(\mathbf{y}; l) \geq \left(1 - \mathbb{P}(\widehat{l}(\mathbf{y}) \neq l)\right) \log_2(L) - 1. \tag{50}$$

Evaluating (50) is simple. For this we refer the reader back to (43) in Section 6.2, which bounds the error probability $\mathbb{P}(\widehat{l}(\mathbf{y}) \neq l)$ for the recovery of hypothesis $\underline{\mathbf{B}}_l$ using the minimum distance decoder. Lemma 7 bounds this probability, with proof that follows exactly that for Lemma 8 by Jung et al. (2016), and thus is omitted in this work.

**Lemma 7.** *Consider the minimum distance decoder in* (42). *Consider also the LSR-TGLM regression problem in* (13) *with minimax risk* $\varepsilon^*$. *Suppose* $\varepsilon^* \leq \delta$ *for some non-negative scalar* $\delta$. *If the* $L$ *tensors distributed in Lemma* 5 *satisfy* $\|\underline{\mathbf{B}} - \underline{\mathbf{B}}_{l'}\| \geq 8\delta$ *then the detection error of the minimum distance decoder is upper bounded by* $\mathbb{P}(\widehat{l}(\mathbf{y}) \neq l) \leq \frac{1}{2}$.

*Proof of Theorem 6.* Fix $(r_1, r_2, \ldots, r_K)$. From Lemma 5 for any $\varepsilon > 0$ satisfying (46) there exists a packing set $\mathcal{B}_L \subset \mathcal{B}_d(\mathbf{0})$ with cardinality $L$ and packing distance $\|\underline{\mathbf{B}}_l - \underline{\mathbf{B}}_{l'}\|_F^2$ satisfying (45) and (47), respectively. Also, from Lemma 6, with the set $\mathcal{B}_L$, $\mathbb{I}(\mathbf{y}; l | \mathcal{X})$ satisfies (49). Now suppose there exists an estimator $\widehat{\underline{\mathbf{B}}}$ which guarantees a risk $\varepsilon^* = \frac{S^2 r}{32(r-1)} \varepsilon^2$. If we set $8\delta = \frac{S^2}{4} \frac{\widetilde{r}}{r-1} \varepsilon^2$ then from Lemma 7 for the set $\mathcal{B}_L$ we have $\mathbb{P}(\widehat{l}(\mathbf{y}) \neq l) \leq \frac{1}{2}$. Combining Lemmas 6 and 7 we have

$$\frac{1}{2} \log_2 L - 1 \leq \mathbb{I}(\mathbf{y}; l | \mathcal{X}) \leq 128 M n \|\Sigma_x\|_2 \varepsilon^*, \tag{51}$$

which can be rearranged to achieve the result in (6). $\qquad \square$

# 7 Conclusion

In this work, we investigate the LSR model on tensor-structured GLM problems. Specifically, we imposed a low-rank LSR structure on the coefficient tensor in GLMs. The parameter estimation problem is highly non-convex, and we propose a block coordinate descent algorithm, called LSRTR, for this purpose. Each convex sub-problem in LSRTR estimates a separate element/component of the LSR-structured coefficient tensor. In our theoretical analysis, we provide a minimax lower bound on the estimation error of the parameter estimation problem of LSR-structured tensors and specialise these bounds for Tucker-structured and CP-structured tensors. These bounds show that the LSR structure reduces the sample complexity compared to that of the vector case. We evaluate the tightness of these bounds numerically and through the comparison of the specific case of Tucker regression to tight bounds in the literature. The methods we use may be of interest to readers as they can also be utilised for deriving minimax bounds on other LSR-structured estimation problems. Furthermore, we evaluate the LSRTR algorithm and the LSR-model on several synthetic and real datasets. These experiments demonstrate that the LSR model is less restrictive than other tensor models (such as Tucker or CP), and can be effectively employed for analysis on balanced and imbalanced medical imaging data. Some possible future work include theoretical analysis of the tightness of the minimax bounds, and theoretical guarantees of the LSRTR algorithm.

**Acknowledgments**

This work was supported by the US National Science Foundation under awards CCF-1910110, CCF-1907658 and OAC-1940074, the US National Institutes of Health under award 2R01DA040487, and the Army Research Office under award W911NF-21-1-0301.

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

# A    Supporting Results

## A.1    Proofs

*Proof of Corollary 3.* By Lemma 3 there exists a packing with minimum distance $\frac{d}{8}$ in the $\ell_1$ norm containing $2^{d/8}$ binary vectors. Mapping $0 \to -\alpha$ and $1 \to \alpha$ for all vectors in this packing shows that there exists a packing of at least $2^{d/8}$ vectors in $\{-\alpha, \alpha\}^d$ with minimum distance $\frac{d\alpha}{4}$ in the $\ell_1$ norm. For any pair $\mathbf{v}, \mathbf{v}'$ in that packing we have $\|\mathbf{v} - \mathbf{v}'\|_0 \geq \frac{d}{8}$, since every entry of $\mathbf{v} - \mathbf{v}'$ is in $\{-2\alpha, 0, 2\alpha\}$. $\qquad\square$

*Proof of Lemma 4.* This is a direct consequence of Corollary 3. $\qquad\square$

The proof of Lemma 5 contains the derivation of a tight upper bound on $\|\underline{\mathbf{B}}_l - \underline{\mathbf{B}}_{l'}\|_F^2$. The following lemma is a component needed to achieve such an upper bound; specifically, we will see that it proves useful in deriving an upper bound on $\|\underline{\mathbf{B}}_l\|_F^2$, for any $l \in [L]$.

**Lemma 8.** *Let $\mathbf{x} \in \mathbb{R}^m$ be any $m$-dimensional real vector and $\mathbf{O} \in \mathbb{O}^{m \times m}$ be any orthogonal basis of $\mathbb{R}^m$. Define $u_i = |\cos \theta_i|$, where $\theta_i$ is the angle between any possible $\mathbf{x}$ and basis vector $\mathbf{o}_i$. The function $f = \sum_{i=1}^{m} (u_i + 1)^2$ is minimized when $u_i = \frac{-1}{\sqrt{m}}$, or when $\mathbf{x}$ is equiangular to all basis vectors $\mathbf{o}_i, \forall i \in [m]$.*

*Proof of Lemma 8.* Consider the function $f = \sum_{i=1}^{m} (u_i + 1)^2$. Additionally, for a basis $\mathbf{O} \in \mathbb{O}^{m \times m}$ and $x \in R^m$, $\sum_{i=1}^{m} (\cos \theta_i)^2 = 1$, thus we have the equality constraint $g = \sum_{i=1}^{m} u_i^2 - 1 = 0$. Denote $\lambda$ as the Lagrange multiplier, and thus the Lagrange function is defined as

$$L_g = \sum_{i=1}^{m} (u_i + 1)^2 - \lambda (\sum_{i=1}^{m} u_i^2 - 1). \tag{52}$$

The partial derivative of $L_g$ with respect to $u_i, \forall i \in [m]$ is

$$\frac{\partial L_g}{\partial u_i} = 2(u_i + 1) - 2\lambda u_i, \ \forall i \in [m]. \tag{53}$$

The partial derivative of $L_g$ with respect to $\lambda$ is

$$\frac{\partial L_g}{\partial \lambda} = -\sum_{i=1}^{m} u_i^2 + 1. \tag{54}$$

By setting (53) and (54) to zero we get

$$u_i = \frac{-1}{1 - \lambda} \ \forall i \in [m], \tag{55}$$

and

$$-\sum_{i=1}^{m} u_i^2 + 1 = 0, \tag{56}$$

respectively. What we have in (55) and (56) is a system of $m + 1$ equations and $m + 1$ unknowns. We solve the system to find the critical points of $f(\cdot)$, which are $u_i = \frac{-1}{\sqrt{m}}$ and $u_i = \frac{1}{\sqrt{m}}$. Since $u_i \geq 0$, the solution is $u_i = \frac{1}{\sqrt{m}}, \forall i \in [m]$ minimizes the function $f(\cdot)$. $\square$

*Proof of Lemma 5.* Fix the following arbitrary real orthonormal bases: $\mathbf{Q}$ of $\mathbb{R}^{\widetilde{r}}$, and $K$ sets of $r_k$ bases, $\left\{ \mathbf{U}_{k,j} \right\}_{j=1}^{r_k}$ of $\mathbb{R}^{m_k}, \forall k \in [K]$.

Next, consider the following hypercubes or subsets thereof: 1) The set of $F$ vectors $\{\mathbf{s}_f\}$ from Lemma 4:

$$\mathbf{s}_f \in \left\{ \frac{-1}{\sqrt{\widetilde{r} - 1}}, \frac{+1}{\sqrt{\widetilde{r} - 1}} \right\}^{\widetilde{r} - 1}, \tag{57}$$

where $f \in [F]$, and 2) $KS$ sets of $P_{(k,s)}$ matrices $\forall k \in [K], \forall s \in [S]$, from Lemma 4:

$$\mathbf{S}_{p_{(k,s)}} \in \left\{ \frac{-1}{\sqrt{(m_k - 1)r_k}}, \frac{+1}{\sqrt{(m_k - 1)r_k}} \right\}^{(m_k - 1) \times r_k} \forall k \in [K], \ s \in [S], \tag{58}$$

where $p_{(k,s)} \in [P_{(k,s)}]$.

We proceed with the following steps in order to construct the final set $\mathcal{B}_L$ of coefficient tensors from the sets in (57) and (58). Since $\mathcal{B}_L \subset \mathcal{B}_d(\mathbf{0})$, we know that the energy of any $\underline{\mathbf{B}}_l$ is upper bounded by $d^2$. We will construct $\underline{\mathbf{G}}_f$, and matrices with orthonormal columns, namely $\mathbf{B}_{p_{(k,s)}} \forall k \in [K], \ s \in [S]$, all of which will be used to construct every $\underline{\mathbf{B}}_l \in \mathcal{B}_L$. Specifically, due to our LSR model, any tensor $\underline{\mathbf{B}}_l$ will have a rank $(r_1, r_2, \ldots, r_K)$ LSR structure.

We use the notation $(f, i)$ to denote the $i^{th}$ step in constructing the $f^{th}$ element of $\underline{\mathbf{G}}_f$. Hence in the first step, we construct vectors $\mathbf{g}_{(f,1)} \in \mathbb{R}^{\widetilde{r}}$ for $f \in [F]$, using $\mathbf{Q}$ and $\mathbf{s}_f$, as follows:

$$\mathbf{g}_{(f,1)} = \mathbf{Q} \begin{bmatrix} \sqrt{\frac{1}{\sqrt{\widetilde{r} - 1}}} \\ \mathbf{s}_f \end{bmatrix}, \forall f \in [F]. \tag{59}$$

From (59), since $\|\mathbf{s}_f\|_2^2 = 1$ we have:

$$\left\|\mathbf{g}_{(f,1)}\right\|_2^2 = \left\|\mathbf{Q}\begin{bmatrix}\sqrt{\frac{1}{\widetilde{r}-1}}\\ \mathbf{s}_f\end{bmatrix}\right\|_2^2 = \frac{\widetilde{r}}{\widetilde{r}-1}.$$

Now, we define each vector $\mathbf{g}_f$ as

$$\mathbf{g}_f = \frac{\varepsilon}{\sqrt{\widetilde{r}}}\mathbf{g}_{(f,1)}, \forall f \in [F], \tag{60}$$

for some positive number $\varepsilon$.

Similarly, we use the notation $(p_{(k,s)}, i)$ to denote the $i^{th}$ step in constructing the $p_{(k,s)}^{th}$ element of $\mathbf{B}_{p_{(k,s)}}$. Hence, we construct matrices $\mathbf{B}_{(p_{(k,s)},1)} \in \mathbb{R}^{m_k \times r_k}$, for $p_{(k,s)} \in [P_{(k,s)}], \forall k \in [K], s \in [S]$. Define $\mathbf{B}_{(p_{(k,s)},1)}^{(j)}$ as the $j^{th}$ column of $\mathbf{B}_{(p_{(k,s)},1)}, \forall k \in [K], s \in [S]$, and $\mathbf{S}_{p_{(k,s)}}^{(j)}$ as the $j^{th}$ column of $\mathbf{S}_{p_{(k,s)}}$. Let the columns be constructed as follows:

$$\mathbf{B}_{(p_{(k,s)},1)}^{(j)} = \mathbf{U}_{k,j}\begin{bmatrix}1\\ \mathbf{S}_{p_{(k,s)}}^{(j)}\end{bmatrix}, \forall p_{(k,s)} \in [P_{(k,s)}], k \in [K], s \in [S]. \tag{61}$$

From (61) we have:

$$\left\|\mathbf{B}_{(p_{(k,s)},1)})^{(j)}\right\|_2^2 = = \frac{r_k+1}{r_k} \ \forall k \in [K].$$

We now construct matrices $\mathbf{B}_{p_{(k,s)}} \in \mathbb{R}^{m_k \times r_k}$, for $p_{(k,s)} \in [P_{(k,s)}], k \in [K], s \in [S]$. The construction of each $\mathbf{B}_{p_{(k,s)}}$ follows the same procedure for all $k \in [K]$ and $s \in [S]$. Define $\mathbf{B}_{p_{(k,s)}}^{(j)} \in \mathbb{R}^{m_k}$ as the $j^{th}$ column of $\mathbf{B}_{p_{(k,s)}}$, for $j \in [r_k]$. We set

$$\mathbf{B}_{p_{(k,s)}}^{(1)} = \frac{\mathbf{B}_{(p_{(k,s)},1)}^{(1)}}{\left\|\mathbf{B}_{(p_{(k,s)},1)}^{(1)}\right\|_2}, \tag{62}$$

and define

$$\mathbf{a}^{(j+1)} \triangleq \mathbf{B}_{(p_{(k,s)},1)}^{(j+1)} - \sum_{j'=1}^{j}\langle \mathbf{B}_{(p_{(k,s)},1)}^{(j+1)}, \mathbf{B}_{p_{(k,s)}}^{(j')}\rangle \mathbf{B}_{p_{(k,s)}}^{(j')}, \tag{63}$$

and

$$\mathbf{B}_{p_{(k,s)}}^{(j+1)} \triangleq \frac{\mathbf{a}^{j+1}}{\|\mathbf{a}^{j+1}\|_2}. \tag{64}$$

The steps in (62), (63) and (64) constitute the well-known Gram-Schmidt process. Thus, the set of vectors $\mathbf{B}_{(p_{(k,s)},1)}^{(j)}$, for $j \in [r_k], p_{(k,s)} \in [P_{(k,s)}], k \in [K]$ and $s \in [S]$ are orthonormal, i.e., $\left\|\mathbf{B}_{p_{(k,s)}}^{(j)}\right\|_2^2 = 1$ and $\mathbf{B}_{p_{(k,s)}}^{(j)} \perp \mathbf{B}_{p_{(k,s)}}^{(j')}$, for any two distinct $j, j' \in [r_k]$. Consequently, $\left(\mathbf{B}_{p_{(k,s)}}\right)^T\left(\mathbf{B}_{p_{(k,s)}}\right) = \mathbf{I}_{r_k}$. Now by defining the set

$$\mathcal{L} \triangleq \left\{(f, (p_{(k,s)})_{k \in [K], \ s \in [S]}) : f \in [F], p_{(k,s)} \in [P_{(k,s)}], k \in [K], s \in [S]\right\} \tag{65}$$

as the set of all tuples, $(f, p_{(1,1)}, \ldots, p_{(1,S)}, \ldots, p_{(K,1)}, \ldots, p_{(K,S)})$, we have

$$L = |\mathcal{L}| \overset{(a)}{\geq} 2^{(1/8)\left[(\widetilde{r}-1)+S\sum_{k=1}^{K}(m_k-1)r_k\right]}, \tag{66}$$

where $(a)$ follows from Lemma 4 and Corollary 4. We define the set of coefficient tensors, $\mathcal{B}_L$ as,

$$\mathcal{B}_L \triangleq \left\{ \underline{\mathbf{B}}_l = \sum_{s=1}^{S} \underline{\mathbf{G}}_f \times_1 \mathbf{B}_{p_{(1,s)}} \times_2 \mathbf{B}_{p_{(2,s)}} \cdots \times_K \mathbf{B}_{p_{(K,s)}} : l \in [L], f \in [F], p_{(k,s)} \in [P_{(k,s)}], k \in [K], s \in [S] \right\},$$

(67)

and we restrict $\varepsilon$ such that

$$\frac{1}{S}\sqrt{\frac{32(\widetilde{r}-1)}{\widetilde{r}}} < \varepsilon < \frac{d}{S}\sqrt{\frac{\widetilde{r}-1}{\widetilde{r}}}.$$

(68)

We make the final note that, due to the Kronecker product, we can express $\mathrm{vec}(\underline{\mathbf{B}}_l)$ as:

$$\mathrm{vec}(\underline{\mathbf{B}}_l) = \sum_{s=1}^{S} (\mathbf{B}_{p_{(K,s)}} \otimes \mathbf{B}_{p_{(K-1,s)}} \otimes \cdots \otimes \mathbf{B}_{p_{(1,s)}}) \mathbf{g}_f.$$

(69)

We have the following remaining tasks at hand: 1) We must show that the energy of any $\underline{\mathbf{B}}_l$ is less than $d^2$. 2) We must derive an upper and lower bound on the distance $\left( \|\underline{\mathbf{B}}_l - \underline{\mathbf{B}}_{l'}\|_F^2 \right)$ between any two distinct tensors $\underline{\mathbf{B}}_l, \underline{\mathbf{B}}_{l'}, \in B_L$. We begin by showing $\|\underline{\mathbf{B}}_l\|_F^2 < d^2$:

$$\|\underline{\mathbf{B}}_l\|_F^2 = \left\| \sum_{s=1}^{S} \underline{\mathbf{G}}_f \times_1 \mathbf{B}_{p_{(1,s)}} \times_2 \mathbf{B}_{p_{(2,s)}} \times_3 \cdots \times_K \mathbf{B}_{p_{(K,s)}} \right\|_F^2$$

$$= \left\| \sum_{s=1}^{S} \left( \mathbf{B}_{p_{(K,s)}} \otimes \mathbf{B}_{p_{(K-1,s)}} \otimes \cdots \otimes \mathbf{B}_{p_{(1,s)}} \right) \mathbf{g}_f \right\|_2^2$$

$$\overset{(b)}{\leq} \left\| \sum_{s=1}^{S} \mathbf{B}_{p_{(K,s)}} \otimes \cdots \otimes \mathbf{B}_{p_{(1,s)}} \right\|_F^2 \|\mathbf{g}_f\|_2^2 \tag{70}$$

$$\overset{(c)}{\leq} \left( \sum_{s=1}^{S} \left\| \mathbf{B}_{p_{(K,s)}} \otimes \cdots \otimes \mathbf{B}_{p_{(1,s)}} \right\|_F \right)^2 \|\mathbf{g}_f\|_2^2 \tag{71}$$

$$\overset{(d)}{=} S^2 \prod_k \|\mathbf{B}_{k,p_k}\|_F^2 \|\mathbf{g}_f\|_2^2 = \frac{S^2 \widetilde{r} \varepsilon^2}{\widetilde{r}-1} \overset{(e)}{<} d^2, \tag{72}$$

where $(b)$ follows from the fact that for any matrix $\mathbf{A}$ and any vector $\mathbf{a}$, $\|\mathbf{A}\mathbf{a}\|_2 \leq \|\mathbf{A}\|_2 \|\mathbf{a}\|_2$ and the fact that $\|\mathbf{A}\|_2 \leq \|\mathbf{A}\|_F$ (Petersen & Pedersen, 2012). Additionally, $(c)$ follows the triangle inequality and $(d)$ from the fact that the matrix norm of the Kronecker product is the product of the matrix norms. Additionally, $(e)$ holds due to (68).

We proceed with deriving lower and upper bounds on $\|\underline{\mathbf{B}}_l - \underline{\mathbf{B}}_{l'}\|_F^2$ for any two distinct $\underline{\mathbf{B}}_l, \underline{\mathbf{B}}_{l'} \in B_L$. We first denote the square matrix $\widetilde{\mathbf{B}}_{p_{(k,s)}} \in \mathbb{R}^{m_k \times m_k}$ as the completed orthonormal matrix of each low-rank matrix $\mathbf{B}_{p_{(k,s)}} \in \mathbb{R}^{m_k \times r_k}$. Also, $\widetilde{\underline{\mathbf{G}}}_f \in \mathbb{R}^{m_1 \times \cdots \times m_K}$ has entries $\widetilde{\underline{\mathbf{G}}}_f(\cdot)$ defined as follows:

$$\begin{cases} \widetilde{\underline{\mathbf{G}}}_f(1:r_1, \ldots, 1:r_K) = \underline{\mathbf{G}}_f(1:r_1, \ldots, 1:r_K) \\ \widetilde{\underline{\mathbf{G}}}_f(r_1+1:m_1, \ldots, r_K+1:m_K) = \underline{\mathbf{G}}_f(1:r_1, \ldots, 1:r_K). \end{cases} \tag{73}$$

Also define $\widetilde{\underline{\mathbf{B}}}_l = \sum_{s=1}^{S} \widetilde{\underline{\mathbf{G}}}_f \times_1 \widetilde{\mathbf{B}}_{p_{(1,s)}} \times_2 \cdots \times_K \widetilde{\mathbf{B}}_{p_{(K,s)}}$ for any $l \in [L]$. With these definitions, we have the equality $\|\underline{\mathbf{B}}_l - \underline{\mathbf{B}}_{l'}\|_F^2 = \left\| \widetilde{\underline{\mathbf{B}}}_l - \widetilde{\underline{\mathbf{B}}}_{l'} \right\|_F^2$. Defining $\bigotimes_{k=K}^{k=1} \widetilde{\mathbf{B}}_{p_{(k,s)}} \triangleq \widetilde{\mathbf{B}}_{p_{(K,s)}} \otimes \cdots \otimes \widetilde{\mathbf{B}}_{p_{(1,s)}}$ for any $l \in [L]$ and we

have the following:

$$\|\mathbf{B}_l - \mathbf{B}_{l'}\|_F^2 = \frac{\varepsilon^2}{\widetilde{r}} \left\| \sum_{s=1}^{S} \left( \bigotimes_{k=K}^{k=1} \widetilde{\mathbf{B}}_{p_{(k,s)}} \right) \widetilde{\mathbf{g}}_{(f,1)} - \sum_{s=1}^{S} \left( \bigotimes_{k=K}^{k=1} \widetilde{\mathbf{B}}_{p'_{(k,s)}} \right) \widetilde{\mathbf{g}}_{(f',1)} \right\|_2^2$$

$$= \frac{\varepsilon^2}{\widetilde{r}} \left( \left\| \sum_{s=1}^{S} \left( \bigotimes_{k=K}^{k=1} \widetilde{\mathbf{B}}_{p_{(k,s)}} \right) \widetilde{\mathbf{g}}_{(f,1)} \right\|_2^2 + \left\| \sum_{s=1}^{S} \left( \bigotimes_{k=K}^{k=1} \widetilde{\mathbf{B}}_{p'_{(k,s)}} \right) \widetilde{\mathbf{g}}_{(f',1)} \right\|_2^2 \right.$$

$$\left. -2 \left\langle \sum_{s=1}^{S} \left( \bigotimes_{k=K}^{k=1} \widetilde{\mathbf{B}}_{p_{(k,s)}} \right) \widetilde{\mathbf{g}}_{(f,1)}, \sum_{s=1}^{S} \left( \bigotimes_{k=K}^{k=1} \widetilde{\mathbf{B}}_{p'_{(k,s)}} \right) \widetilde{\mathbf{g}}_{(f',1)} \right\rangle \right).$$

Define $\mathbf{T}_s = \bigotimes_{k=K}^{k=1} \widetilde{\mathbf{B}}_{p_{(k,s)}}$, $\mathbf{V}_s = \bigotimes_{k=K}^{k=1} \widetilde{\mathbf{B}}_{p'_{(k,s)}}$ for any $s \in [S]$, and $\mathbf{T}_s^{(j)}$, $\mathbf{V}_s^{(j)}$ as the $j^{th}$ column of $\mathbf{T}_s$ and $\mathbf{V}_s$, respectively, then we have

$$\|\mathbf{B}_l - \mathbf{B}_{l'}\|_F^2 = \frac{\varepsilon^2}{\widetilde{r}} \left( \left\| \sum_{s=1}^{S} \mathbf{T}_s \widetilde{\mathbf{g}}_{(f,1)} \right\|_2^2 + \left\| \sum_{s=1}^{S} \mathbf{V}_s \widetilde{\mathbf{g}}_{(f',1)} \right\|_2^2 - 2 \left\langle \sum_{s=1}^{S} \mathbf{T}_s \widetilde{\mathbf{g}}_{(f,1)}, \sum_{s=1}^{S} \mathbf{V}_s \mathbf{g}_{(f',1)} \right\rangle \right)$$

$$= \frac{\varepsilon^2}{\widetilde{r}} \left( \left( \sum_{s=1}^{S} \mathbf{T}_s^{T(1)} \widetilde{\mathbf{g}}_{(f,1)} \right)^2 + \cdots + \left( \sum_{s=1}^{S} \mathbf{T}_s^{T(m)} \widetilde{\mathbf{g}}_{(f,1)} \right)^2 + \left( \sum_{s=1}^{S} \mathbf{V}_s^{T(1)} \widetilde{\mathbf{g}}_{(f',1)} \right)^2 + \cdots + \left( \sum_{s=1}^{S} \mathbf{V}_s^{T(m)} \widetilde{\mathbf{g}}_{(f',1)} \right)^2 \right.$$

$$\left. -2 \left( \left( \sum_{s=1}^{S} \mathbf{T}_1^{T(1)} \widetilde{\mathbf{g}}_{(f,1)} \right) \left( \sum_{s=1}^{S} \mathbf{V}_s^{T(1)} \widetilde{\mathbf{g}}_{(f',1)} \right) + \cdots + \left( \sum_{s=1}^{S} \mathbf{T}_s^{T(m)} \widetilde{\mathbf{g}}_{(f,1)} \right) \left( \sum_{s=1}^{S} \mathbf{V}_s^{T(m)} \widetilde{\mathbf{g}}_{(f',1)} \right) \right) \right)$$

We group every $\left( \sum_{s=1}^{S} \mathbf{T}_s^{T(j)} \widetilde{\mathbf{g}}_{(f,1)} \right)^2 + \left( \sum_{s=1}^{S} \mathbf{V}_s^{T(j)} \widetilde{\mathbf{g}}_{(f',1)} \right)^2 - 2 \left( \sum_{s=1}^{S} \mathbf{T}_1^{T(j)} \widetilde{\mathbf{g}}_{(f,1)} \right) \left( \sum_{s=1}^{S} \mathbf{V}_s^{T(j)} \widetilde{\mathbf{g}}_{(f',1)} \right)$, for $j \in [m]$. We get

$$\|\mathbf{B}_l - \mathbf{B}_{l'}\|_F^2 \geq \frac{\varepsilon^2}{\widetilde{r}} \sum_{i=1}^{m} \left( \left| \sum_{s=1}^{S} \mathbf{T}_s^{T(i)} \widetilde{\mathbf{g}}_{(f,1)} \right| - \left| \sum_{s=1}^{S} \mathbf{V}_s^{T(i)} \widetilde{\mathbf{g}}_{(f',1)} \right| \right)^2. \tag{74}$$

The expression in (74) contains inner products. Specifically, $\left| \sum_{s=1}^{S} \mathbf{T}_s^{T(i)} \widetilde{\mathbf{g}}_{(f,1)} \right| = \left| \left\langle \sum_{s=1}^{S} \mathbf{T}_s^{T(i)}, \widetilde{\mathbf{g}}_{(f,1)} \right\rangle \right|$. Denote $\lambda_i = \left| \cos \angle \left( \sum_{s=1}^{S} \mathbf{T}_s^{T(i)}, \widetilde{\mathbf{g}}_{(f,1)} \right) \right|$, then we have

$$\|\mathbf{B}_l - \mathbf{B}_{l'}\|_F^2 \overset{(f)}{\geq} \frac{\varepsilon^2}{\widetilde{r}} \sum_{i=1}^{m} \left( \lambda_i \left\| \sum_{s=1}^{S} \mathbf{T}_s^{T(i)} \right\|_2 \|\widetilde{\mathbf{g}}_{(f,1)}\|_2 - \left\| \sum_{s=1}^{S} \mathbf{V}_s^{T(i)} \right\|_2 \|\widetilde{\mathbf{g}}_{(f',1)}\|_2 \right)^2$$

$$\overset{(g)}{=} \frac{\widetilde{r}\varepsilon^2}{\widetilde{r}(\widetilde{r}-1)} \sum_{i=1}^{m} \left( \lambda_i \left\| \sum_{s=1}^{S} \mathbf{T}_s^{T(i)} \right\| - \left\| \sum_{s=1}^{S} \mathbf{V}_s^{T(i)} \right\| \right)^2$$

$$\overset{(h)}{\geq} \frac{\widetilde{r}\varepsilon^2}{\widetilde{r}(\widetilde{r}-1)} \sum_{i=1}^{m} S^2 (\lambda_i + 1)^2$$

$$\overset{(i)}{\geq} \frac{\widetilde{r}\varepsilon^2}{\widetilde{r}(\widetilde{r}-1)} \sum_{i=1}^{m} S^2 (1 + \frac{1}{\sqrt{m}})^2 \geq \frac{S^2 \widetilde{r}}{\widetilde{r}-1} \varepsilon^2, \tag{75}$$

where $(f)$ is due to applying Cauchy-Schwartz inequality to $\left| \sum_{s=1}^{S} \mathbf{V}_s^{T(i)} \widetilde{\mathbf{g}}_{(f',1)} \right|$, $(g)$ is due to (60), $(h)$ is due to the fact that $\left\| \sum_{s=1}^{S} \mathbf{T}_s^{T(i)} \right\|$ and $\left\| \sum_{s=1}^{S} \mathbf{T}_s^{V(i)} \right\|$ are lower and upper bounded by $-S$ and $S$, respectively,

and $(i)$ is from the result in Lemma 8. Finally, for finding upper bounds on $\|\mathbf{\underline{B}}_l - \mathbf{\underline{B}}_{l'}\|_F^2$, we have:

$$
\begin{aligned}
&\|\mathbf{\underline{B}}_l - \mathbf{\underline{B}}_{l'}\|_F^2 \\
&\stackrel{(j)}{\leq} \left( \left\| \sum_{s=1}^{S} \left( \mathbf{B}_{p_{(K,s)}} \otimes \cdots \otimes \mathbf{B}_{p_{(1,s)}} \right) \mathbf{g}_f \right\|_F + \left\| \sum_{s=1}^{S} \left( \mathbf{B}_{p'_{(K,s)}} \otimes \cdots \otimes \mathbf{B}_{p'_{(1,s)}} \right) \mathbf{g}_{f'} \right\|_F \right)^2 \\
&\leq \left( \sum_{s=1}^{S} \prod_k \left\| \mathbf{B}_{p_{(k,s)}} \right\|_F \left\| \mathbf{g}_f \right\|_2 + \sum_{s=1}^{S} \prod_k \left\| \mathbf{B}_{p'_{(k,s)}} \right\|_F \left\| \mathbf{g}_{f'} \right\|_2 \right)^2 \\
&\stackrel{(k)}{=} \left( 2 \sum_{s=1}^{S} \prod_k \left\| \mathbf{B}_{p_{(k,s)}} \right\|_F \left\| \mathbf{g}_f \right\|_2 \right)^2 \\
&= \frac{4S^2 \widetilde{r}}{\widetilde{r} - 1} \varepsilon^2,
\end{aligned}
\tag{76}
$$

where $(j)$ follows from the triangle inequality, and $(k)$ follows from the fact that $\left\| \mathbf{B}_{p_{(k,s)}} \right\|_F = \left\| \mathbf{B}_{p'_{(k,s)}} \right\|_F$ and that $\|\mathbf{g}_f\|_2 = \|\mathbf{g}_{f'}\|_2$. $\qquad\square$

*Proof of Lemma 6.* Consider the set $\mathcal{B}_L$ from Lemma 5, where the bounds in (75) and (76) hold. For the LSR-TGLM model in (13), consider $n$ i.i.d samples, with covariate tensors $\mathbf{\underline{X}}_i \in \mathbb{R}^{m_1 \times \cdots \times m_K}, \forall i \in [n]$, where $\text{vec}(\mathbf{\underline{X}}_i) \sim \mathcal{N}(\mathbf{0}, \mathbf{\Sigma}_x)$. According to (13), observations $y_i$ follow an exponential family distribution when conditioned on $\mathbf{\underline{X}}_i$, $\forall i \in [n]$. Consider the vector of $n$ observations, $\mathbf{y}$, and the tensor of $n$ samples, $\mathcal{X}$. Define also $\mathbb{I}(\mathbf{y}; l|\mathcal{X})$ as the mutual information between observations $\mathbf{y}$ and index $l$ conditioned on side-information $\mathcal{X}$. From (Cover & Thomas, 2012; Wainwright, 2009), we have,

$$
\mathbb{I}(\mathbf{y}; l|\mathcal{X}) \leq \frac{1}{L^2} \sum_{l,l'} \mathbb{E}_{\mathcal{X}} D_{KL}(f_l(\mathbf{y}|\mathcal{X}) || f_{l'}(\mathbf{y}|\mathcal{X}),
\tag{77}
$$

where $D_{KL}(f_l(\mathbf{y}|\mathcal{X}) || f_{l'}(\mathbf{y}|\mathcal{X})$ is the Kullback-Leibler (KL) divergence of probability distribution $f_l(\mathbf{y}|\mathcal{X})$ and $f_{l'}(\mathbf{y}|\mathcal{X})$ of $\mathbf{y}$ given $\mathcal{X}$ for some $\mathbf{\underline{B}}_l, \mathbf{\underline{B}}_{l'} \in \mathcal{B}_L$. Denote $\eta_{l_i}$ and $\eta_{l'_i}$ as the link functions associated with $f_l(y_i|\mathbf{\underline{X}}_i)$ and $f_{l'}(y_i|\mathbf{\underline{X}}_i)$, respectively. Also denote $\mu_{l_i}$ as the expectation of sufficient statistic $T(y_i)$ conditioned on $\mathbf{\underline{X}}_i$ under model $\mathbf{\underline{B}}_l$ (otherwise known as the canonical parameter). We evaluate the KL divergence which is given as follows (Nielsen, 2022):

$$
D_{KL}(f_l(\mathbf{y}|\mathcal{X}) || f_{l'}(\mathbf{y}|\mathcal{X})) = \sum_{i \in [n]} (\eta_{l_i} - \eta_{l'_i}) \mu_l - a(\eta_{l_i}) + a(\eta_{l'_i}).
\tag{78}
$$

Now, we take the expectation of (78) with respect to the side-information $\mathcal{X}$. We have $\mathbb{E}_{\mathbf{\underline{X}}}[(\eta_{l_i} - \eta_{l'_i})\mu_l - a(\eta_{l_i}) + a(\eta_{l'_i})] = \mathbb{E}_{\mathbf{\underline{X}}}[(\eta_{l_i} - \eta_{l'_i})\mu_l]$, due to the fact that $\mathbb{E}_{\mathbf{\underline{X}}}[a(\eta_{l_i})] = \mathbb{E}_{\mathbf{\underline{X}}}[a(\eta_{l'_i})]$. We now have:

$$
\begin{aligned}
\mathbb{E}_{\mathbf{\underline{X}}} D_{KL}(f_l(\mathbf{y}|\mathcal{X}) || f_{l'}(\mathbf{y}|\mathcal{X}) &= \sum_{i \in [n]} \mathbb{E}_{\mathbf{\underline{X}}} \left[ (\langle \mathbf{\underline{B}}_l, \mathbf{\underline{X}}_i \rangle - \langle \mathbf{\underline{B}}_{l'}, \mathbf{\underline{X}}_i \rangle) \mathbb{E}[T(y_i)|\mathbf{\underline{X}}_i, l] \right] \\
&\stackrel{(l)}{=} \sum_{i \in [n]} \mathbb{E}_{\mathbf{\underline{X}}} \left[ (\langle \mathbf{\underline{B}}_l - \mathbf{\underline{B}}_{l'}, \mathbf{\underline{X}}_i \rangle) \frac{\partial a(\eta_{l_i})}{\partial \eta_{l_i}} \right] \\
&\leq \sum_{i \in [n]} \sqrt{\mathbb{E}_{\mathbf{\underline{X}}} [\langle \mathbf{\underline{B}}_l - \mathbf{\underline{B}}_{l'}, \mathbf{\underline{X}}_i \rangle] \mathbb{E}_{\mathbf{\underline{X}}} \left[ \frac{\partial a(\eta_{l_i})}{\partial \eta_{l_i}} \right]^2 } \\
&\leq n \|\Sigma_x\|_2 \|\mathbf{\underline{B}}_l - \mathbf{\underline{B}}_{l'}\|_F^2 M,
\end{aligned}
\tag{79}
\tag{80}
$$

where $(l)$ follows from the fact that $\mu_{l_i} = \mathbb{E}[T(y_i)|\mathbf{\underline{X}}_i, l] = \frac{\partial a(\eta_{l_i})}{\partial \eta_{l_i}}$. We achieve (79) through Cauchy-Schwartz inequality. We make some remarks regarding (80): First, we replace the summation over $n$ samples with $n$

since each sample $\underline{\mathbf{X}}_i$ is independent. Secondly, Assumption 2 allows us to bound $\frac{\partial a(\eta_{l_i})}{\partial \eta_{l_i}}$ with $M$. Thirdly, the conditions on $\varepsilon$ in (68) mean $\|\underline{\mathbf{B}}_l - \underline{\mathbf{B}}_{l'}\|_F > 1$ thus $\|\underline{\mathbf{B}}_l - \underline{\mathbf{B}}_{l'}\|_F^2 > \|\underline{\mathbf{B}}_l - \underline{\mathbf{B}}_{l'}\|_F$. Plugging in (80) into (77) gives us

$$\mathbb{I}(\mathbf{y}; l | \mathcal{X}) \le n \|\mathbf{\Sigma}_x\|_2 \|\mathbf{B}_l - \mathbf{B}_{l'}\|_F^2 M \overset{(m)}{\le} 4S^2 Mn \|\Sigma_x\|_2 \frac{\widetilde{r}}{\widetilde{r} - 1} \varepsilon^2,$$

where $(m)$ follows from (76). $\qquad\square$

## A.2 Numerical Results for Poisson Regression

Figure 10 reports the estimation and prediction accuracy for Poisson regression from the experiments on synthetic data discussed in Section 5.1. The shaded regions depict one standard deviation of mean estimation and prediction accuracy, based on 50 replications.

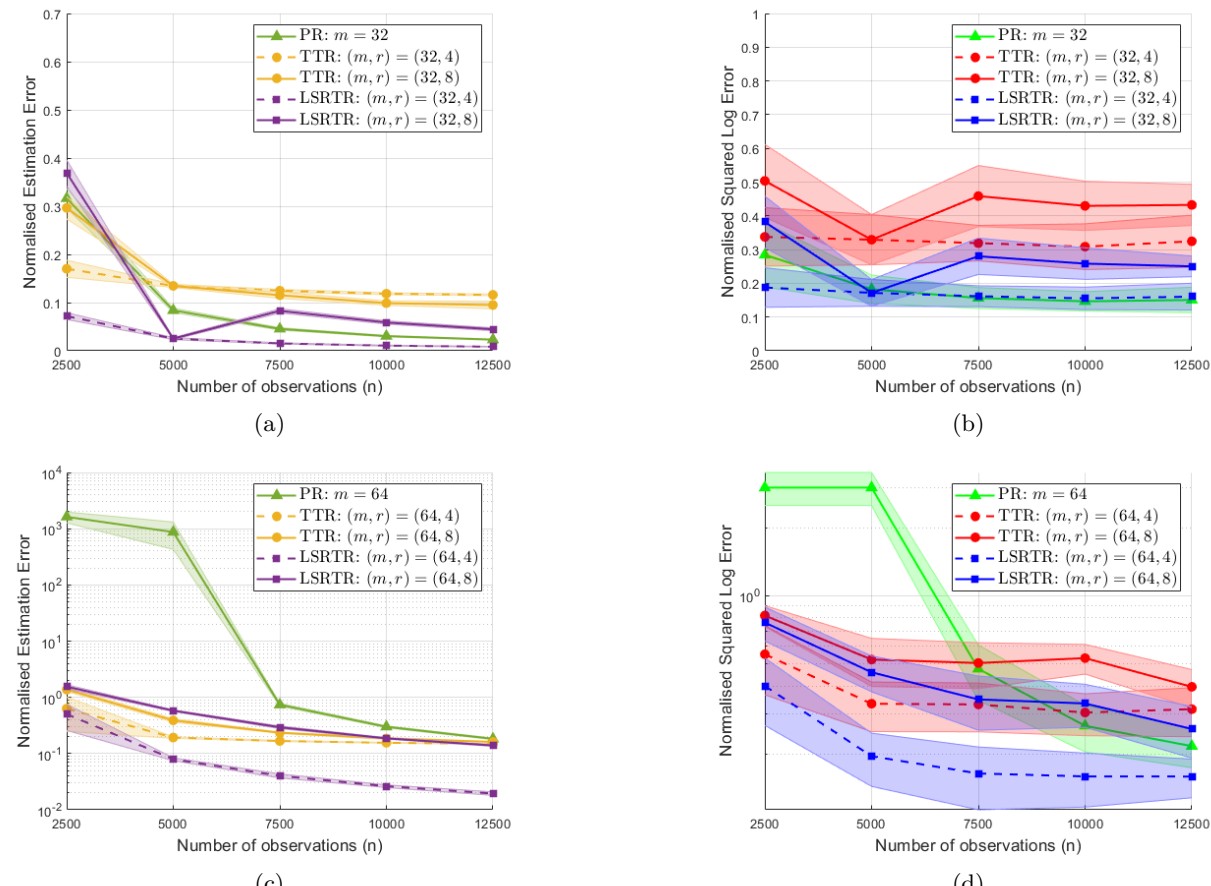

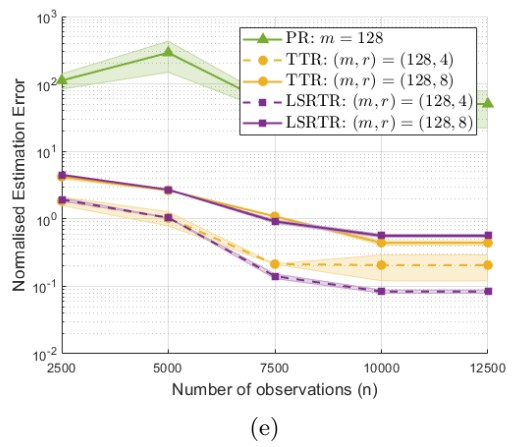

(e)

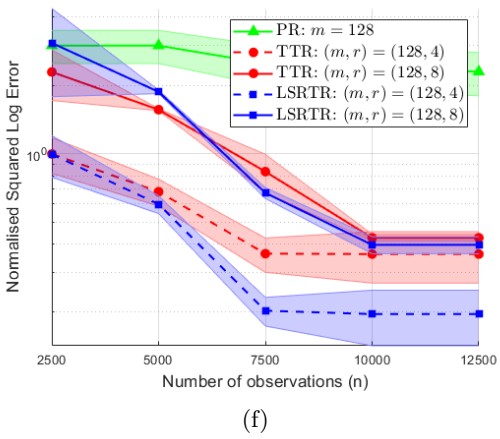

(f)

Figure 10: Comparison of LSRTR with PR and TTR for two-dimensional synthetic data when $m \in \{32, 64, 128\}$, $r \in \{4, 8\}$ and $S = 2$. Normalised estimation error for $m = 32$, 64, and 128 is shown in $(a)$, $(c)$, and $(e)$, respectively. Normalised prediction error for $m = 32$, 64, and 128 is shown in $(b)$, $(d)$, and $(f)$, respectively. Each marker represents the mean normalised estimation/prediction errors, over 50 repetitions. The shaded regions correspond to one standard deviation of the mean normalised errors.

