# OpenReview forum: "Structured Low-Rank Tensors for Generalized Linear Models"
_TMLR — Accepted by TMLR_

### Review · Reviewer_zqEm · 2023-05-10

**Summary Of Contributions:**

In this paper, the authors impose the Low Separation Rank (LSR) model onto the coefficient tensor in the Generalized Linear Model (GLM) problem. They further apply a block coordinate descent algorithm to estimate parameters. In addition, they derive a minimax lower bound on the error threshold on estimating the coefficient tensor in LSR tensor GLM problems. Extensive experiments show the superiority of the LSR model.

Overall, the LSR model is shown to be one effective and efficient representation in the GLM problem.


**Audience:**

Yes

**Claims And Evidence:**

Yes

**Requested Changes:**

1.In the LSR model described by equation (9), the use of different matrices $\underline{G}_{s}$ is not implemented. One might wonder if including such matrices would enhance the model's generalization capability.

2.When using a block coordinate descent algorithm for the tensor model, the initialization should be close to the ground truth. What method do you use to choose initialization? Furthermore, how do you ensure, theoretically, that the initialization is close to the true values?

3.To ensure the orthogonal structure of each matrix $\bf{B}$, the authors employ QR decomposition. It is advisable for the authors to demonstrate that the QR step in the algorithm is non-expansive. Otherwise, concerns may arise regarding potential error amplification following the QR decomposition.

4.The authors mentioned that the minimax risk increases with the variance of $y$. Intuitively, for the linear regression, when we increase the variance of input $x$, the variance of output $y$ will also be improved, further increasing the minimax risk. However, there exist $||\Sigma_x ||_2$ in the denominator of (38). This means that the minimax risk is decreasing with increasing variance of $x$.

5.Why is the normalized squared error corresponding to (m,r) = (256,8) in Figure 2(e) and (f) greater than that of 1?

6.The reviewer suggests that the authors should give the meaning of $M$ in the theorem 6.

7.The reviewer suggests that the authors should simplify the introduction of the experiment and keep necessary experimental results to increase readability. In addition, sections 5 and 7 should be combined.

8.There are some typos: (1) in the beginning of Problem Statement, the dimension of $\underline{X}$ and $\underline{B}$ should be $m_1\times \cdots \times m_k$; (2) in the description of equations (24)-(26), (24) should be (23); (3) in the Algorithm 2, there are extra square brackets in $\hat{\mu}$ and $\underline{\mathcal{X}}$; (4) in Theorem 8, "destributed" should be "distributed".; (5) The equation (79) should be aligned.


**Strengths And Weaknesses:**

Strength:  The analysis of generalized linear model problem is comprehensive.

Weakness: There should be a brief analysis of the proposed algorithm in the paper. In addition, the experimental section in the paper is excessively lengthy.

---

> ### Author Response · Authors · 2023-06-19
> **Response to Reviewer zqEm: Part I**
>
> We thank the reviewer for their careful reading of our manuscript and valuable feedback. In the following we first explain how we plan on addressing the reviewer's feedback as well as the requested changes in the revised version of the manuscript. We then address the general feedback from the reviewer.  **Please note that this response is being broken into three parts because of the limit of 5000 characters in a comment.**
>
> **Revision addressing requested changes**
>
> **_(Q1) Core tensor $G_s$:_** We appreciate the reviewer's remark regarding the model's possible enhanced generalizability through the use of different core tensors, $G_s$. In fact, the LSR tensor decomposition can be viewed as a special case of the Block Tensor Decomposition (BTD) introduced in (“Decompositions of a Higher-Order Tensor in Block Terms”, L De Lathauwer). Having different core tensors $G_s$ would also yield a tensor with BTD structure.  A more comprehensive discussion of BTD is included in the response to reviewer 75Jv. In the revised manuscript, we will shed light onto this as an interesting research direction and elaborate further in order to highlight the motivation for preserving a common core tensor $G$ in our LSR tensor model. More specifically, we will call attention to the following points. The "Low Separation Rank" (LSR) **matrix** structure as introduced and extended in the referenced works  (Tsiligkaridis & Hero, 2013) and (Ghassemi et al., 2019), respectively, can be described as a sum of $S$ Kronecker-structured matrices, or alternatively, a Kronecker-structured matrix with separation rank $S$. In the LSR-TGLM setting, this described matrix structure appears when the coefficient tensor $B$ is vectorized, as depicted in equations (9) and (10) of our work. Though the difference may seem nuanced, a common core tensor $G$ is critical in maintaining this structure, since different $G_s$ does not allow for summing the Kronecker-structured factor matrices. It is this very LSR structure that lends several conveniences: an algebraic convenience in the derivation of the minimax lower bound, an algorithmic convenience in possible extensions of our work, and an intuitive reasoning pertaining to the role of the core tensor $G$ in the LSR-structured tensor.
>
> * **As for the algebraic convenience**, the analysis of the minimax lower bound in our work requires deriving tight bounds on the distance between any two distinct elements of the constructed packing set, as exhibited in equation (46) of Lemma 6. The LSR tensor structure induced by having a common core tensor $G$ is critical to such an analysis, specifically in deriving the lower bound in equation (46), which is a result of the steps shown between equations (73) and (74), in the proof of Lemma 6. Not only would these steps be considerably more challenging without the convenience of the LSR matrix structure of equation (10), as described above, but the effect of not achieving tight bounds on the packing distance may ultimately lead to minimax bounds that do not reflect the intrinsic degrees of freedom of the chosen GLM model.
>
> * **As for the algorithmic convenience**, by keeping a common core tensor $G$ and maintaining the LSR-matrix structure, one may explore and build other parameter estimation algorithms for the LSR-TGLM problem that make explicit use of said matrix structure. Though Block Coordinate Descent algorithms are popular amongst tensor-structured regression works and have been shown to be effective in practice, these algorithms iterate over the tensor factors and do not explicitly exploit the tensors' Kronecker structure that appears upon vectorization. That being said, there do exist efficient algorithms in the literature, such as that in referenced work (Ghassemi et al., 2019), that exploit the separation rank matrix structure. Keeping the same $G$ serves as an impetus for developing similar algorithms.
>
>  * **As for the intuitive reasoning** behind keeping a common core tensor, if one was to view $G$ as a "weight tensor", the same $G$ implies that the $S$ groups of factor matrices must be given the same weight.
>
> **--- response continued in Part II ---**

---

> > ### Author Response · Authors · 2023-06-19
> > **Response to Reviewer zqEm: Part II**
> >
> > **--- continuation of the response from Part I ---**
> >
> > **_(Q2) Block coordinate descent algorithm:_**  The constraint set in LSRTR is a product space of Stiefel manifolds and neither closed nor convex. This makes a convergence analysis for BCD a challenge since we cannot apply general results for the convergence of BCD as in ("Non-linear Programming", 1999). It also presents challenges for analyzing projections, in particular when using the QR decomposition (see response to Q3 below). We will augment Section 4.3 (Final Algorithm: LSRTR) with a brief discussion of the practical and theoretical convergence of the proposed BCD algorithm, and the initialization of the factor matrices and the core tensor. More specifically, we will emphasize that the parameter estimation problem in LSR-TGLMs is non-convex and thus we expect LSRTR to converge to a local minimum. As with many non-convex problems, guaranteeing an initialization close to the true tensor is impractical and many works provide "local convergence guarantees" whose validity is contingent on a "close" initialization to a stationary point. In our work, the initialization of LSRTR is a random initialization on the constraint set. That is, we initialize with an LSR structured tensor (with $K\times S$ orthogonal factor matrices, randomly generated on the Stiefel manifold, and a core tensor with random entries). Therefore, not only will the addition in Section 4.3 draw from the above discussion, we will also supplement Section 5 (Numerical Study) with two sets of experiments to better understand the **practical** behaviour of our algorithm. The first will depict the convergence behaviour of LSRTR, i.e., the reduction of the normalized error as a function of increasing iterations. The second will depict the magnitude of the computed gradient at the final iteration of every subproblem in LSRTR i.e., the convergence to a stationary point. We hope that this discussion will enrich the current manuscript and provide interested readers a more comprehensive understanding of the performance of the proposed algorithm.
> >
> > **_(Q3) Non-expansiveness of the QR factorization:_** The reviewer raises an important point regarding the non-expansiveness of the QR projection step. In many works, such as ("Optimization Algorithms on Matrix Manifolds", 2008) we see that the QR decomposition is a common tool for projecting onto and/or optimizing over the Stiefel manifold. It can be shown that the QR step behaves like a "Restricted Isometry": the **distance between the QR projection of any two points** $A$ and $B$ is upper and lower bounded by **the distance between the same points,** $A$ and $B$ ($+/-$ a known "epsilon"). However, more work is needed to prove strict non-expansiveness of QR projection.
> >
> > We will address the non-expansiveness query in both the empirical and theoretical discussions in the paper. **As for the empirical discussion**, we see from the numerical study in Section 5 (Numerical Analysis) that the QR decomposition as a projection step works well in practice. That being said, we will also augment Section 5 by referring to the discussion above regarding the BCD algorithm and the additional set of experiments that will be added to the manuscript to depict the convergence behaviour of LSRTR. These results will show that the per-iteration normalized error decreases (i.e., the error does not amplify, and the QR step does not prohibit LSRTR from finding a "good estimate" of the underlying coefficient tensor). **As for the theoretical discussion**, we will discuss in Section 4.3 (Final Algorithm: LSRTR) the challenges of showing non-expansiveness of QR projection in the general sense, since the general method of showing the non-expansiveness of a projection assumes a closed and convex constraint set, (“Convex Optimization Theory”, 2008) – which the Stiefel manifold is not .
> >
> > Proving stronger guarantees on optimization over Stiefel manifold is a larger and non-trivial problem. We feel it is better to focus on the empirical properties of LSRTR in this paper and provide in future work a more comprehensive theoretical analysis.
> >
> > **_(Q4) Denominator of the minimax lower bound:_** We appreciate the reviewer pointing out this aspect of our minimax lower bound. We will use this to enrich the discussion of the theoretical results in the manuscript. In particular, we will add a paragraph to Section 6 (Minimax Lower Bound for Tensor Structured GLMs) discussing the intuitively pleasing nature of the derived bound. The addition to the manuscript will be a variation of the language of the following discussion:
> >
> > **--- response continued in Part III ---**

---

> > > ### Author Response · Authors · 2023-06-19
> > > **Response to Reviewer zqEm: Part III**
> > >
> > > **--- continuation of the response from Part II ---**
> > >
> > > The variance of the covariates $X$ symbolizes the signal power held by the GLM, in the sense that the signal-to-noise ratio increases with increasing variance of $X$. To illustrate perspicuously, we can take the specific case of binary logistic regression (BLR). In BLR, an increased variance of $X$ induces a more varied natural parameter, $\eta$, which in turn causes easier distinction between the two response classes. In a similar manner, a smaller variance of $X$ induces a more difficult classification problem (the classes are harder to distinguish). In the extreme case, a variance of $0$ causes all observations to collapse onto a single point, making the classes indistinguishable. Therefore with increased variance of $X$, the estimation error should **decrease**.  This argument is also consistent with various other minimax lower bounds in the literature such as referenced work (Barnes and Ozgur, 2019) and ("Minimax Lower Bounds on Dictionary Learning for Tensor Data", 2016).
> > >
> > > **_(Q5) Normalized Estimation and Prediction Error:_** We will include a paragraph in Section 5.1.1 providing a more meaningful and comprehensive discussion of the normalized errors that exceed $1$. Indeed, the normalized error compares the performance of an estimator relative to the "trivial estimator" which always outputs $0$. A normalized error of $1$ occurs when the estimate is $0$, yet an algorithm can potentially perform *worse* than the trivial estimator allowing an error greater than $1$. Therefore, while we agree with the reviewer that a normalized error greater than 1 is insubstantial and must be disregarded when analyzing the performance of models and algorithms, such an error is not "incorrect" in the practical sense. Regarding the particular figures pointed out by the reviewer (fig. 2(e) and 2(f) for $r=8$), we clarify that in these specific cases, LSRTR is operating in the under-sampling regime, where the model size is very large and the number of available samples $(9000)$ is just about the number of learnable parameters of the model. In such a setting we expect the estimation error to be large and to decrease with increasing number of samples.
> > >
> > > **_(Q6,7,8) Other Edits:_** We appreciate the reviewer's careful reading and identification of the various typographical errors which will be addressed in the final version of the manuscript. Additionally, we will incorporate the reviewer's suggestion to merge Sections 5 (Numerical Analysis) and 7 (Experiments on Medical Imaging Data) for readability. Regarding the simplification of the introduction of Section 5, we believe that the transparency and details of the discussed experiments is essential for method reproducibility. We hope that the various readers who may wish to refer to our work would appreciate the clarity and detail of the discussed experiment objectives, experiment design, chosen variables and error measures. However, we will transfer some experimental results (such as fig. 4) to the appendix for interested readers.
> > >
> > > **Revision addressing general comments**
> > >
> > > _Length of paper:_ As mentioned above, we will move the Poisson experimental section (fig.4) to the appendix to enhance the readability of our manuscript and avoid redundancy.
> > >
> > > *LSRTR algorithm:* In addition to the discussions regarding (Q2) and (Q3) above, we will also augment Section 4.3 (Final Algorithm: LSRTR) with a brief analysis on the per-iteration computational complexity of each sub-problem of the proposed algorithm.

---

> > > > ### Comment · Reviewer_zqEm · 2023-06-19
> > > > **To authors' feedback**
> > > >
> > > > I am very grateful for the detailed explanation provided by the authors. Although this model may not be very innovative, the derived minimax lower bound is highly significant. Therefore, this work is still worth recommending. I am also very much looking forward to the presentation of the experiments in the revised version.

---

> > > > > ### Author Response · Authors · 2023-06-28
> > > > > **Revision Submitted**
> > > > >
> > > > > Thank you; we have submitted the revised manuscript. The requested discussion on the convergence of the algorithm and the non-expansiveness of the QR projection is in Section 5.1.4, and  a more complete discussion on the derived minimax risk is in Section 6. We have also provided a discussion on the BTD and LSR decomposition, as mentioned in our response letter.

---

### Review · Reviewer_sup2 · 2023-05-12

**Summary Of Contributions:**

This paper considers the Low Separation Rank (LSR) tensor models in Generalized Linear Models (GLM) by proposing a block coordinate descent algorithm for parameter estimation in LSR-structured tensor GLMs. It also derives a minimax lower bound on the error threshold on estimating the coefficient tensor in LSR tensor GLM problems. Numerical experiments on both synthetic and real data show the effectiveness of the proposed algorithm.

**Audience:**

Yes

**Claims And Evidence:**

Yes

**Requested Changes:**

Q1: It is suggested to highlight the conceptual novelty of the proposed LSR tensor models.

Q2: It is also suggested to discuss the detailed differences and difficulties in deriving the minimax lower bound in comparison with results for other tensor low-rank models.

Q3: I am more interested in the estimation error of an iterative solution in the algorithm, i.e., the optimization error. Can you give some ideas about how to control/bound it?

Q4: This is a general issue of tensor-based ML models. The low-rank tensor models may be interesting in statistical analysis. However, the empirical performance of general tensor-based ML models is not comparable to the DNNs in most settings. Although this paper gives some hopeful results on real, imbalanced and small-size data, I still appreciate it if the authors give more honest and calm discussions on the effectiveness of tensor-based ML models especially on the following points:

Q4.1) Can tensor-based models be comparable or better than SOTA DNNs in popular classification tasks?

Q4.2) If not, are there any settings where tensor-based learning is a must or very beneficial to the empirical performance?

**Strengths And Weaknesses:**

[Strengths]

S1: A new tensor GLM model based on the LSR structure is proposed.

S2: The derived minimax lower bound is new.

S3: The performance of the proposed model is promising.

[Weaknesses]

W1: (Conceptual novelty). The notion of LSR tensor is not new and the combination of GLM and low-rank tensor/matrix structure has already been well studied.  Thus, the prosed model seems of little conceptual novelty.

W2: (Technical novelty). The derived minimax lower bound seems a somewhat straightforward extension of the analysis for CP and Tucker models. It is unclear what are the technical difficulty in deriving this bound provided the well developed analysis for other low-rank tensor models.

---

> ### Author Response · Authors · 2023-06-13
> **Response to Reviewer sup2: Part I**
>
> We thank the reviewer for their careful reading of our manuscript and valuable feedback. In the following we first comment on the perceived weaknesses listed in the review. We then explain how we plan on addressing the reviewer's feedback as well as the requested changes in the revised version of the manuscript. **Please note that this response is being broken into two parts because of the limit of 5000 characters in a comment.**
>
> **Revision addressing the general feedback**
>
> **_1. Conceptual novelty of the paper:_** We appreciate the reviewer's point regarding the existing literature on tensors (specifically low-rank tensor decompositions) and the combination of Generalized Linear Models (GLMs) with low-rank tensor/matrix structures. However, we would like to emphasize that the conceptual novelty of our work resides not in introducing these concepts but in our unique application of the LSR decomposition to the GLM model and our comprehensive analysis of these constructions.
>
> We have already identified and discussed the main prior works on tensor-structured GLMs by (Zhou et al., 2013) and (Li et al., 2018) in the introduction of the manuscript. These studies fall short of providing a sample complexity analysis. This analysis, which we include in our work, is instrumental in revealing the potential benefits of implementing tensor structure within the GLM context. We show that the LSR model uses a tractable number of parameters to provide an expressive representation of tensor data.
>
> In further support of our paper's novelty, we direct the reviewer to Table 3 in Section 6 of our manuscript. This table provides a summary of existing minimax bounds for various structured and unstructured minimax lower bounds in the GLM and other regression settings. Specifically, it references two papers: (Lee and Courtade, 2020) and (Zhang et al., 2020). These studies respectively present a minimax lower bound for vector-structured GLMs and a tight minimax lower bound in the Tucker-structured linear regression setting. However, neither of these papers address the LSR-structured GLM setting that our work explores.
>
> Furthermore, our proposed minimax lower bound is a **unified bound** - it encompasses minimax lower bounds for the Tucker and CP-structured GLM models that were introduced in (Zhou et al., 2013) and (Li et al., 2018), a feature that has yet to be examined in existing literature. This is made clear in Corollaries 1 and 2 in Section 6 of our work. This novel approach of providing theoretical guarantees in the structured-tensor GLM setting represents a significant contribution to the field.
>
> Finally, regarding the LSR decomposition presented in our study, we would like to politely disagree with the reviewer's assertion that it has been extensively examined, especially in relation to tensor-structured GLMs. As far as we are aware, an LSR-like decomposition has only been analyzed in "Learning Mixtures of Separable Dictionaries For Tensor Data" (2020). However, that study investigates the Dictionary Learning problem and imposes an "LSR" structure - defined as a sum of Kronecker-structured matrices - onto the dictionaries. The problem of representation learning is different from prediction: we focus on applying the LSR decomposition within tensor-structured GLMs, signifying a clear divergence in context and application.
>
> **_2. Technical Novelty of the paper:_** In response to the comment on the apparent straightforward extension of the minimax lower bound analysis from CP and Tucker models, we would first like to reaffirm, as noted in our discussion on conceptual novelty, that such analysis for CP and Tucker models in the context of GLMs is not yet established in the existing literature. This highlights a fundamental distinction in the scope and application of our work.
>
> We understand the perspective that using Fano's inequality for deriving minimax lower bounds might seem routine. We believe that the novelty lies in the construction of the packing set which gives insight into the structure of the space of LSR tensors. Traditional packing sets designed for vector-structured, low-rank matrix, or low-rank Tucker-structured GLMs do not capture this structure and would yield suboptimal lower bounds. Thus, while the end step of using Fano's inequality is standard, designing the right geometric insight into LSR tensors is novel.
>
> Another challenge in "simply" using Fano's inequality is actually bounding the relevant information-theoretic quantities. To that end, we derive tight bounds on the Kullback-Leibler divergence in GLMs, a task not yet tackled in the tensor setting to the best of our knowledge. These bounds can be used in other problems involving LSR tensors and distinguish our work from previous studies.
>
> **--- response continued in Part II ---**

---

> > ### Author Response · Authors · 2023-06-13
> > **Response to Reviewer sup2: Part II**
> >
> > **--- continuation of the response from Part I ---**
> >
> > **Revision addressing the "Requested Changes"**
> >
> > **_(Q1. and Q2.) Conceptual and technical novelty:_** We appreciate your suggestions and agree that further elaboration on these aspects would enrich our paper. In the revised manuscript, we plan to emphasize the conceptual novelty of the proposed LSR tensor models, as well as detail the specific difficulties encountered in deriving the minimax lower bound compared to other tensor low-rank models. To do this, we will revisit sections such as Section 3 (Problem Statement) and Section 6 (Minimax Lower Bound), refining the text to better highlight the conceptual and technical novelties of our work. **The revisions will draw from our previous discussions**, thus providing a more comprehensive understanding of the unique contributions and challenges presented by our research.
> >
> > **_(Q3.) Estimation error of the iterative solution:_** We appreciate your interest in the optimization error of our algorithm and acknowledge its significance within our research. Considering the novelty of applying LSR-structured models to GLM problems, numerous intriguing and complex questions arise, including the one about estimation error of an iterative algorithm. However, the first foray into a novel research area, like our study, must delicately balance between breadth and depth. While it's crucial to address important aspects like optimization error, undertaking an exhaustive exploration within the confines of the current paper could shift the focus from our core findings and contributions. Importantly, we wish to assure the reviewer that this topic is not being disregarded. In our manuscript's revision, we will underscore the importance of investigating optimization error, marking it as a significant direction for future research. Additionally, in order to demonstrate the empirical convergence of our iterative algorithm, we will augment our experiments on synthetic data to include a set of numerical results showcasing the convergence behavior of LSRTR. Similar to those shown in referenced work (Sun and Zhang, 2014), these results will depict the estimation error of the iterative solution as a function of iterations, providing insight into the convergence ability of LSRTR. We hope the reviewer acknowledges the challenging balance we are trying to strike in a work like ours and the strategic focus required in this context.
> >
> > **_(Q4.) Comparison to DNNs:_** Placing our research within the broader context of machine learning, especially when contrasting tensor-based regression models with Deep Neural Networks (DNNs), is indeed essential. As such, we will augment Section 7 (Experiments on Medical Imaging Data) in the revised manuscript to elucidate the advantages and performance of tensor-based models in relation to DNNs. The revision will highlight the strengths of regression models when working with limited sample sizes often associated with medical datasets from clinical studies. We will further illuminate the interpretability of regression models—an indispensable attribute in numerous applications—that DNNs often lack. To offer comprehensive insight, we'll reference relevant studies such as "Logistic regression and artificial neural network classification models: a methodology review" (2002), and "Classification Performance of Neural Networks Versus Logistic Regression Models: Evidence From Healthcare Practice" (2022). These studies underline that in various instances, logistic regression and neural networks show comparable performances. Briefly, while DNNs often outperform regression models in certain fields like computer vision due to their increased flexibility, potential overfitting in a small sample regime remains a concern. Moreover, regression models are frequently termed "white-box models" owing to the transparency and interpretability of their parameters. This interpretability allows for a detailed analysis of covariates' statistical significance and the generation of confidence intervals as posterior probabilities—qualities that are challenging to obtain with DNNs. These discussions will form the heart of our revision in Section 7.
> >
> > To further illustrate the efficacy of tensor-based regression models versus DNNs, we will also include in Section 7 a more comprehensive comparison between our proposed LSRTR algorithm and a variety of DNN benchmarks using the Vessel MNIST dataset from our study. This analysis will reference the benchmark work "MedMNIST Classification Decathlon: A Lightweight AutoML Benchmark for Medical Image Analysis" (2021). Although our LSRTR may be outperformed by some complex DNNs, it exhibits comparable and occasionally superior performance to several others. Given the earlier points about overfitting and interpretability—vital considerations in clinical studies and many other disciplines—we believe that our revision will provide a balanced viewpoint on the real-world application of tensor-based GLMs.

---

> > > ### Comment · Reviewer_sup2 · 2023-06-19
> > > **To the authors' feedback**
> > >
> > > Thanks for the authors' feedback. Based on the feedback, I acknowledge the following points:
> > >
> > > 1. The conceptual novelty of this work lies in the unique application of the LSR decomposition to the GLM model and the comprehensive analysis of these constructions. While the concepts themselves may not be new, their specific application and analysis in this work are novel.
> > >
> > > 2. The technical novelty is demonstrated through the construction of the packing set, which provides insights into the structure of the space of LSR tensors. Traditional packing sets designed for other types of models would yield suboptimal lower bounds in capturing this specific structure.
> > >
> > > 3. Regarding the optimization error, the authors acknowledge the need to strike a balance between breadth and depth in the work. They agree that providing an error bound for the iterative solution may not be necessary given the nature of the research.
> > >
> > > 4. While it is acknowledged that more complex deep neural networks (DNNs) may outperform LSRTR in certain scenarios, the LSRTR model still shows comparable and sometimes superior performance compared to several other models.
> > >
> > > These points addressed my previous concerns and provide a clearer understanding of the novel contributions and performance of the LSRTR model in the context of the research.

---

> > > > ### Author Response · Authors · 2023-06-28
> > > > **Revision Submitted**
> > > >
> > > > Thank you; we have submitted the revised manuscript. The requested discussion on the convergence/estimation error of the algorithm is in Section 5.1.4. We have also highlighted the conceptual novelty in Sections 1-3, the technical novelty in Section 6, and a more comprehensive discussion on DNNs in Section 5.

---

### Review · Reviewer_75Jv · 2023-06-13

**Summary Of Contributions:**

This study explores a new low-rank tensor model, named Low Separation Rank (LSR), within the context of Generalized Linear Model (GLM) problems. The LSR model expands upon established tensor models, Tucker and CANDECOMP/PARAFAC (CP), by imposing itself on the coefficient tensor in the GLM model. The researchers propose a block coordinate descent algorithm for parameter estimation in LSR-structured tensor GLMs. Additionally, a minimax lower bound is derived for the error threshold in estimating the coefficient tensor in LSR tensor GLM problems, indicating that its sample complexity could be significantly lower than vector-based GLMs. These results also apply to lower bounding the estimation error in CP and Tucker-structured GLMs. The derived bounds align with the strictest bounds in the literature for Tucker linear regression. The effectiveness of the LSR tensor model is demonstrated through numerical experiments on synthetic datasets for three regression types: linear, logistic, and Poisson. Tests on a selection of medical imaging datasets indicate the utility of the LSR model over other tensor models (Tucker and CP) with real, unbalanced data with limited samples.

**Audience:**

Yes

**Claims And Evidence:**

Yes

**Requested Changes:**

While the paper is generally well-presented, some statements could be better articulated to improve clarity and comprehension.

1. On page 5, “… the factor matrices $\mathbf{B}_k\in\mathbb{R}^{m\times{}r}\,\forall{}k\in[K]$ are low-rank matrices”. I understand the authors would like to say that $r<<m$, but it does not imply the low rankness of a B. A similar assertion also appears on page 7 and might need to be revisited.
2. On page 6, the phrase "- which implies that the coefficient vector b is composed of separable sub-matrices weighted by some vector g.. — is also quite restrictive" calls for additional explanation. It's not immediately evident that the Tucker model is more restrictive than the proposed LSR. The definition of LSR (as per Eq. (9) or Fig. 1b) suggests that the LSR could be modeled as a Tucker model, where the core is restricted to super-diagonal blocks. From this perspective, the relative restrictiveness of the Tucker and LSR models remains ambiguous.
3. The claim on page 8 stating, "....that since find the rank r of a tensor NP-hard, then finding the separation rank of a Kronecker structured matrix is also NP-hard" appears non-intuitive. A more detailed discussion or a more comprehensive justification would bolster this argument.
4. In the experimental section (page 26), the paper emphasizes the unbalance issues prevalent in real-world medical imaging data. However, the rationale as to why the proposed GLM should outperform other methods under these circumstances is not clear. Revisiting and elaborating on this point could strengthen the argument.

Questions:

1. On page 9, the paper notes, "the link function in GLMs makes part of our analysis non-trivial and fundamentally different to such works." Could the authors clarify which part of the analysis required non-trivial derivation, and what fundamental difference this brought to their work?

Typographical Errors:
1. In Equation (41), there is an extraneous closing bracket in "$l'\in[L]]$." This should be rectified for better readability.

**Strengths And Weaknesses:**

The overall strength of this paper lies mainly in its execution and its presentation of experimental data, although there are significant concerns regarding the novelty of the ideas and methods presented.

Strengths:

1. The overall quality of the writing in the paper is commendable. However, I suggest that some sentences could benefit from careful rewording for enhanced clarity.
2. The minimax lower bound for the proposed structured GLMs is proven with rigor. Even though my expertise does not extend into the realm of statistical learning, I found the proof to be understandable and illuminating.
3. The authors have conducted extensive and detailed experiments using real-world data, enhancing the value of their findings.

Weaknesses:

The key area where this paper falls short is in its novelty, or lack thereof.

1. The proposed LSR model does not seem to offer anything new. It appears to be equivalent to the block term decomposition suggested by L.D. Lathauwer in 2006, albeit with what might be a minor variation in the constraint imposed.
2. The algorithm used for estimation seems rather mundane and doesn't present anything innovative. Moreover, the analysis of its convergence is not adequately addressed, which could limit its applicability or validity.

---

> ### Author Response · Authors · 2023-06-19
> **Response to Reviewer 75Jv: Part I**
>
> We thank the reviewer for their careful reading of our manuscript and valuable feedback. In the following we first explain how we plan on addressing the reviewer's feedback as well as the requested changes in the revised version of the manuscript. We then address the general feedback from the reviewer. **Please note that this response is being broken into two parts because of the limit of 5000 characters in a comment.**
>
> However, first, and most importantly, we would like to address the novelty of our contributions. The main novelty of our work resides not in introducing concepts in tensor decompositions, but in **our unique application of the LSR decomposition to the GLM and our comprehensive analysis of the resulting tensor GLM, which is non-trivial**. Indeed, such tensor structures / tensor decompositions have not been examined in relation to their performance in tensor-structured GLMs. A more comprehensive discussion on conceptual and technical novelty of our work has also been provided in the response to reviewer sup2.
>
> **Revision addressing requested changes**
>
> **_(Q1) Low-rankness of $B$:_** We agree with the reviewer that the language around “low-rank tensors” in our work must be tightened. We assume that the coefficient tensor $B$ is “low-rank”, which implies a low Tucker rank $(r_1, \dots, r_K)$, or that $r_k \ll m_k $ . However, due to the orthogonality constraint, each factor matrix $B_{(k,s)}$ is, in fact, **exactly** of rank $r_k$ (a full-rank matrix, where $r_k$ is assumed to be very “small”). We will make this distinction clear in the revised manuscript, which we hope will render a more intelligible portrait of the LSR structure for our readers.
>
> **_(Q2) Restrictiveness of Tucker:_** We would like to thank the reviewer for introducing the work (“Decompositions of a Higher-Order Tensor in Block Terms”, L. De Lathauwer). Indeed, we acknowledge that the LSR model can be re-arranged as a special case of the Block Tensor Decomposition (BTD); see also the section “Revision addressing general comments” in the response below. This BTD structure is a specialized Tucker-structured tensor with core tensor of dimensions $Kr_1 \times \dots \times Kr_K$ and factor matrices of size $m_k \times Sr_k$. In our work, however, the language referred to by the reviewer is comparing the number of learnable parameters between tensor decompositions (CP, Tucker and LSR) of same sized core tensor (of same rank). In Section 2 (Preliminaries) and Section 3 (Problem Statement), we will discuss the BTD decomposition and its relation to LSR. We will also clarify that we fix the rank $(r_1, \dots r_K)$ when comparing tensor decompsitions, which is why, for a fixed rank, LSR generalizes, and is less restrictive than, the Tucker decomposition.
>
> **_(Q3) NP-hardness of finding the separation rank:_** The reviewer raises an important point regarding the tensor rank. We will enrich our discussion in Section 3.1 (Parameter Estimation for LSR-TGLM) regarding this point. Specifically, in Eq (10), the vectorized LSR-structured tensor shows a sum of $S$ Kronecker-structured matrices. Additionally, Lemma $1$ in the referenced work (Ghassemi et al., 2019) shows an isomorphic mapping such that a **Kronecker-structured matrix with separation rank $S$ can be rearranged into a rank $S$ tensor** . Additionally, the works (“Tensor rank is NP-complete”, 1990) and (“Most tensor problems are NP-hard, 2013) prove that finding the rank of a tensor is NP-hard, and thus so is finding the separation rank of a matrix. Therefore, in the context of our work on LSR-TGLM, finding the LSR rank $S$ is NP-hard.
>
> **_(Q4) Imbalanced Datasets:_** Thank you for the opportunity to clarify this point. Our assertion is not that tensor-structured GLMs will universally outperform all other predictive methods in the face of imbalanced datasets. Our argument is more nuanced. We have observed that medical imaging data often exhibit complex, high-dimensional and imbalanced characteristics, and we contend that our model, the LSR-TGLM, is particularly well-suited to handling such data. Our contention is rooted in the favorable bias-variance tradeoff and augmented representational power of the LSR-TGLM, which allows it to perform more robustly than more compact tensor-structured GLMs. To make this point more evident, we will elaborate on the advantages of the LSR-TGLM model in Section 7 (Experiments on Medical Imaging Data) in our revised manuscript. The additional clarification will underscore how the unique characteristics of the LSR-TGLM could enhance its performance when dealing with complex, high-dimensional, and imbalanced datasets.
>
> **--- response continued in Part II ---**

---

> > ### Author Response · Authors · 2023-06-19
> > **Response to Reviewer 75Jv: Part II**
> >
> > **--- continuation of the response from Part I ---**
> >
> > **_Additional question on GLM link function:_** We will add a more comprehensive discussion on page 9 to elaborate on the non-trivial nature of our analysis. Briefly, a key component in the analysis of the minimax bound is deriving tight bounds on the Kullback-Leibler (KL) divergence in tensor-structured GLMs, which includes the analysis of the link function with an LSR-structured coefficient tensor. This is a non-trivial derivation, as analyzing the link function entails the careful construction of a packing set and the derivation of tight bounds on the packing distance as in equation (46). We will direct our interested readers to the Proof of Lemma 7, specifically the steps between equation (76) and (79) for a clearer understanding on how the analysis of the KL divergence involves both the link function and the bounds of equation (46).
> >
> > **Revision addressing general comments**
> >
> > *LSR structure and BTD*: We thank the reviewer for drawing comparisons between our proposed LSR model and the Block Tensor Decomposition (BTD) discussed in L. De Lathauwer's 2006 paper. This provides us with an opportunity to underline both the similarities and crucial differences.
> >
> > Indeed, the BTD can be rearranged into a specialized Tucker tensor, and its contributions have been widely acknowledged, as highlighted in works such as ("Block-Term Tensor Decomposition: Model Selection and Computation", 2020), and ("Combining Large-Scale Matrix and Tensor Decomposition with Structured Factors”, 2020). It's accurate to view our LSR tensor structure as a specialized form of BTD, *equipped with orthogonality constraints and common diagonal tensor blocks.*
> >
> > However, as mentioned in our overarching response, the primary novelty of our work resides in our distinctive application of LSR decomposition to the GLM and our in-depth analysis of the ensuing tensor GLM. Additionally, the LSR tensor structure in our work draws inspiration from the Low Separation Rank (LSR) **matrix** structure, as examined in the works by (Tsiligkaridis & Hero, 2013), and (Ghassemi et al., 2019). In our LSR-TGLM framework, this structure becomes relevant when the coefficient tensor is vectorized, as outlined in equations (9) and (10) of our paper. While the differences between LSR and a general BTD might appear nuanced, they are significant, particularly the requirement for a common core tensor in the LSR structure. This distinction is fundamental for retaining the LSR matrix structure, thereby offering a number of benefits.
> >
> > Further elaboration on these points, along with the uniqueness of our approach and its novelty, can be found in our responses to reviewer sup2 and (Q1) for reviewer zqEm. We trust that these responses, combined with our initial assertion at the beginning of this reply, make the innovative aspects of our work clear.
> >
> > *The BCD algorithm*: We appreciate the reviewer's comments regarding the Block Coordinate Descent (BCD) algorithm used in our study. Indeed, while BCD may not present a novel approach in the traditional sense, its application in the LSR-TGLM setting via the LSRTR algorithm is the first of its kind. We have provided numerical results in Section 5 (Numerical Analysis) and Section 7 (Experiments on Medical Imaging Data) as empirical evidence of the algorithm's practical behaviour. Though BCD algorithms, commonly used in the tensor regression literature, as referenced in works (Tan et al., 2012), (Zhou et al., 2013), (Zhang and Jiang, 2016), and (Li et al., 2018), do not directly exploit the tensors' structure, they still prove efficient and effective. Further, there do exist efficient algorithms in the literature, such as those in referenced work (Ghassemi et al., 2019), that exploit the separation rank matrix structure, serving as an impetus for developing further algorithms in future works.
> >
> > Regarding the convergence analysis of the proposed algorithms, the constraint set in LSRTR is a product space of Stiefel manifolds and neither closed nor convex. This makes a convergence analysis for BCD a challenge since we cannot apply general results for the convergence of BCD as in ("Non-linear Programming", 1999). We will augment Section 4.3 (Final Algorithm: LSRTR) with a brief discussion of the practical and theoretical convergence of the proposed BCD algorithm. We will also supplement Section 5 (Numerical Study) with an additional set of numerical results. These results will depict the convergence behaviour of LSRTR, i.e., the normalized error as a function of increasing number of iterations. We hope that this discussion will enrich the current manuscript and provide interested readers with a more comprehensive understanding of the performance of the proposed algorithm.

---

> > > ### Author Response · Authors · 2023-06-28
> > > **Revision Submitted**
> > >
> > > Dear Reviewer 75Jv, please know that we have submitted a revision of our original manuscript as per the plan stated in the response letter. We look forward to further feedback from you. Thank you.

---

### Decision · Action_Editors · 2023-07-09

**Recommendation:** Accept with minor revision

**Comment:**

This work considers a low-rank tensor factorization model called low separation rank (LSR) for generalized linear models. The authors provide a comprehensive analysis of the resulting nonconvex problems involved in estimating the tensor factors. Specifically, they derive a tight minimax lower bound for estimating the coefficient tensor, which captures the intrinsic degrees of freedom in the LSR tensor. The work also proposes a block coordinate descent algorithm to solve the nonconvex problem and estimate the tensor factors. Numerical experiments on both synthetic and real datasets demonstrate the usefulness of the LSR model for various generalized linear models.

The main concern raised by the reviewers is the clarification of the novelty, as the LSR tensor model is not new. To address this comment, the authors have revised the paper and clearly highlighted the major contribution, which is the introduction of the LSR tensor model for generalized linear models rather than the LSR itself, and the comprehensive analysis of the resulting nonconvex problem. All the reviewers acknowledge the revisions and the contribution of the paper, and recommend acceptance or leaning towards acceptance. I agree with the reviewers and recommend acceptance.

Very minor comment: both $\Sigma_x$ and ${\mathbf\Sigma}_x$ (bolded) are used in Section 6. The authors may unify them.

**Audience:**

The work could be of interest to researchers working on tensor structures for machine learning, such as tensor factorization, topic modeling, tensor networks, etc.

**Claims And Evidence:**

This work considers a low-rank tensor factorization model called low separation rank (LSR) for generalized linear models. The authors provide a comprehensive analysis of the resulting nonconvex problems involved in estimating the tensor factors. Specifically, they derive a tight minimax lower bound for estimating the coefficient tensor, which captures the intrinsic degrees of freedom in the LSR tensor. The work also proposes a block coordinate descent algorithm to solve the nonconvex problem and estimate the tensor factors. Numerical experiments on both synthetic and real datasets demonstrate the usefulness of the LSR model for various generalized linear models.

Overall, the derived minimax low bound is technically challenging and is accompanied by detailed proofs. The performance of the proposed approach and algorithm is supported by various experimental results.

---

> ### Author Response · Authors · 2023-08-04
> **Camera-Ready Submission**
>
> Dear Action Editor, please know that we have submitted the Camera-Ready of our manuscript.